# Human autoimmunity at single cell resolution in aplastic anemia before and after effective immunotherapy

Zhijie Wu [1,4] ✉, Shouguo Gao [1,4], Xingmin Feng[1,4], Haoran Li [1], Nicolas Sompairac [2], Shirin Jamshidi [2], Desmond Choy [2], Rita Antunes Dos Reis [2], Qingyan Gao[1], Sachiko Kajigaya[1], Lemlem Alemu[1], Diego Quinones Raffo[1], Emma M. Groarke [1], Shahram Kordasti [2,3,5], Bhavisha A. Patel[1,5] & Neal S. Young[1,5]

Severe immune aplastic anemia is a fatal disease due to the destruction of marrow hematopoietic cells by cytotoxic lymphocytes, serving as a paradigm for marrow failure syndromes and autoimmune diseases. To better understand its pathophysiology, we apply advanced single cell methodologies, including mass cytometry, single-cell RNA, and TCR/BCR sequencing, to patient samples from a clinical trial of immunosuppression and growth factor stimulation. We observe opposing changes in the abundance of myeloid cells and T cells, with T cell clonal expansion dominated by effector memory cells. Therapy reduces and suppresses cytotoxic T cells, but new T cell clones emerge hindering robust hematopoietic recovery. Enhanced cell-cell interactions including between hematopoietic cells and immune cells, in particular evolving IFNG and IFNGR, are noted in patients and are suppressed post-therapy. Hematologic recovery occurs with increases in the progenitor rather than stem cells. Genetic predispositions linked to immune activation genes enhances cytotoxic T cell activity and crosstalk with target cells.

Aplastic anemia (AA) was recognized well over a century ago, by Paul Ehrlich, and named by Louis Henri Vaquez. Considered rare in Western countries, AA is far more common in Asia[1-3]. Severe aplastic anemia (SAA) presenting in the community is fatal if untreated, and effective therapies, stem cell transplant and immunosuppression, have reversed the dire prognosis—a success story in twentieth-century hematology[4].

The disease AA can be separated from both much less frequent congenital syndromes, which include bone marrow (BM) hypocellularity and frequent iatrogenic suppression of blood cell production due to chemo- and radiotherapy marrow injury. Immune AA has clinical similarity to immune-mediated hematologic diseases: large granular lymphocytosis, paroxysmal nocturnal hemoglobinuria,

hypoplastic myelodysplastic syndrome, and single lineage syndromes like pure red cell aplasia[5]. Additionally, immune AA shares pathophysiologic features with other human autoimmune diseases, such as multiple sclerosis, uveitis, and type I diabetes. The study of AA is advantaged by the ready availability of blood and marrow cells, and techniques to identify, separate, grow, and manipulate them in vitro.

The best evidence for an immune pathophysiology in AA derives from the clinical response to immunosuppressive treatments[6]; dependence of blood counts on continuation of such therapies[7,8]; and the occurrence of AA as a rare complication of transfusion of reactive lymphocytes[9] as well as a toxicity of cytotoxic lymphocyte activation in

[1]Hematology Branch, National Heart, Lung, and Blood Institute, National Institutes of Health, Bethesda, MD, USA. [2]Comprehensive Cancer Centre, School of Cancer and Pharmaceutical Sciences, Faculty of Life Sciences and Medicine, King's College London, London, UK. [3]Haematology Department, Guy's Hospital, London, UK. [4]These authors contributed equally: Zhijie Wu, Shouguo Gao, Xingmin Feng. [5]These authors jointly supervised this work: Shahram Kordasti, Bhavisha A. Patel, Neal S. Young. ✉e-mail: zhijie.wu@nih.gov

cancer immunotherapies[10–12]. Murine models support specific immune mechanisms of marrow destruction and drug mechanisms of action[13–18].

Many tissue culture experiments also provide evidence of immune pathophysiology. Examples include colony culture and co-culture[19,20], cell surface phenotyping by flow cytometry[21–23], bulk analysis of RNA transcriptomes[24], and detection of cytokines in the circulation and within cells[25–28]. However, most of these methods require extensive and non-physiological manipulations and conditions, limitations of starting cell number, and unpredictable cell loss. Most in vitro experiments have been conducted with a necessarily limited focus of interest, due to the paucity of marrow cells in AA and the rarity of the disease. Reproducibility of results among laboratories is not insignificant and is likely due to differences in technique. Remarkably, the pathophysiology of AA continues to be ascribed to an "immune hypothesis."

Advanced single cell methodologies—single cell DNA genomics, RNA sequencing, and proteomics, as well as high-resolution time-of-flight cytometry (CyTOF)—combined with computational algorithms for the large amount of generated information, afford an extraordinary opportunity for dramatic visualization and deep analysis of the affected marrow compartment, with minimal laboratory manipulation, comprehensive inclusion of all retrievable cells, and multiplexing of orthogonal datasets. We and others have reported the successful application of these methods to human hematologic, genetic, and immune diseases[29–35], in which modern methods both confirm earlier conclusions from simpler approaches and provide startling new insights into pathophysiology.

In this study, we apply advanced single-cell methodologies to primary BM samples from SAA patients, in the context of in vivo perturbation: before and after current first-line therapy. We provide a comprehensive transcriptome atlas of target hematopoietic cells and effector immune cells in these patients, and investigate the dynamic changes of cell populations, gene programs, T cell clonality, and association with response to treatment. Our findings have implications not only for BM failure syndromes but for other human autoimmune diseases with analogous pathophysiology.

## Results
### Patient cohort
The current work is unique in that a prospective clinical protocol (NCT01623167) was the basis for sample collection and specific experiments. Fresh BM samples were obtained from 20 out of 137 persons with SAA who were treated with hATG, cyclosporine and eltrombopag in combination, a standard immunosuppression regimen (IST, Supplementary Fig. 1). The median age (36.5 with range 11–73), sex distribution, and severity of blood counts were typical of immune AA. At the primary efficacy endpoint (6 months after IST), 15 patients had achieved hematopoietic response (R), 4 patients had withdrawn from the protocol, and 1 was a nonresponder (NR). With a median follow-up time of 24 months, 5 responders had a relapse requiring re-initiation of IST and 4 had developed secondary myeloid malignancy (Fig. 1b, Supplementary Fig. 1d, Supplementary Data 1). Immunoprofiling of hematopoietic stem and progenitor cells (HSPCs), cytometry by time-of-flight (CyTOF), and single-cell RNA sequencing (scRNA-seq), and single-cell T cell receptor/B cell receptor sequencing (scTCR/BCR-seq) were performed for 93 BM samples and sorted cell populations of patients, and 16 total or sorted samples of 4 healthy donors (Fig. 1a).

### Broad, low-resolution assessment of the bone marrow by single-cell sequencing and cytometry
CyTOF showed marked differences between AA samples at baseline relative to healthy donors. Unsupervised cell clusters were annotated as granulocytic lineage, monocytic lineage, NK cells, T and B lymphocytes and abundance analysis using Milo[36] was

performed (Fig. 1c); as expected, proportions of CD8+ and CD4+ T cells were overrepresented while myeloid cells were lower in all patients' samples (Fig. 1e–g). Subpopulation analyses within cell types, for example CD8+ T cells, revealed a less uniform pattern of abundance (Supplementary Fig. 2). After IST, percentages of CD8+ T, CD4+ T, NK, and B cells decreased and myeloid cells increased; however, subgroup clustering within each cell type again revealed more complicated and heterogenous responses (Supplementary Fig. 3). These results were confirmed using a different cell-type assignment methodology and analysis (Supplementary Data 2, Supplementary Figs. 4–9, Fig. 1d).

Using scRNA-seq, we explored functional heterogeneity, transcriptome programs, TCR/BCR usage, and their dynamics after treatment in paired samples from SAA. BM mononuclear cells (BMMNCs) clusters were annotated to major cell types, including HSPCs and differentiated cells of all lineages (Fig. 1h), and expression of well-established marker genes was plotted in Supplementary Fig. 9e. Similar to CyTOF, there were increased T lymphocytes and decreased myeloid cells before treatment compared with healthy donors, and decreased lymphocytes and recovery of myeloid cells after treatment (Fig. 1i).

### High-resolution visualization of target and effector cell populations and imputed function by single-cell methodologies
scRNA-seq and CyTOF can discriminate among subpopulations and allow inferences of function from the genes and gene pathways expressed (or suppressed), corresponding cell membrane and intracellular proteins. As examples, the stem and progenitor cells and lineage-committed precursor cells can be denominated, and naïve, memory, effector, and regulatory cell subsets, as well as their state of activation and exhaustion, clonality, and dynamics, can be distinguished, as discussed in detail below.

### Loss of early hematopoietic stem and progenitor cells and recovery of late-stage cells after treatment
Overall, lineage−CD34+ HSPCs were much reduced in pre-treatment samples and recovered after treatment. However, for more primitive CD34+CD38− stem and multipotent progenitors, there was a notable lack of recovery at 6 months, while CD34+CD38+ differentiated lineage-specified progenitor cells (including granulocytic and monocytic progenitors, GMPs; common myeloid progenitors/progenitors of erythroid-megakaryocytes, CMPs/MEPs; and lymphoid progenitors, LymPs) dramatically increased (Fig. 2a, b).

From scRNA-seq data, sorted CD34+ HSPCs were clustered to include hematopoietic stem cells and multipotent progenitor cells (HSCs), MEPs, GMPs, and LymPs (Fig. 2c), validated by expression of lineage-specific genes, indicating differentiation from HSCs to ME, GM, and Lym lineages (Supplementary Fig. 10). CD34+ HSPCs from SAA patients had higher expression of inflammatory pathways, including hallmark IFN-γ and IFN-α, which were downregulated post-treatment. Conversely, DNA damage repair and cell cycling genes were downregulated at baseline and showed upregulation after therapy, accompanied by increased expression of genes in cell stress pathways (unfolded protein response, Fig. 2d).

Both CyTOF and scRNA-seq data showed incomplete hematopoietic recovery and continued immune dysregulation after treatment (Fig. 2e). There were dramatic differences in gene expression in BM between SAA patients and healthy donors, but, the variation was less apparent between pre- and post-treatment samples (Fig. 2f). Differentiation potentials of HSCs to MEP, GMP, and LymP trajectories in pre-treatment, post-treatment, and healthy samples were comparable, appropriately governed by key transcriptional factors (Fig. 2g, h), supporting the inference that destruction of target cells occur after lineage specification and differentiation rather than biased cell fate determination.

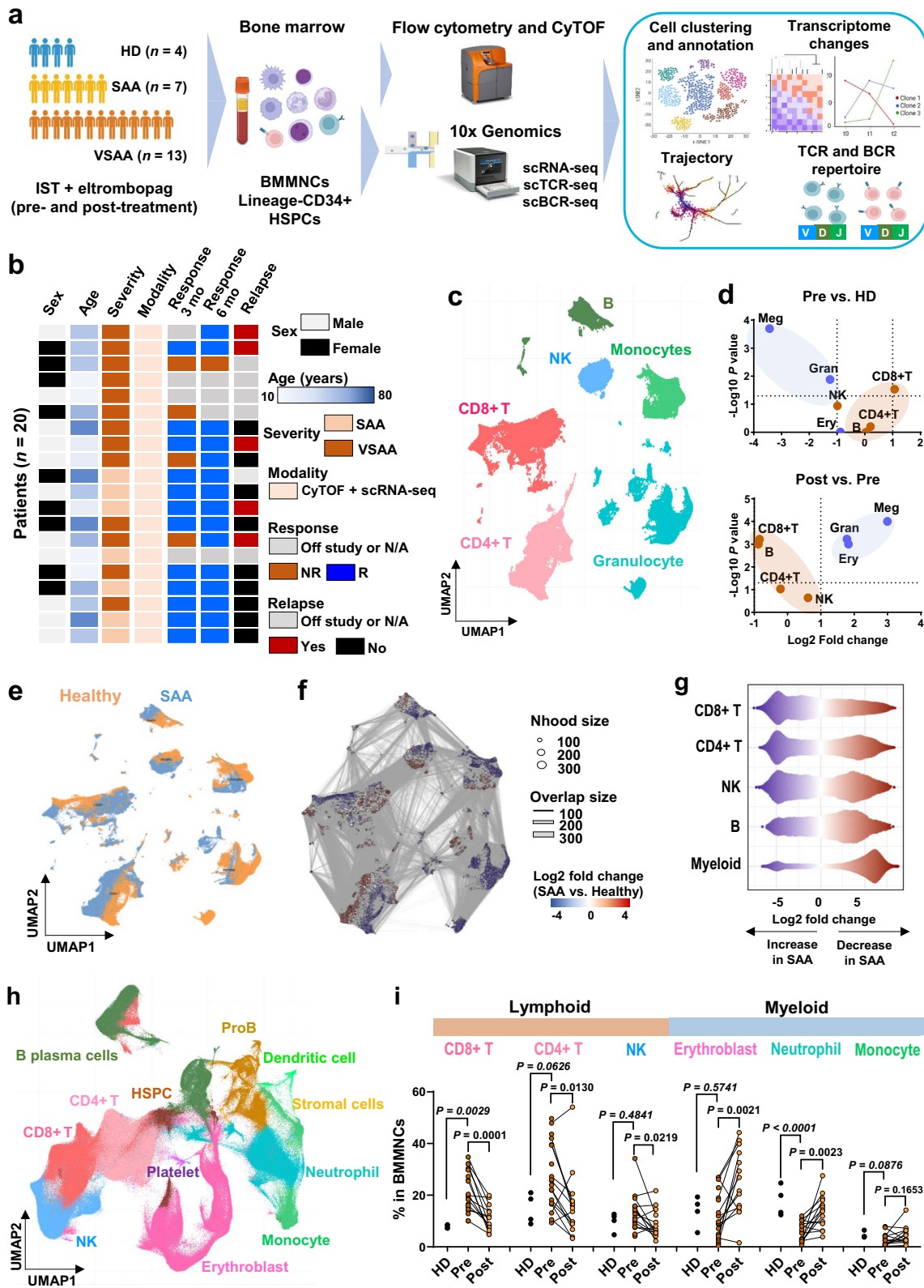

On the assumption that eltrombopag stimulated late-stage HSCs rather than stem cells[37], dynamic changes of HSC numbers distributed along pseudotime differentiation were examined. For pre-treatment samples, there were less early HSCs for multilineage differentiation, suggesting that immune targeting of early stem cells resulted in BM failure. Late-stage HSCs contributed to trilineage differentiation post-treatment (Fig. 2i).

**CD8⁺ T cells and CD4⁺ T cell subpopulations expanded and activated in SAA**

CD8⁺ T cells from scRNA-seq data of total BMMNCs, plotted along pseudotime (Slingshot[38]) (Fig. 3a), showed progressive differentiation from naïve T cells to central memory and then to effector T cells, validating the Velocity analysis[39] (Fig. 3b). Frequencies of effector memory CD8⁺ T cells were higher at baseline and decreased after

**Fig. 1 | Resolving hematopoietic and immune cell complexity using CyTOF and scRNA-seq. a** Experimental workflow. BMMNC samples from patients and healthy donors (HD) were analyzed by CyTOF and flow cytometry to profile hematopoietic and immune cells and subpopulations. BMMNCs and Lineage⁻CD34⁺ cells were subjected to scRNA-seq, followed by data analysis including single-cell transcriptome profiling (gene expression and cell–cell interaction) and scTCR/BCR profiling. Created in BioRender. Wu, Z. (2025) https://BioRender.com/t6tfzun. **b** Characterization of the patient cohort. **c** UMAP projection of BMMNCs at all time points from patients and healthy donors in CyTOF. **d** Log fold changes and log *P* values (with the two-sided unpaired Mann–Whitney test) of cell abundance between pre-treatment samples of SAA patients and controls (top) and post-treatment samples (bottom). Red dots represent lymphoid cells, and blue dots represent myeloid cells. **e** UMAP embedding of CyTOF data as shown in Fig. 1c, colored by differential abundance. Red indicates clusters with patients' cells (pre-treatment samples) constituting >95% of cells, and blue clusters with healthy donors' cells >95%. **f** Graphic of Nhoods identified by Milo. Nodes are Nhoods, colored by log2 fold-changes (log2FC) between pre-treatment samples of SAA patients (n = 20) and healthy donors (n = 4). Nondifferential abundance Nhoods (FDR ≥ 0.1) are colored white, and sizes correspond to the number of cells in a Nhood. Graph edges depict the number of cells shared between adjacent Nhoods. **g** beeswarm plot showing adjusted log2FC of cell populations abundance between pre-treatment samples of SAA patients and healthy donors in Nhoods. **h** A UMAP plot of single-cell gene expression of BMMNCs samples at different time points of patients (n = 45) and healthy donors (n = 4). Cells are colored by type (HSPC, ProB as B cell progenitors, erythroblast, neutrophil lineage, monocytic lineage, CD8⁺ T cell, CD4⁺ T cell, NK cell, B cell and plasma cell, dendritic cell, stromal cell, and platelet). **i** Frequency (% in BMMNCs) of CD8⁺ T, CD4⁺ T, NK, erythroblast, neutrophil, and monocyte in pre- (n = 20), post-treatment (n = 14) samples of SAA patients and healthy donors (n = 4). P values with the two-sided unpaired and paired Mann–Whitney test are shown.

treatment, with relative increases in naïve and central memory CD8⁺ T cells (Fig. 3c). The genetic programs of CD8⁺ T cells showed progressive gain in effector memory cell markers and loss of naïve cell markers, and increased cytotoxicity gene transcription with differentiation, a pattern more prominent in patients than in healthy individuals (Fig. 3d). AA samples exhibited higher CD8⁺ effector memory potential and increased cell cytotoxicity compared to healthy donors (Fig. 3e).

We integrated CyTOF data from pre- and post-treatment samples and healthy donors, and constructed density plots of CD8⁺ and CD4⁺ T cells based on Binaryclust data[40]; naïve CD8⁺ T (cluster 4) and naïve CD4⁺ T (cluster 18) cells were dominant in healthy, nearly absent in pre-treatment SAA samples and recovered after treatment. CD8⁺ effector T cells (cluster 5) and central memory cells (cluster 15), as well as CD4⁺ central memory T cells (cluster 16), were prominent in baseline samples and decreased after treatment; a pattern also observed with scRNA-seq (Fig. 4a, b).

CD4⁺ regulatory T cells (Tregs) abundance was similar in SAA samples before and after treatment and also similar to controls (Supplementary Fig. 5). Both Treg-A (Cluster 6: CD4⁺CD25⁺FOXP3⁺CD95⁻) and Treg-B (Cluster 8: CD4⁺CD25⁺FOXP3⁺CD95⁺)[41] decreased after treatment (Fig. 4b, Supplementary Fig. 5). We calculated a ratio of Treg versus CD8⁺ effector cells; the ratio of Treg-B (but not Treg-A) to CD8⁺ effector cells (cluster 2 or 5) was increased after treatment relative to pre-treatment (Fig. 4c), consistent with earlier reports of the potential modulatory benefit of Treg-B in AA[41,42]. A cluster of nonA or nonB Tregs (Cluster 11) in CD4⁺ T cells was lower in pre-treatment than in healthy and increased after treatment (Fig. 4b).

To explore transcriptome features of T cells that were identified by CyTOF, we annotated CD8⁺ and CD4⁺ T cell subpopulations in BMMNCs using scRNA-seq. Effector memory CD8⁺ T cells highly expressed CD57 and showed a high cytotoxicity score (Fig. 4d). Among CD4⁺ T cells, Treg-B were transcriptionally defined in a confined cell population while Treg-A were more diverse (Fig. 4e). CD4⁺Treg-B were more enriched in the effector memory population (Fig. 4f), and there was increased IFN-γ and IFN-α signaling in Treg-B pre-treatment compared to healthy donors, which decreased after treatment, indicating an activated phenotype (Fig. 4g, h). The Fas/FasL pathway score was lower in Treg-B from SAA samples compared to healthy, likely suggesting resistance to highly inflammatory environment of BM in SAA (Supplementary Fig. 5f).

NK cells, annotated as CD56^bright, CD56^early dim, CD56^dim, and adaptive NK cells, were similar in patients' pre-treatment samples as in healthy donors, but CD56^bright NK cells increased and CD56^early dim NK cells decreased after treatment, also several CD57⁺ subclusters of NK cells were higher in pre-treatment samples than in healthy donors, and decreased after treatment (Supplementary Figs. 8h and 13d), suggesting a "switch" of activated NK cells to naïve NK cells.

**Effector memory CD8⁺ T cells were clonally expanded in SAA, with TCR usage highly individual-private**

T cells were clonally expanded in AA samples. Abundancy of clonally expanded T cells (>2 cells with an identical TCR) was nearly double in pre-treatment samples as compared to healthy (average 25% vs 10%). Post-therapy, clonal expansion was not attenuated, and remained at 23% of average and with an equivalent Gini index of TCR clonality (Fig. 5a, b, Supplementary Fig. 14a–c). No significant differences were noted in BCR clonality (Supplementary Fig. 14d–f). Expanded TCR clones compared with all other T cells had higher IFN-γ and cytotoxicity gene expression. Phenotypically, clonally expanded T cells were mainly of the effector memory population (Fig. 5c). These results were consistent using different cut-offs in defining T cell clonal expansion (Supplementary Fig. 15a–c).

TCR usage in SAA was individual-private rather than disease-specific. There was very little sharing of top TCR clones among patients; similar findings were observed among healthy donors (Supplementary Fig. 15). TCR sequences in SAA patients were examined for inference of potential antigens, for which TCR data from the current cohort (20 patients and 4 healthy donors) were merged with TCR data from our previously published AA cohort (12 patients, and 9 healthy donors)[43]. Four hundred and fifty-five CDR3 sequences had a higher frequency in SAA than in the generation probability. TCR sequences in SAA patients were annotated to be related with autoimmune, infectious and neoplasm diseases, higher percentages than in healthy donors, and they were related to common pathogens including SARS-COV2, human herpes virus, and influenza virus; but these pathogen-specific TCR sequences present with a similar frequence in patients and healthy donors (Supplementary Figs. 16 and 17). GLIPH2 was used to explore common TCR clusters and differential abundance in patients and controls (Supplementary Data 3); 24 TCR groups were significantly more frequent among SAA patients. Each TCR cluster consisted of TCRs from multiple patients (P < 0.05, Supplementary Fig. 18), indicating more homogenous TCR usage in SAA than in the general population, but potential antigens (if any) inducing an aberrant immune response in SAA could not be identified due to limitations of study cohorts (patient number and heterogenous HLA background) and existing limited and biased virus databases.

**T cell clonal dynamics and hematopoietic recovery after immune suppression**

T cell clonal size dynamics after treatment was categorized into 3 groups: increased (>2 fold, treatment-resistant), unchanged (0.5–2 fold, treatment-insensitive), and decreased (<0.5 fold, treatment-sensitive). IFN-γ signaling in T cells was uniformly decreased post-treatment in all three groups; a more dramatic decrease in cytotoxicity and increase in cell apoptosis were observed in unchanged and decreased clones compared to increased clones (Fig. 5d). For statistical

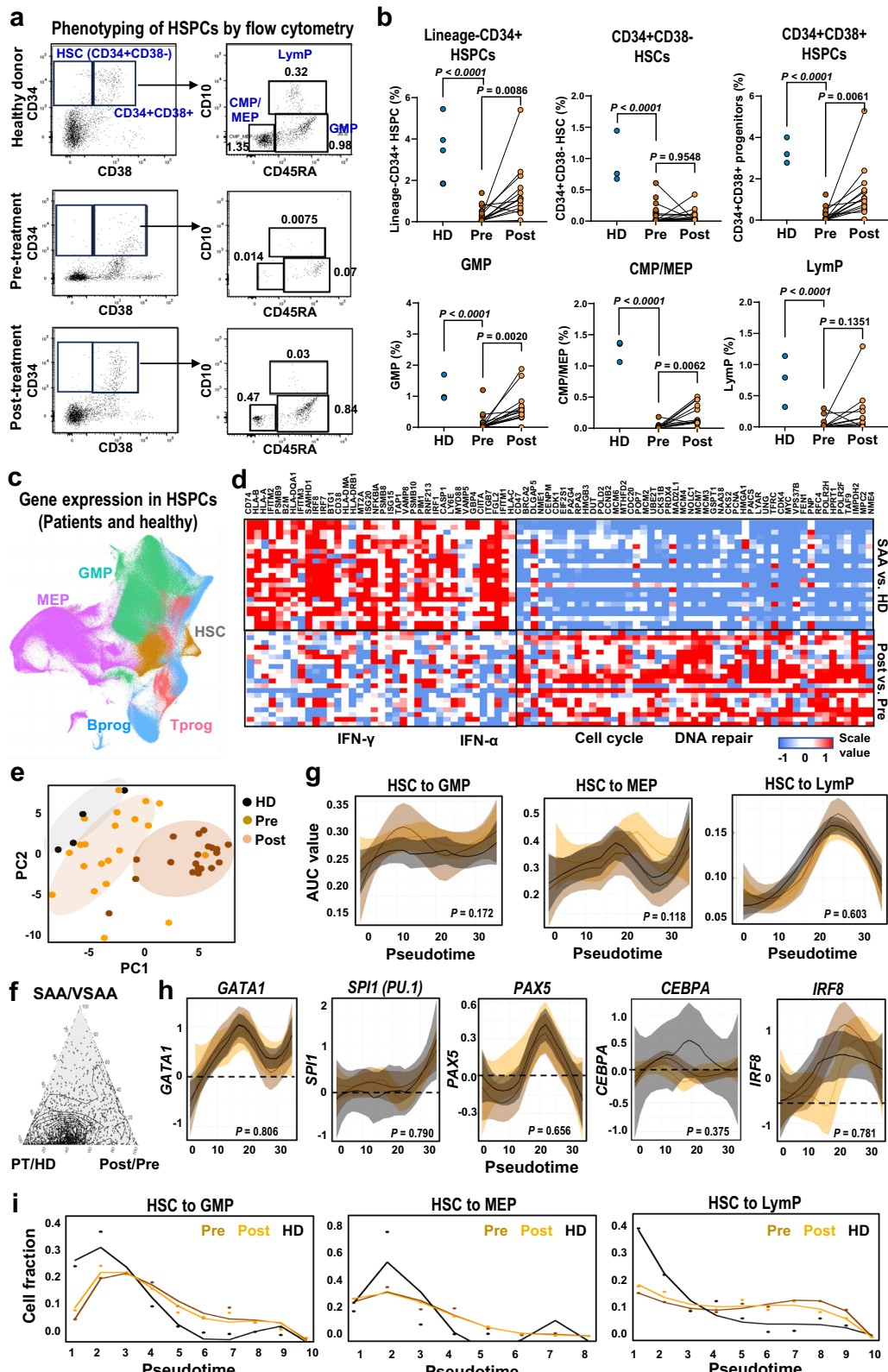

determination, we also defined clone size dynamics using an alternative approach, edgeR[44,45], and consistent results were obtained in categorizing clone size dynamics and transcriptomic changes (Supplementary Fig. 15d, e). These results suggest suppression of T cell functions generally with IST, regardless of clone size changes post-treatment. Transcriptional score of cytotoxicity and apoptosis associated with a treatment-resistant subset of T cell clones.

Among 16 patients who were evaluable at 6 months, 9 of them had clone size increased and 7 decreased (Fig. 6a, b, Supplementary Fig. 20). A clone size change was inversely correlated with diversity of TCR usage (using Gini index) and with robustness of blood count recovery (Supplementary Fig. 21a–c). New clones, not present at baseline, predominantly contributed to clone size increase after treatment; pre-existing clone size decreased in all but two cases

**Fig. 2 | Loss of early-stage and recovery of late-stage HSPCs after treatment.** **a** Representative plots of phenotypes of HSPCs in healthy donors (HD) and patients by flow cytometry. **b** Proportions of progenitor populations were compared between pre- ($n = 20$) and post-treatment samples ($n = 16$) and HD ($n = 4$). $P$ values with the two-sided unpaired and paired Mann–Whitney test. **c** A UMAP plot of single-cell gene expression in HSPCs of all patients and HD, colored by cell types as HSC, MEP, GMP, Tprog, and Bprog. **d** A heatmap of representative differentially expressed genes (grouped by functional pathways) of HSPCs between patients ($n = 20$) and HD ($n = 4$) on the top panel, between post- ($n = 16$) and pre-treatment samples ($n = 20$) at the bottom. Values are presented as log2FC. **e** A principal component analysis plot of samples based on cell cluster abundance measured by CyTOF. Pre- and post-treatment samples and samples from HD were clustered separately. **f** A plot showing fraction of variance (gene expression of BMMNCs) explained by diseases (PT/HD), disease severity (SAA/VSAA), and treatment (Post/Pre-immunosuppression). **g** Dynamic changes of lineage priming of HSCs to LymP,

GMP, and MEP, along with pseudotime differentiation. $X$-axis, pseudotime ordering from HSCs to lineage-restricted progenitors estimated by Palantir. $Y$-axis, Log(lineage signature gene expression) in pre-treatment (dark orange), post-treatment samples (orange), and HD (gray). Data are presented as mean with 1.96SE. $P$ values with the Analysis of Covariance (ANCOVA). **h** Dynamic changes of expression levels of transcription factors along pseudotime differentiation. $Y$-axis, expression of transcription factors in pre-treatment (dark orange), post-treatment samples (orange), and healthy donors (gray). Data are presented as mean with 1.96SE. $P$ values with ANCOVA. **i** Dynamic changes of lineage primed HSC fraction in pre-treatment (dark orange), post-treatment samples (orange), and HD (gray) along differentiation. $Y$-axis, Log(cell numbers of HSCs). For each lineage, cell fractions of early (pseudotime 1–4 on $x$-axis) and late stages (pseudotime 5–10) were compared: a ratio in pre-treatment/HD, and a ratio in post-treatment/pre-treatment. $P$ values with the two-sided unpaired $t$-test: HSC to GMP, 0.016 and 0.007; HSC to MEP, 0.929 and 0.96; HSC to LymP, 0.004 and 0.029.

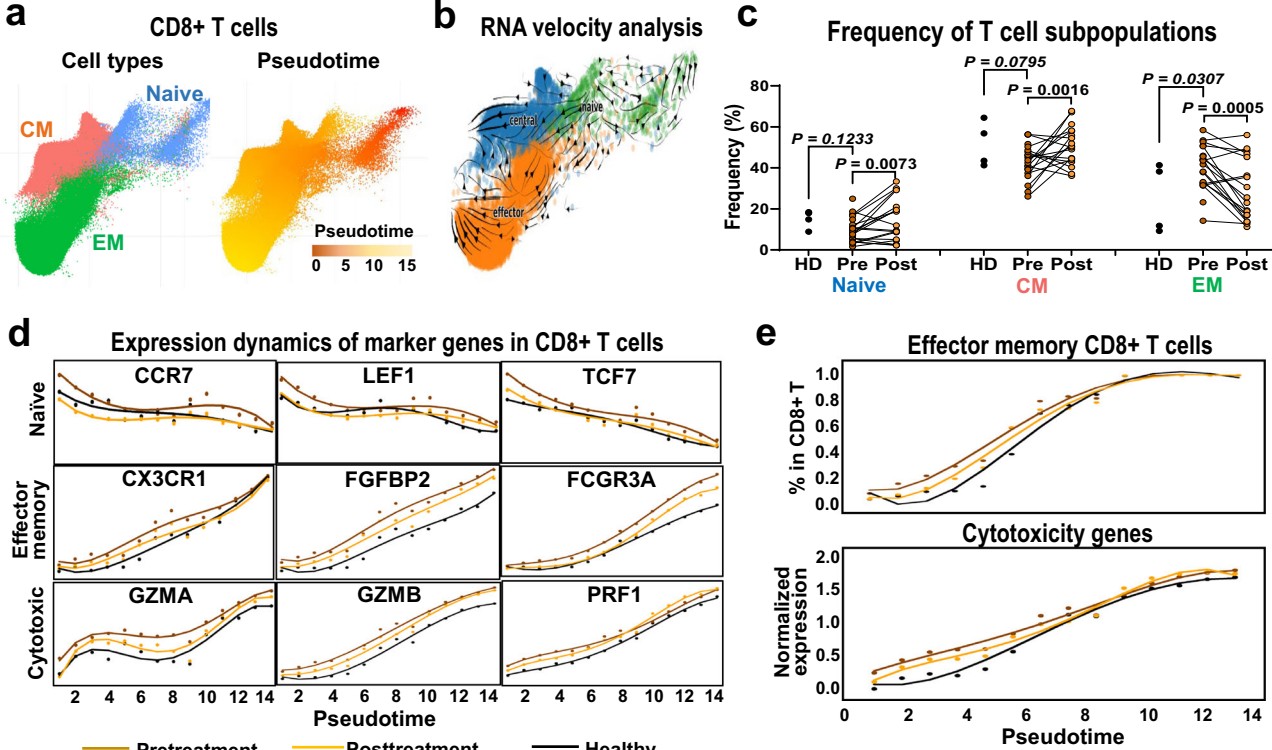

**Fig. 3 | CD8⁺ T cells expanded and exhibited cytotoxicity in SAA. a** In a UMAP only showing CD8⁺ T cells, subtypes including naïve, central memory (CM), and effector memory (EM) were indicated. Pseudotime based on Slingshot was calculated and colored accordingly. **b** RNA velocity analysis of CD8⁺ T cell phenotypes independently confirmed the differentiation trajectory. **c** Dot plots showing frequency of naïve, CM, and EM subpopulations of CD8⁺ T cells in pre- ($n = 20$), post-

treatment ($n = 18$) samples of SAA patients and healthy donors ($n = 4$). $P$ values with the two-sided unpaired and paired Mann–Whitney test are shown. **d** Plots showing expression dynamics of marker or functional genes of CD8⁺ T cells in pre- and post-treatment samples in SAA patients and healthy donors, along pseudotime. **e** Plots of frequency of effector memory T cells and cytotoxicity gene expression of CD8⁺ T cells in pre- and post-treatment samples of SAA patients and healthy donors.

(Fig. 6c, d). Acquisition of larger new clones negatively correlated with hematopoietic recovery (Fig. 6e, Supplementary Fig. 21d). We focused on dominant clones that diminished after treatment (treatment-sensitive) and new clones arising only after treatment (treatment-resistant). Compared with other clones, treatment-sensitive clones at baseline and treatment-resistant clones that arose after treatment had higher IFN (IFN-γ and IFN-α) genes but lower cell cycling gene expression at baseline, indicating activation and exhaustion. TCR sequences of pre-existing clones (in baseline samples) and new clones (in post-treatment samples) overlapped, with higher similarity compared to TCR sequences identified in healthy donors. These results are concordant with clonal TCR usage in SAA patients, suggesting that pre-existing and new clones may target shared (unknown) antigens.

Samples with higher TCR similarity in existing clones showed higher similarities in new clones and also between new and pre-existing clones (Fig. 6f).

Functionally, IFN-γ scores were decreased more dramatically in individuals with a decreased clone size after treatment compared to those with an increased clone size (Fig. 6g). Higher baseline IFN-γ, cell activation, TNF, and exhaustion scores correlated with better blood counts at 6 months, and decreased scores post-treatment positively correlated with more robust hematologic recovery (Supplementary Fig. 21e, f). Hematopoietic recovery appeared dependent on effective suppression of pre-existing T cell clones in size and cytotoxicity, and conversely emergence of new clones post-treatment was associated with less robust blood count improvement.

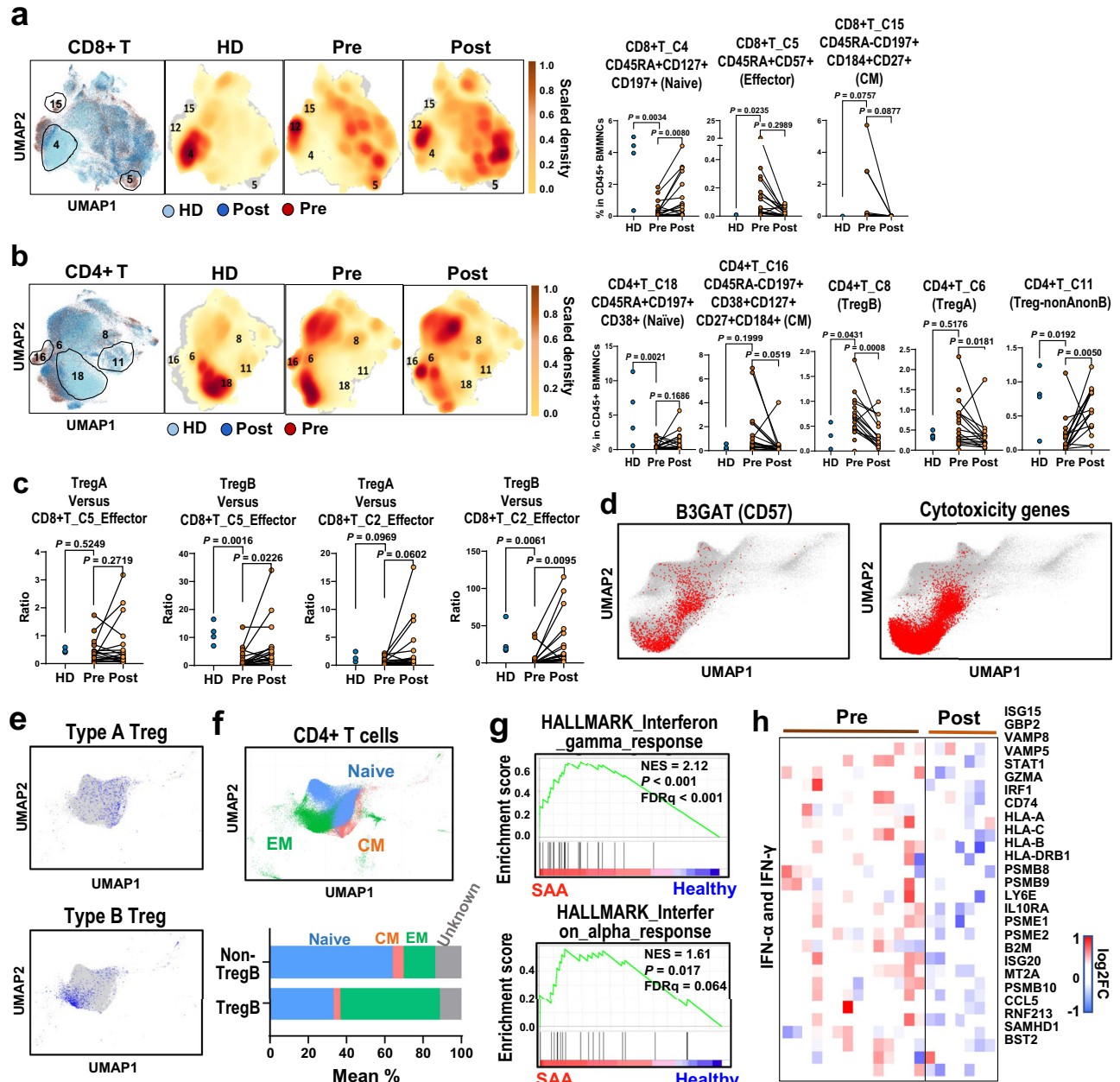

**Fig. 4 | CD8⁺ T cells and CD4⁺ T cell subpopulations expanded and activated in SAA. a** UMAP plots showing overlay of CD8⁺ T cell density in healthy controls (*n* = 4), SAA patients pre- (*n* = 20) and post-treatment (*n* = 18), as well as the individual conditions. Dot plots showing abundance (% in CD45⁺ BMMNCs) of CD45RA⁺CD8⁺CD127⁺CD197⁺ naïve T cells (cluster 4), CD45RA⁺CD8⁺CD57⁺ effector T cells (cluster 5), and CD45RA⁺CD197⁺CD184⁺CD127⁺ central memory T cells (CM, cluster 15) in pre- and post-treatment samples of SAA patients and healthy donors identified by Binary Clust. *P* values with the two-sided unpaired and paired Mann–Whitney test are shown. **b** UMAP plots showing overlay of CD4⁺ T cell density in healthy donors (*n* = 4), SAA patients pre- (*n* = 20) and post-treatment (*n* = 18) as well as individual conditions. Dot plots showing abundance (% in CD45⁺ BMMNCs) of CD45RA⁺ CD4⁺CD197⁺CD38⁺ naïve T cells (Cluster 18), CD45RA⁺CD4⁺CD197⁺CD38⁺CD127⁺CD27⁺CD184⁺ central memory T cells (CM, cluster 16), CD4⁺CD25⁺CD127⁻CD45RO⁺CD150⁺FOXP3⁺CD95⁺ CD194⁺CD39⁺ Treg-B (cluster 8), CD4⁺CD25⁺CD127⁻CD45RA⁺Foxp3⁺ CD197⁺ Treg-A (cluster 6), and nonA nonB-Treg (cluster 11) in pre- (*n* = 20), post-treatment (*n* = 18) samples of SAA patients and healthy donors (*n* = 4) identified by

BinaryClust. *P* values based on the two-sided unpaired and paired Mann–Whitney tests. **c** Ratios of Treg-A and Treg-B abundance versus CD8⁺ effector T cells (cluster 5 and cluster 2) in pre- (*n* = 20), post-treatment (*n* = 18) samples of SAA patients and healthy donors (*n* = 4). *P* values based on the two-sided unpaired and paired Mann–Whitney tests. **d** In a UMAP only showing CD8⁺ T cells, CD57⁺ T cells, and expression of cytotoxicity genes were indicated. **e** In a UMAP only showing CD4⁺ T cells, Treg-A and Treg-B were shown. **f** In a UMAP only showing CD4⁺ T cells, subtypes including naïve, central memory (CM), and effector memory (EM) were indicated. A bar chart showing percentages of Treg-B and non-Treg-B CD4⁺ T cells as naïve, CM, and EM. **g** Gene Set Enrichment Analysis (GESA) enriched plots of differentially expressed genes for Hallmark interferon gamma response and Hallmark interferon alpha response in Treg-B compared with non-Treg-B CD4⁺ T cells in pre-treatment samples of SAA patients. GSEA is based on the one-sided Kolmogorov–Smirnov test. **h** A heatmap showing expression of representative differentially expressed genes grouped by their functional pathways in IFN-γ and IFN-α signaling between Treg-B and non-Treg-B CD4⁺ T cells pre- and post-treatment in SAA patients. Values are presented as log2FC.

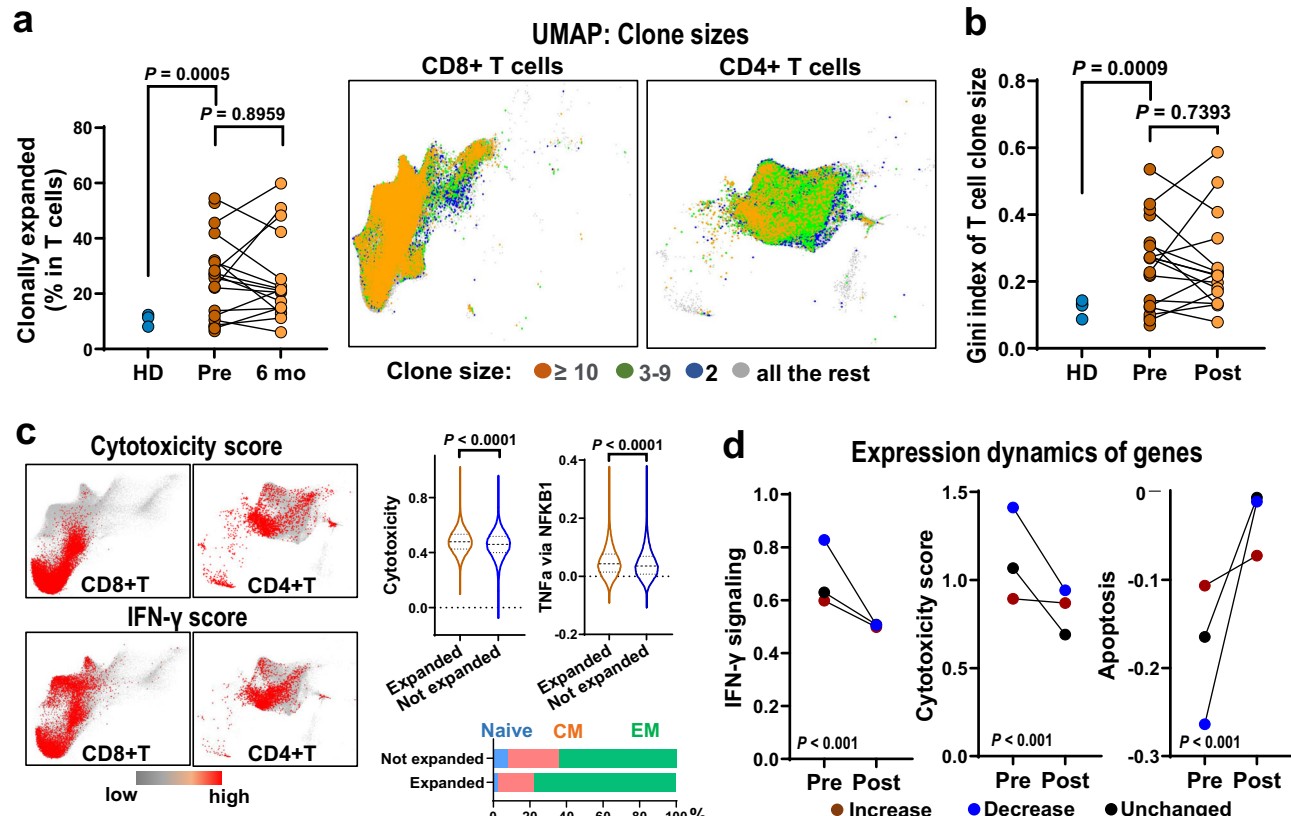

**Fig. 5 | Effector memory CD8+ T cells were clonally expanded in SAA.**
**a** Frequency of clonal expanded T cells in pre- ($n = 20$), post-treatment ($n = 16$) samples, and HD ($n = 4$). $P$ values with the two-sided unpaired and paired Mann–Whitney test. Clone sizes were projected to a UMAP of T cells in pre-treatment samples, colored based on clone sizes. **b** Gini index of TCR clone sizes in pre- ($n = 20$), post-treatment ($n = 16$) samples, and HD ($n = 4$). $P$ values with the two-sided unpaired and paired Mann–Whitney test. **c** CD4+ T and CD8+ T cells expressing the highest (top 10%) cytotoxicity score and IFN-γ signaling score are highlighted in red, and all the rest in gray. Expression of T cytotoxicity genes and TNF-α via NFκB signaling genes was compared between clonally and non-clonally expanded T cells. $P$ values with the two-sided unpaired and paired Mann–Whitney test. Bar chart showing percentage of clonally expanded and nonexpanded T cells as naïve, CM, and EM. **d** Expression dynamics of gene scores in averaging increased, decreased, and stable clones after treatment. $P$ values were generated using the two-way ANOVA with factors pre/post and increase/decrease/unchanged clones.

## Intercellular interactions in patient and healthy marrows

Intercellular relationships can be imputed from expression of ligand-receptor pairs across cell populations. Analysis using CellPhoneDB[46] revealed interactions among many cell types, but cross-talks were most pronounced among immune cell populations, in both patients and controls. Cell–cell interactions imputed using this computational approach were enhanced in SAA patients compared to controls (Fig. 7a), suggesting upregulation and cooperation of innate and acquired immune compartments in disease. Most enhanced cell–cell interactions at baseline were reduced after IST (Fig. 7b). We plotted overrepresented interactions of HSPCs with other cell types in SAA and compared with controls (Fig. 7c, Supplementary Fig. 22). IFNG secreted by CD8+ T cells was imputed to interact mainly with IFNGR on neutrophils and other cell types, including HSPCs, while IFNG secreted by other cell types interacted moderately with IFNGR on CD8+ T cells. Overall, there was an abnormally increased IFNG/IFNGR interaction between CD8+ T cells and other cells in SAA (Fig. 7d). CD8+ T and NK cells were the major sources of IFNG interacting with the IFNGR on HSPCs; there were increased IFNG/IFNGR interactions between HSPC and other cell types in SAA compared to healthy donors (Fig. 7e). These aberrant cellular interactions were reduced after treatment. Ligand-receptor pairs between HSPC and pre-existing clones, imputed computationally, were plotted in pre-treatment samples, and between HSPC and residual pre-existing as well as new clones in post-treatment samples. New T cell clones exhibited similar ligand-receptor interactions with HSPC as observed with pre-existing clones, suggesting functional similarity and similar antigen recognition (Fig. 7f). There were more ligand-receptor pair interactions between early HSCs and CD8+ T cells than between late HSCs with CD8+ T cells (Fig. 7g).

## Malignant clonal evolution at single-cell resolution

Development of secondary myeloid neoplasm is the most serious long-term complication in SAA patients treated with IST. In the current series, UPN10 suffered high-risk evolution to monosomy 7 at 6 months after treatment, and there was a large proportion of monosomy 7 (7-) cells in post-treatment CD34+ HSPCs detected by scRNA-seq (but not observed in total BMMNCs, Supplementary Fig. 23a, b, d–f). Gene expression was compared between 7- and diploid cells; genes involved in immune response pathways (including TNF-α signaling via NF-κB signaling, inflammatory response, and interferon gamma and alpha response) were downregulated in 7- cells, consistent with our original report[30]. There was a significant increase of cell cycling gene expression in 7- cells at disease evolution (Supplementary Fig. 23c). We did not identify aneuploid cells in pre-treatment sample from this patient (Supplementary Fig. 23g); other patients with chromosomal abnormalities were not included in the analysis due to timing of sampling, a clone size, or a small fraction of cells with aberrant chromosomes (Supplementary Data 1).

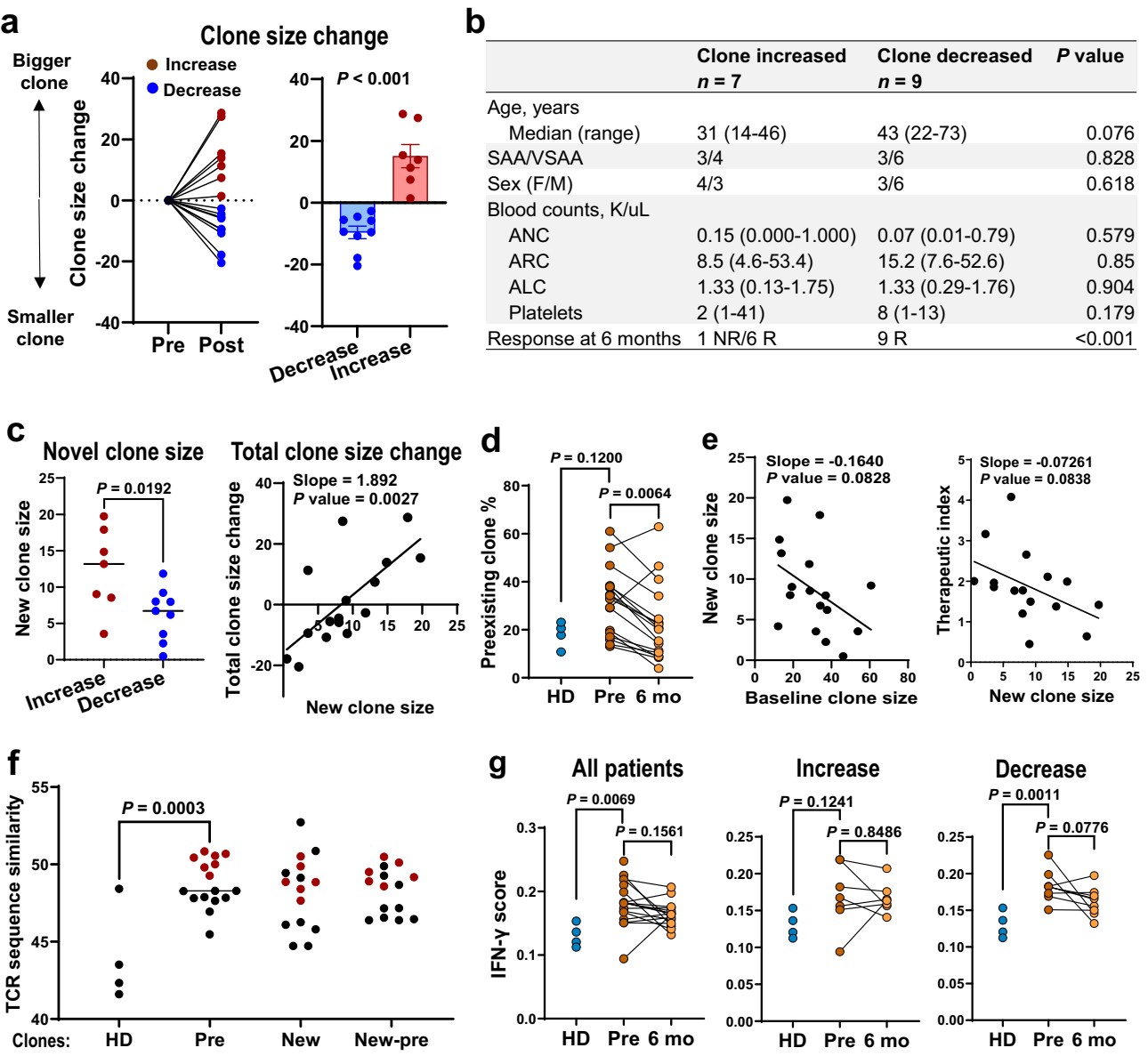

**Fig. 6 | T cell clonal expansion dynamics associated with hematopoietic recovery. a** Patients were grouped: clone sizes increased (red, $n = 7$) and decreased (blue, $n = 9$). Data are presented as mean values ± SEM. $P$ values with the two-sided unpaired Mann–Whitney test were shown. **b** Clinical characteristics of patients with clone sizes increased ($n = 7$) and decreased ($n = 9$) post-treatment. $P$ values with the two-sided unpaired Mann–Whitney test are shown. **c** New clone sizes in two groups of patients who had clone sizes increased ($n = 7$) and decreased ($n = 9$) were compared. $P$ values with the two-sided unpaired Mann–Whitney test are shown. Correlation of new clone sizes with total clone size changes after treatment. $P$ values and slopes with the Pearson correlation test are shown. **d** Clone sizes (frequency %) of pre-existing clones in paired samples of patients ($n = 16$) and HD ($n = 4$). $P$ values with the two-sided unpaired and paired Mann–Whitney test are shown.

**e** Correlation of pre-existing clone sizes with new clone sizes (left) and new clone sizes with therapeutic scores (right). $P$ values and slopes with the Pearson correlation test are shown. **f** TCR sequence similarity is plotted for T cells in HD, pre-existing clones in pre-treatment samples (pre), new clones in post-treatment samples (post), and between pre-existing and new clones in post-treatment samples (post). Those who had higher similarities of pre-existing clones are highlighted in red across sample groups. $P$ values with the two-sided unpaired Mann–Whitney test are shown. **g** Expression of IFN-γ signaling scores was plotted for all patients ($n = 20$), for patients who had clone sizes increased ($n = 7$) and decreased ($n = 9$), and HD ($n = 4$), respectively. $P$ values with the two-sided unpaired and paired Mann–Whitney test are shown.

## Constitutional risk factors in an immune-mediated disease: germline variants related to specific cell types

To evaluate for genome-wide heritable genetic factors in AA and disease-associated cell populations in BM, we integrated the current scRNA-seq data with polygenic signals from GWAS. A gene-level association analysis was performed using MAGMA[47]. GWAS-based pathway enrichment analysis revealed that top 100 genes associated with AA (Fig. 8a, b) were enriched in immune response pathways, including IFN-γ and IFN-α (Fig. 8c). We then examined the most

involved cell types by computing a disease score (scDRS)[48] for each single cell in our current scRNA-seq dataset, based on the disease-specific genes identified by MAGMA. Disease scores were similar across BMMNC cell types in controls, but highest for CD8+ T and NK cells in SAA patients (Fig. 8d, Fisher test, $P = 0.04$). For cell populations examined using RolyPoly[49], CD8+ T cells were the most relevant (Fig. 8e). We next examined if these top disease-associated genes exhibit cell-type-specific expression. *S1PR5* was preferentially expressed in NK and CD8+ T cells, *CCDN2* in CD4+ T, CD8+ T, and NK cells, and

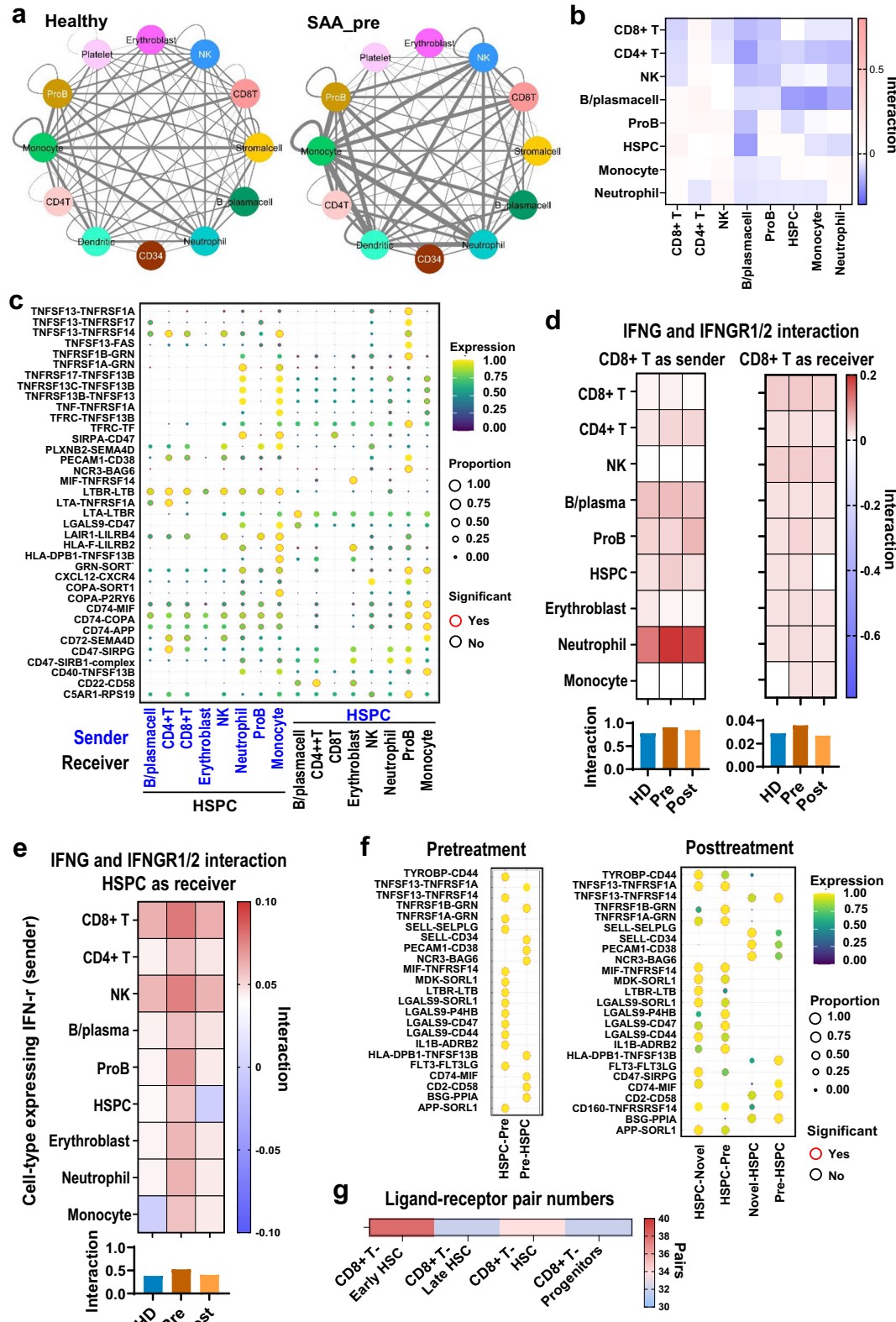

*DGKA* in CD4$^+$ T cells. *DERL3* had preferential expression in B and plasma cells; expression of *FBP1*, *IRF5*, *VAMP3*, and *NFKB1* was largely limited in myeloid cells (Fig. 8f, g). Further analysis was thus concentrated on a few subpopulations: S1PR5$^+$ NK and CD8$^+$ T cells, and IRF5$^+$ and NFKB1$^+$ myeloid cells.

The frequency of S1PR5$^+$ NK cells and S1PR5$^+$CD8$^+$ T cells was higher in pre-treatment samples than in healthy donors, and

significantly decreased after treatment (Fig. 8h). S1PR5$^+$ NK cells, S1PR5$^+$CD8$^+$ T cells, and NFKB1$^+$ myeloid cells had higher expression of many immune activation genes in the corresponding cell population, compared with negative cells (Fig. 8I, j). There were increased interactions of S1PR5$^+$ NK cells and S1PR5$^+$CD8$^+$ T cells with other cells, compared to interactions of negative cells, and also increased interactions of IRF5$^+$ myeloid cells and NFKB1$^+$ myeloid cells with other cells

**Fig. 7 | Enhanced cell–cell interactions in SAA and attenuation by therapy.**
**a** Ligand-receptor pairs among cell types in BMMNCs were estimated by
CellPhoneDB[89]. Color legends for cell types are the same as in Fig. 2a. The Thickness
of lines connecting cell types indicates the total number of ligand-receptor pairs
between two cell types estimated by CellPhoneDB. In general, there were more
ligand-receptor interactions between cell types in BMMNCs of SAA (left) than in
those of healthy donors (right). CD8T CD8⁺ T cell, CD4T CD4⁺ T cell, NK natural
killer cell, CD34 CD34⁺ cell, ProB proB cell, B_Plasma B cell_Plasma cell, Ery ery-
throblast, Neut neutrophil, Mono monocyte, DC dendritic cell. **b** Heatmaps show-
ing LogFC of cell–cell interaction scores estimated by CellPhoneDB across cell
types in BMMNCs post-treatment as compared to pre-treatment. **c** Ligand-receptor
pairs that were overrepresented in SAA patients than in healthy donors across cell
types in the BMMNCs are presented, with HSPCs as sender (left) and receiver
(right). Significance indicates if the ligand-receptor pair is overrepresented in pre-
treatment samples compared with controls. **d** Cell–cell interactions of IFNG/IFNGR.
Heatmaps depicting the interactions of IFNG and IFNGR1/2 between CD8⁺ T cells
(left, as sender and right, as receiver). A sum of IFNG/IFNGR interactions in CD8⁺
T cells with all other cell types in the BM is shown at the bottom for healthy donors,
pre-, and post-treatment samples. **e** A heatmap depicting interactions of IFNG and
IFNGR1/2 between various cell types in the BM (as sender) and HSPC (as receiver).
Sum of IFNG/IFNGR interactions in HSPCs with all other cell types in the BM is
shown at the bottom for healthy donors, pre-, and post-treatment samples.
**f** Heatmaps showing ligand-receptor pairs between HSPCs and pre-existing T cell
clones in pre-treatment samples (left), and between HSPCs and residual pre-
existing T cell clones, and HSPCs and new T cell clones (right). **g** Heatmaps showing
numbers of ligand-receptor pairs between CD8⁺ T cells and early-stage HSCs, late-
stage HSCs, HSCs, and progenitors.

compared to interactions with negative cells (Fig. 8k). These data,
uncontrived in silico, implicate genetic backgrounds as contributors to
disease phenotype: cell-type specific expression S1PR5⁺CD8⁺ T to
dysregulated immune activation, and IRF5/NFKB1⁺ myeloid cells to
exaggerated inflammation when interacting with lymphocytes.

## Discussion

Single-cell methodologies, especially those based on nucleotide
sequencing, are now appreciated as much more than a technical
improvement, but rather revolutionary in the depth, breadth, and
complexity of the results that they generate[50,51]. Single-cell methods
have marked advantages: unbiased as to data collection, minimal
in vitro perturbation, and easily shared (and mined) data files. Single-
cell techniques are particularly well suited to examining limited cell
numbers, as in the marrow failure syndromes, and to hematologic
diseases in general, for which blood and marrow cells are readily
obtained in the clinic[29–33,52,53]; even from patients with SAA. In the
current work, we prospectively incorporated single-cell methodolo-
gies into a clinical research protocol, in order to compare BM before
and after a defined therapeutic regimen that leads to improved blood
counts and sustained remissions in this disease. Single-cell methods
have been criticized for their reliance on "black box" algorithms and
analytical pipelines, and the issue of "validation," as initially required
for the measurement of RNA abundance or avoidance of DNA errors,
and later for the requirement of "functional" confirmation, despite the
absence of satisfactory statistical standards or privileging of less
accurate historical assays. We sought to avoid such problems by per-
forming simultaneous orthogonal testing for internal validation of
results, as well as comparison with similar, if more restricted, pub-
lished datasets[43].

Immune AA is broadly understood as the destruction of hema-
topoietic stem cells by T cells. By conventional histology, the appear-
ance of the affected organ, the marrow, is dramatic but bland, with
replacement of a hematopoietic tissue with fat. Histochemistry is
unrevealing, with variable residual "stromal" elements and lympho-
cytes. Methods reliant on cell separation are also fraught due to the
paucity of targets (HSPCs absent or severely decreased) and effectors
(CD4 and CD8+ lymphocyte populations within mononuclear cell
sedimented fractions). In striking contrast, single-cell methods using
freshly isolated and minimally manipulated total BM cells allowed
comprehensive visualization of heterogeneous populations–confirm-
ing suspected features of stem cell loss and immune pathophysiology,
revealing unexpected complexity of interactions among cell popula-
tions, and allowing inferences concerning specific cell function. Of
importance, comparison of marrow on presentation of disease and
after effective therapy indicated surprising features of incomplete
stem cell repopulation, dynamic effector T cell oligoclonality, and a
complex network of hematopoietic and immune cell relationships.

Some of our results, satisfyingly, confirm and expand suspected
and accepted components of the immune pathophysiology of AA.

HSPCs as targets have been difficult to study quantitatively because
of their scarcity and poor correlation with blood counts. We unex-
pectedly observed that clinical hematologic recovery was not
accompanied by an increase in stem cells in the BM, but among the
heterogeneous population of early multipotential and lineage-
committed progenitor cells. In retrospect, this result is less surpris-
ing and likely explanatory: AA is a relapsing syndrome, especially
early after immunosuppression[8,54]; eltrombopag appears to pre-
ferentially expand progenitors[37]; and attempts to measure and har-
vest CD34 cells post-hematologic recovery have failed. The status of
stem and progenitor cells at more distant time periods after treat-
ment and after years of stable near-normal blood counts, as well as
the characteristics of the hematopoietic compartment in the setting
of relapse, should be examined in future studies. Malignant clonal
evolution and the development of myeloid neoplasia in immune AA
were not the focus of the current work and remains difficult to study
systematically. Nevertheless, we detected monosomy 7 cells,
acquired at 6 months in a single patient, to exhibit upregulation of
cell cycling genes, not seen in similar patients who received immu-
nosuppression alone[30], possibly indicating stimulation of aneuploid
cells by eltrombopag, a growth factor that acts on stem cells. Phar-
macologic doses of a thrombopoietin mimetic may elicit prolifera-
tion of deficient monosomy 7 cells that are unresponsive to
physiologic thrombopoietin concentrations.

For the immune system, single-cell methods provide a global
view of manifold components and a striking visualization of the
effector cells and associated regulatory elements. Cytotoxic CD8⁺
T cells have long been suspected to be the main proximal effectors of
tissue destruction[26,55]: this population is now dramatically evident at
low resolution, in all patients at presentation, as dominant, and the
apparent target of treatment. Single-cell methods disclose con-
siderable heterogeneity of tissue transcriptomic programs and cell
surface phenotypes within the CD8⁺ T cell compartment; some
subsets are strikingly abnormal in number and inferred function, and
correlate with clinical status. Subsets of effector memory cells can be
inferred to interact with hematopoietic target populations, and their
transcriptomes show upregulation of IFN and other cytokine sig-
naling pathways. Features of oligoclonality, previously reported at
comparatively low resolution[43,56], are also revealed in detail: TCR
oligoclones were limited in number but common to all patients
examined; they showed private, not disease-specific specificities,
although there were shared TCRs among patients compared to
healthy controls. The inciting antigenic events in immune marrow
failure syndromes are not known; they have been postulated to
derive from antigenic mimicry with epitopes of frequently encoun-
tered pathogens[57,58]. Cross-reactivity of TCRs from AA cases with
herpesvirus antigens has been recently reported[59]. Although not the
main focus of the present work, addressing viral reactivity from
single RNA sequences is limited by patient numbers (a large enough
cohort with a heterogenous HLA background is required) and the

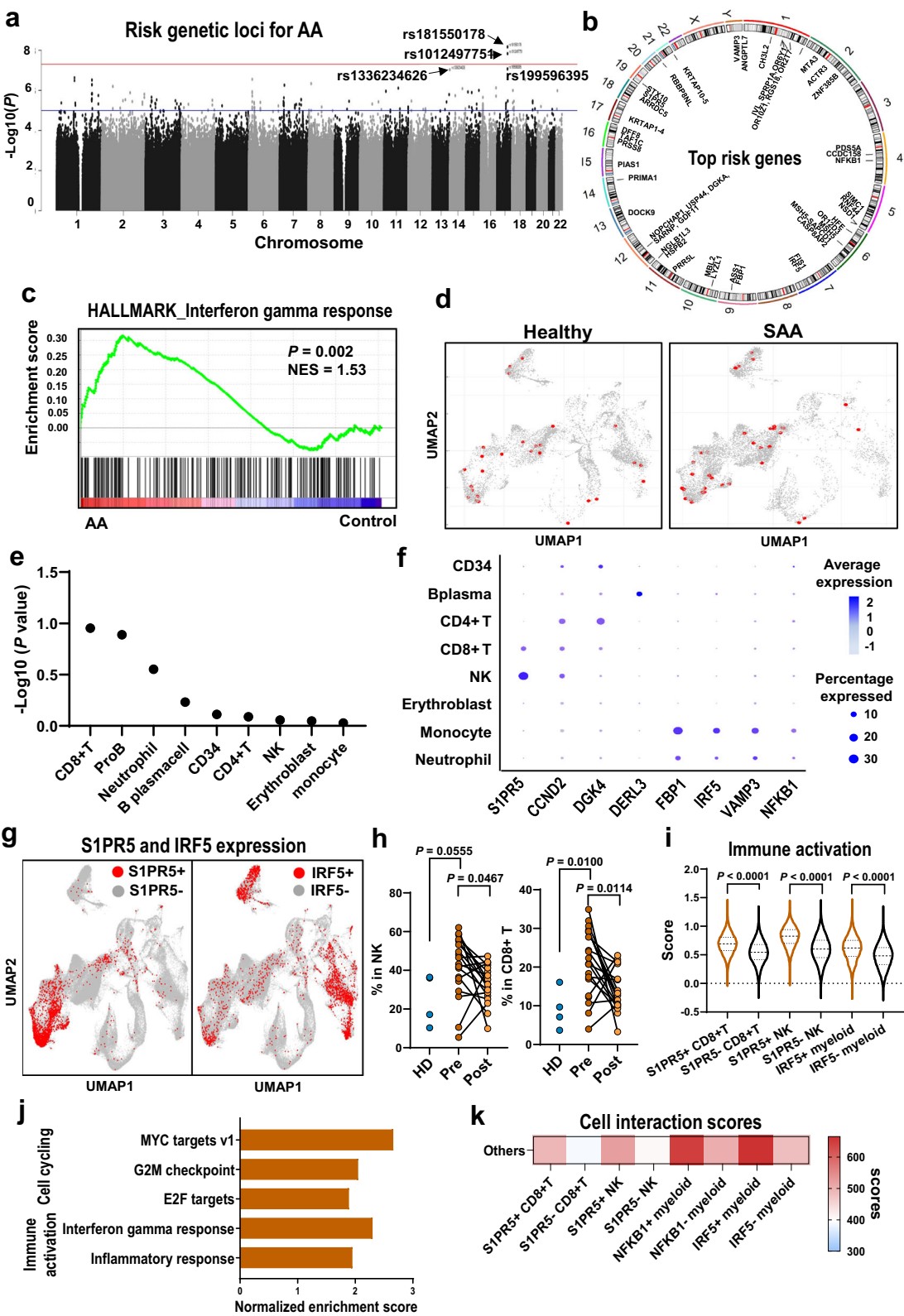

existing virus databases (limited in sequence coverage and biased). CD8+ effector T cell oligoclonality appeared dynamic over time, with early clones decreasing or disappearing with treatment, while new clones also targeting stem and progenitors arose after cytotoxic lymphocyte reduction, and during and despite suppressive immune therapy. Selection for TCR and clones may be based on resistance to such treatments as well as through "antigen spread,"

uncovering of novel epitopes expressed on damaged and regenerating marrow cells.

The behavior of specific CD8+ cell subsets correlated with clinical responsiveness, especially the robustness of hematologic recovery. Our results support previous observations in a pediatric very SAA (VSAA) cohort ($n = 10$) that the highly skewed TCR repertoire in the BM is associated with response to immunosuppression[60], which we

**Fig. 8 | Genetic risk and cell-type-specific disease-related variants contribute to disease phenotype. a** A Manhattan plot of GWAS analysis highlighting risk genetic loci for AA, based on GCST9001879 data, in which *P* values were calculated by the generalized linear mixed model with adjusting for age, sex, and top 20 principal components. **b** A Circus plot showing top risk genes of AA of MAGMA analysis. The outer ring demonstrates 22 autosomal chromosomes. In the inner ring, a circular symbol represents a specific risk gene. **c** GESA enriched plots of risk genes identified in GWAS analysis were enriched in Hallmark interferon gamma response. GSEA based on the one-sided Kolmogorov−Smirnov test. **d** UMAP embedding BMMNCs from healthy donors (left) and SAA patients (right), colored by scDRS disease scores calculated from GWAS summary statistics. **e** A dot plot showing results of combination of scRNA-seq data and GWAS summary statistics on AA based on RolyPoly analysis among patients. *Y*-axis shows mean negative log-transformation *P* value (-Log2P), *x*-axis shows cell types. *P* values (one-sided) were calculated by

estimating the association of each cell type with a phenotype. **f** A heatmap showing percentages of cells expressing selected risk genes in multiple cell types. **g** Same UMAP projection of BMMNCs as in Fig. 2a. Expression of S1PR5 and IRF5 is plotted, showing cells (in red) with top 10% expression levels of S1PR5 and IRF5. **h** Frequency of S1PR5$^+$CD8$^+$ T cells and S1PR5$^+$NK cells were compared between pre-treatment ($n = 20$) and post-treatment samples ($n = 16$) of SAA patients and healthy donors ($n = 4$). *P* values with the two-sided unpaired and paired Mann−Whitney test are shown. **i** Immune activation scores in S1PR5$^+$ and S1PR5$^-$ CD8$^+$ T cells, S1PR5$^+$ and S1PR5$^-$ NK cells, and IRF5$^+$ and IRF5$^-$ myeloid cells. *P* values with the two-sided unpaired and paired Mann−Whitney test are shown. **j** Top downregulated gene pathways in NFκB$^+$ compared with NFκB$^-$ myeloid cells. **k** Cell interaction scores calculated by CellPhoneDB in S1PR5$^+$ and S1PR5$^-$CD8$^+$ T cells, NK cells, IRF5$^+$ and IRF5$^-$ myeloid cells, and NFκB$^+$ and NFκB$^-$ myeloid cells.

investigated further in a larger adult cohort and with clonal dynamics before and after treatment. An initial larger total T cell clone size, lower TCR diversity, and higher clonality pre-treatment associated with a decreased clone size after treatment; conversely, patients with a smaller total T cell clone size, higher TCR diversity, and lower Gini index showed an increased clone size after treatment. Clonotype dynamics were associated with clinical course in a few cases who were assessed longitudinally[43,56]; not being explorative, these studies only identified and tracked one or a few top pre-existing clones, without examination of global clonal dynamics patterns or new clones. Indeed, pre-existing clones almost uniformly decreased with immunosuppression, and new clones almost exclusively contributed to clonal expansion after treatment. Clonal dynamics inferred from serial samples and single-cell TCR usage suggest that IST is effective in suppressing pre-existing clones, but newly arising clones may limit blood count recovery. These results allow re-interpretation of clinical observations, such as the paradoxical action of more intensive immune regimens, which are generally inferior to milder conventional therapy in treatment-naïve cases yet can salvage patients refractory to initial therapy, often leading to sustained hematologic recovery[61–64]. Monitoring T cell clonal dynamics could guide intensification strategies for patients showing inadequate recovery.

At single-cell resolution, there was heterogeneity across and within other immune cell types. While Treg numbers were not reduced compared to matched healthy donors and were not changed after therapy, frequency of Treg-B (specified by higher IL-2 receptor, FAS and CD45RO expression) was higher in pre-treatment samples was surprising[23], but most likely due to BM rather than blood as the source of samples; regulatory T cells may relocate to the site of inflammation. Additionally, as our cohort included many patients with VSAA, in which Tregs would be exposed to higher levels of inflammation, the most functional and proliferative Tregs expressing CD95 (FAS) would be sensitive to FASL-induced apoptosis and adopt a "protective" phenotype by downregulating FAS and their functional suppressive capabilities[41,42].

In general, our analysis of immune cells in AA BM is consistent with previously published work using scRNA-seq, describing dominant CD8$^+$ cytotoxic T cells, oligoclones based on TCR, and pathophysiologic effector memory cells[50,51,65–67]. However, we could not identify T cells with clear viral targets, as mentioned, and in addition, our results and a comparable patient dataset differ in the identification of an NKT cell population distinct to AA. We would note that NK cells have generally been reported to be decreased in number and activity in AA. Only enriched NK cells were identified by CyTOF due to the limitation of the antibody panel, and their number was not significant in the current series, but a few CD57$^+$ cytotoxic NK clusters were over-represented in disease and were suppressed after treatment. Ongoing collaborative work seeks to further resolve this result.

That our previous view of the pathophysiology of AA has been too simple is supported by at least two new analyses in our study. First,

utilizing a computational method for imputing cell−cell interactions from ligand-receptor pairs, we visualized a highly complex network of relationships that cross traditional boundaries of lineage, differentiation, and compartment. These suggestive individual relationships invite experimental verification but may already inform translational reports and animal models. Some undoubtedly represent the complexity of a pathological immune response. Yet even in healthy donors, interactions were inferred between mature and immature cells, between myeloid and non-myeloid cell types, and among immune system cells—many intensified in diseased marrow, and potential targets of novel therapeutic interventions. Interactions between CD8$^+$ T cells and HSPCs are among the strongest: IFNGR on HSPC interactions with IFNG from CD8$^+$ T and NK cells are the strongest. Second, when we linked our RNA sequencing results to existing GWAS datasets, we discovered multiple immune-related genes that potentially suggest a germline background to an acquired immune disease. Polymorphisms may indicate constitutional genes as genetic susceptibility factors and disease modifiers, based not simply on association but on plausible causal relationships. Such relations may include vulnerability to unregulated responses to antigen stimulation, activation of myeloid cell targets in response to T cell attack, and exaggerated specific immune pathway responses, especially for IFN-γ signaling and production. As one example, increased cell−cell interactions between S1PR5/IRF5$^+$ cytotoxic cells and other T cell and major marrow cell types accompany enhanced crosstalk, in a genetic background immune activation, particularly of the IFN-γ pathway.

Among the obvious limitations of a sample size, clinical follow-up, drop-out RNA sequences, and antibody selection for cytometry, we note more specific difficulties. First, there were too few non-responders after highly effective therapy to allow an analysis of categorized response. Second, computational imputation of cell−cell interactions through ligand and receptor gene expression has been widely used, and algorithms now incorporated curated interacting proteins from well-established databases and consider multi-subunit structures for accuracy, but datasets are by far from complete and accurate; testing by individual functional assays was not feasible, nor is there a statistic to validate the validation procedure itself. Nevertheless and important observations from scRNA-seq were consistent with CyTOF and conventional flow cytometry, with historical cell co-culture observations decades ago[19,20], and with other published single-cell data in human marrow failure syndromes[50,51,64–67]. Third, inference from GWAS data was made with publicly available datasets, and linking germline background with transcriptome data was indirect. Although costly, simultaneous GWAS and single-cell multi-omics (genomics, transcriptomics, and phenotyping) in individual patients could provide more comprehensive and direct information as to the cell- and cell population-specific roles of genetic background in modulating disease[68,69]. Fourth, although in theory there is a large pool of TCR sequences in the population, profiling of a "complete" T cell clonotype in the current and many other studies is inevitably limited by a patient

cohort and individual heterogeneity, availability of samples, disease and pathological or biological conditions, cost, and sequencing techniques. Last, direct and more detailed comparison of our results with published work using CyTOF and/or scRNA-seq was not undertaken, due to heterogeneity of patient cohorts (disease severity and treatment regimens) and specimen types (peripheral blood, BM, and isolated cell populations), and variations in algorithms for cell identification from study to study. Integration and comprehensive analysis of currently available datasets of AA patients and, more broadly, datasets in other autoimmune diseases are desirable in future mining expeditions. "Incompleteness" is not a fault of single-cell methodologies but a powerful feature of these novel, developing approaches.

The vast amounts of data from single-cell experiments and the complexity of the analytic programs limit any single descriptive report, but one publication from a single laboratory is only the beginning of global analyses and extension of reported results. Multiple sclerosis, uveitis, diabetes mellitus, and inflammatory bowel disease share pathophysiologic features with immune AA and respond to immunosuppressive therapies. Our approach offers a guide to laboratory investigations of these and other diseases of altered immunity and tissue destruction.

## Methods

### Patient cohort

We conducted a prospective phase 2 study under the protocol (www.clinicaltrials.gov NCT01623167) approved by the Institutional Review Boards of the National Heart, Lung, and Blood Institute, in accordance with the Declaration of Helsinki. A total of 139 persons were enrolled: their SAA had not been definitively treated with ATG-based IST, and they lacked a suitable matched sibling marrow donor or were not candidates for hematopoietic stem cell transplantation, due to patients' choices, advanced ages, or socioeconomic factors. Patients, 2 years of age or older, were enrolled from December 2014 to May 2022 and treated with hATG, CSA, and eltrombopag combination (hATG from day 1 for 4 days, therapeutic dosing of CSA from day 1 for 6 months, and eltrombopag starting from day 1 to 6 months). All patients met clinical criteria for SAA. For persons who met the criteria for very SAA (VSAA; absolute neutrophil count (ANC) $\leq 200 \times 10^9$/l), treatment was initiated based on morphologic confirmation while cytogenetic studies were pending. Age- and sex-matched healthy donors were enrolled as controls under protocol NCT00442195 in NHLBI: UPN111: 62-year-old female; UPN112, 27-year-old female; UPN113, 28-year-old female; UPN114, 23-year-old male.

### BM processing

BM specimens were obtained from patients and healthy donors after written informed consent under the corresponding protocols, and processed within 6 h after collection. BMMNCs were isolated from each person by density centrifugation using LSM Lymphocyte Separation Medium (Cat# 50494X, MP Biomedicals). Briefly, BM was diluted twofold using phosphate-buffered saline (PBS) (Cat# 17-516Q, Lonza), layered on top of 1 volume LSM Lymphocyte Separation Medium in a 50-ml Falcon tube, and spun down at $1140 \times g$ for 25 min at room temperature with the brake off. A BMMNC layer was isolated and washed with PBS after red blood cell lysing with ACK lysing buffer (Cat# 118-156-101, Quality Biological). BMMNCs were resuspended in the IMDM (Cat# 12440053, Thermo Fisher Scientific) + 2% fetal bovine serum (Cat# 12306C, Sigma-Aldrich) before fluorescence-activated cell sorting (FACS) to enrich lineage⁻CD34⁺ HSPCs. BMMNCs were stained in 50 µl volume with monoclonal antibodies for 30 min on ice: 5 µl anti-human lineage cocktail (CD3, CD14, CD16, CD19, CD20 and CD56; clones UCHT1, HCD14, 3G8, HIB19, 2H7, and HCD56, respectively, Cat# 348805, Biolegend) in Pacific Blue; 5 µl anti-CD34 Ab (clone 581, Cat# 555822, BD Biosciences) in PE and 5 µl anti-CD38 Ab (clone HIT2, Cat#

555462, BD Biosciences) in APC. Cells were sorted using the FACSAria Fusion Flow Cytometer (BD Biosciences). Aliquots of BMMNCs were subjected to multi-color flow cytometry to profile HSP subpopulations. BMMNCs and purified lineage⁻CD34⁺ cells were subjected to scRNA-seq.

### Flow cytometry profiling of HSPCs

BMMNCs were stained in a 50 µl volume with antibody mixtures on ice for 30 min in RPMI 1640 (Cat# 11875093, Life Technologies). Samples were subsequently acquired using the BD LSR Fortessa cytometer (BD Biosciences), followed by post-acquisition analysis using Flowjo software (v.7.6.4; Flowjo LLC, BD Biosciences). Antibodies used for flow cytometry analyses were: 5 µl anti-human lineage cocktail (CD3, CD14, CD16, CD19, CD20, and CD56; clones UCHT1, HCD14, 3G8, HIB19, 2H7, and HCD56, respectively, Cat# 348805, Biolegend) in Pacific Blue; 5 µl anti-human CD34 in PE (clone 581, Cat# 550761, BD Biosciences), 5 µl anti-human CD38 in APC (clone HIT2, Cat# 555462, BD Biosciences), 5 µl anti-CD90 in FITC (clone 5E10, Cat# 328108, Biolegend), 5 µl anti-human CD10 in BV605 (clone HI10A, Cat# 562978, BD Biosciences), 5 µl anti-human CD135 in PE/Cy7 (clone BV10A4H2, Cat# 313314, Biolegend), and 5 µl anti-human CD45RA in BV510 (clone HI100, Cat# 304142, Biolegend). Cell populations were defined as reported[36]: HSC, Lineage⁻CD34⁺CD38⁻; CMP/MEP, Lineage⁻CD34⁺CD38⁺CD10⁻CD45RA⁻; GMP, Lineage⁻CD34⁺CD38⁺CD10⁻CD45RA⁺; LymP, Lineage⁻CD34⁺ CD38⁺CD10⁺.

### CyTOF

A panel of antibodies based on surface markers and transcription factors (Supplementary Data 2) was designed to identify different cell populations for CyTOF analysis. Most meta-isotope-labeled antibodies were purchased from Fluidim. Some unlabeled antibodies were purchased from BioLegend and eBioscience, and were tagged with rare metal isotopes in King's College London, UK. BMMNCs ($10 \times 10^6$)/patient were stained with surface antibody mixtures on ice for 30 min in staining buffer. Subsequently, intracellular staining for transcription factors was performed after fixation and permeabilization according to the manufacturer's instructions (eBioscience). The CyTOF-2 mass cytometer (Fluidigm) was used for data acquisition. Acquired data were normalized based on normalization beads (Ce 140, Eu151, Eu153, Ho165, and Lu175)[70]. Automated clustering was performed on a subset of 800,000 cells sampled from all individuals.

**Data processing, scale transformation, automated clustering, and distance computations.** Data were initially processed and analyzed using Cytobank[71]. CD45⁺ cells were gated to eliminate debris, doublets, and dead cells. The stand-alone analysis tool cyt was also used for performing t-SNE dimensionality reduction and merging distinct FCS files[72]. We analyzed mass cytometry complex data using viSNE[72] to visually identify delineated subpopulations[73] in combination with SPADE[74] and heat maps to distinguish different types of cells.

**Identifying cell populations with unsupervised machine learning.** We employed a data analysis pipeline inspired by the T-REX algorithm[75] to uncover cell clusters associated with treatment response in BM samples. This involved analyzing data from 33 patients before and after treatment, and 4 healthy controls. We used a custom R workflow integrating dimensionality reduction (UMAP), nearest neighbor search (KNN), and marker enrichment modeling (MEM). The analysis scripts are publicly available online (https://github.com/cytolab/T-REX). UMAP was used to compress high-dimensional data while preserving relevant relationships. Default parameters from the uwot package were employed. KNN identified nearest neighbors for each cell using the low-dimensional data. Optimization based on tetramer enrichment determined the optimal k value (number of neighbors) to be 100.

Based on the KNN graph, a change in cell abundance between two samples was calculated.

Regions were analyzed using DBSCAN for clustering and MEM for detailed phenotype characterization. Regions demonstrating significant changes (<5% and >95%) were highlighted as orange and blue, respectively. MEM was applied to assess feature enrichment within KNN regions surrounding each cell. Instead of requiring a reference control, a statistical null reference was employed, representing a median IQR of all analyzed features. Enrichment values were mapped from 0 to +10.

**Differential abundance analysis.** We used Milo (v1.2.0) to test for the differential abundance of cells within the inferred neighborhoods, between two conditions (that is, SAA patients versus healthy controls or post-treatment versus pre-treatment samples). We first used the buildGraph function to construct a KNN graph with $k = 100$. Next, we used the make_Nhoods function to assign cells to neighborhoods based on their connectivity over the KNN graph. The neighborhoods were projected to UMAP for visualization. To test for differential abundance, Milo fit a Non-Binomial GLM to the counts of each neighborhood, accounting for different numbers of cells across samples using the Trimmed Mean of M values normalization. The spatial FDR and log2foldchange of the number of cells between two conditions in each neighborhood were used for visualization. The beewarm plot of the distribution of log2foldchange across neighborhoods was used to present the differential abundance of cell populations.

**BinaryClust.** CyTOF analysis was further performed, separately, in a semi-supervised way using an in-house pipeline, BinaryClust (https://github.com/desmchoy/BinaryClust)[40]. Briefly, FCS files were imported using the R package flowCore. Data were arcsinh transformed with a cofactor of 5, and key channels of CyTOF data had an expected behavior of being log normal with zero inflation (confirmed with the function examine_data). By applying 2-means clustering for each of the key channels, it was possible to binary classify each marker into positive and negative populations to generate a "classification matrix" for all the cell subpopulations. BinaryClust was applied to each sample separately in a parallel manner. Cell abundances and median expression for each marker were then tallied for each sample. Differential expression and abundances between conditions were then computed in a pairwise manner. Statistical significances were determined by the Mann–Whitney $U$ tests with the Benjamini–Hochberg corrections. Summary heatmaps of differential abundances and expression results were plotted using the R package ComplexHeatmap.

**Density plot.** Processed CyTOF data from BinaryClust has been exported to Python and used for further analyses with the "scanpy" package (version 1.9.6)[76]. The neighborhood graph for the UMAP has been calculated using the "neighbors" function with the parameters of 30 neighbors and a cosine distance metric. The density has been calculated with the "embedding_density" function.

**Definition of a predicative index and a therapeutic index**
Serial sampling of acutely ill patients is difficult, but rigorously processed tissues, collected before therapy and after response, allows for strong inferences as to pathophysiology and drug mechanisms of action. As there was only a single non-responding patient, direct comparisons between R and NR as categorical variables could not be performed. Instead, we generated predictive and therapeutic indices by conversion of blood cell counts to scores as continuous variables. This method yielded correlations of baseline parameters and hematologic response, a predictive index, and dynamic changes in parameters to correlate with hematopoietic recovery at 6 months, a therapeutic index (Supplementary Figs. 11 and 12)[77].

To circumvent the limitation of sample sizes and the fact that only one non-responder at a primary end point, we designed two indices: $Pi$ and a $Ti$ to leverage the single-cell and clinical data in a continuous axis, and thus could detect consistent patterns among immune cell compositions and their connections with blood counts after treatment in all patients. We calculated a blood count score for each sample, combining white blood cell counts, hemoglobin level, and platelet counts to represent trilineage hematopoietic recovery. Hematopoietic score changes were calculated by changes of blood count scores between post- and pre-treatment samples. The $Pi$ was to calculate the correlation of baseline parameters with blood count scores at 6 months after treatment; the Ti was to calculate the correlation of changes of parameters (cellular proportion, dynamic gene expression, TCR clonality, and others) with changes of blood count scores between post- and pre-treatment samples[78].

We converted ANC, platelet count, and absolute reticulocyte count (ARC) at baseline and post-treatment to blood count scores on the same scale based on blood count distributions, by calculating their distance[times of standard deviation] to the median of those from two large Caucasia cohorts[78,79]. First, as expected, ANC, platelet, and ARC scores were positively correlated with ANC, platelet count, and ARC; we defined an average of these three counts as a "hematopoietic score", and a change of this average score post-treatment as a "hematopoietic recovery score." We checked validity of this approach by correlating ANC, platelet count, ARC, and the average hematopoietic scores with disease severity and response to treatment, and found these scores representing residual hematopoiesis and recovery, were largely consistent with responses (complete respnse (CR), partial response (PR), and NR; and CR/strong PR and NR/weak PR) (Supplementary Figs. 11 and 12). In both pre- and post-treatment samples, we identified a myeloid cell abundance largely positively correlated with the hematopoietic score while a lymphoid cell abundance largely negatively correlated with the hematopoietic score (Supplementary Fig. 13a). Baseline myeloid cell abundance predicted a better response at 6 months and baseline lymphoid cell abundance negatively strongly correlated with hematopoietic recovery (Supplementary Fig. 13b). As expected, increased myeloid and B cell abundance and decreased CD8$^+$ T and CD4$^+$ T cell abundance following therapy correlated with recovery (Supplementary Fig. 13c).

**scRNA-seq library and scTCR/BCR-seq library preparation and sequencing**
scRNA-seq coupled with scTCR/BCR-seq analysis was performed with the 10x Genomics System using the 10x Genomics Single Cell Immune Profiling Solution v 1.1 (the Chromium Single Cell 5′ Reagent Kit v1 and v1.1, Cat# 1000165, 10x Genomics), following the manufacturer's protocols (www.10xgenomics.com)[80]. Briefly, BMMNs and FACS-sorted BM lineage$^-$CD34$^+$ cells were washed with 1X PBS with 0.04% (wt/vol) bovine serum albumin. Cell concentration and viability were determined using the Countess II automatic cell counter (Thermo Fisher Scientific) and the trypan blue staining method. Cell loading and capturing were done on the Chromium Controller. Following reverse transcription and cell barcoding in droplets, emulsions were broken, and cDNA was purified using Dynabeads MyOne SILANE, followed by PCR amplification. Amplified cDNA was then used for 5′ gene expression library construction and TCR/BCR enrichment. For gene expression library construction, the amplified cDNA was fragmented, end-repaired, and double-sided size-selected with SPRIselect beads. For TCR/BCR library construction, TCR/BCR transcripts were enriched from amplified cDNA by PCR, and were sequenced on the Illumina NovaSeq system using read lengths of 26-bp read 1, 8 bp i7 index, 98-bp read 2. The scTCR/BCR libraries were sequenced using read lengths of 150-bp read 1, 8 bp i7 index, 150-bp read 2.

### scRNA-seq data preprocessing and quality control

The sequencing data was analyzed with the cellranger pipeline (https://support.10xgenomics.com/single-cell-gene-expression/software/pipelines/latest/what-is-cell-ranger, v7.0.1) to process scRNA-seq raw data in order to align reads to the genome, and to generate gene–cell expression matrices. Specifically, sequencing reads were aligned to the hg38 reference genome by STAR with annotation of ENSEMBL. Uniquely aligned reads were used to quantify gene expression levels for all genes with Unique Molecular Identifiers (UMIs). We filtered and removed low-quality cells from further analyses if the number of genes detected was fewer than 300 (low quality, potential fragments) or more than 6,000 (potential doublets). We also excluded those cells with a high percentage of mitochondrial gene reads (>10%)[81], and the remaining single cells were subjected to subsequent data analysis.

TCR reads were aligned to the GRCh38 reference genome, and consensus TCR annotation was performed using the cellranger vdj program (10x Genomics, version 7.0.1). TCR libraries were sequenced to a final 1662 mean read pairs/cell. TCR annotation was performed using the 10x cellranger vdj pipeline as described at https://support.10xgenomics.com/single-cell-vdj/software/pipelines/latest/using/vdj. Barcodes with a higher number of UMI counts than a threshold derived from the simulated backgrounds were considered as cell-associated barcodes. V(D)J read filtering and assembly were implemented as in a previous study[82]. In summary, cellranger firstly trimmed known adapters and primer sequences from the 5′ and 3′ ends of reads and then filtered away reads lacking at least one 15-bp exact match against at least one reference segment (*TCR*, *TRA*, and *TRB* gene annotations in ENSEMBL version 87). Next, cellranger performed de novo assembly for each barcode by building a De Bruijn graph of reads independently. The assembler output contig sequences which were assigned at least one UMI. Finally, each assembled contig was aligned against all of the germline segment reference sequences of the V, D, J, C, and 5′ UTR regions. Only the productive contigs were kept. Most cell barcodes contained two matching productive contigs, comprising either a TCRA or a TCRB, though it was biological possibility that fewer productive contigs (low sensitivity) or >2 productive contigs (some cells do contain more than one TCRB or TCRA chain) were associated with one cell barcode[83]. Similarly, BCR reads were also processed using the cellranger vdj program with the IMGT database of the GRCh38 genome as reference. Only productive contigs of BCR were kept for analysis. Sequencing metrics are shown in Supplementary Data 4.

Downstream analyses were mainly performed using the R software[84] package Seurat (http://satijalab.org/seurat/, v4.1.2) on BMMNCs and lineage⁻CD34⁺ cells separately[85]. Raw reads in each cell were first scaled by a library size to 10000 and then log-transformed. To improve downstream dimensionality reduction and clustering, we performed a data integration with the robust principal component analysis (RPCA) algorithm, instead of canonical correlation analysis in Seurat, due to memory issues brought by data of millions of cells. RPCA is less memory-intensive at the cost of being more conservative with integration. The 2000 highly variable genes were used for PCA of high-dimensional data. Top 30 principal components were selected for unsupervised clustering of cells with a graph-based clustering approach, and further dimensional reduction with UMAP.

### Quantification and statistical analysis

After alignment with RPCA, cells from different participants were well mixed and separated by cell-type categories. Clusters were identified by the FindClusters function in Seurat[86] and shown in UMAP plots. Accordingly, marker genes in each cluster were identified using the Wilcoxon Rank Sum test implemented in the Seurat v.4.1.2 package.

**Cell-type assignment.** For lineage⁻CD34⁺ cells, an HSPC type was assigned to each cluster based on significance in overlapping between HSPCs and cluster-specific genes (Fisher's exact test)[30,87]. Top 250 overexpressed genes in each HSPC population were downloaded from http://www.jdstemcellresearch.ca/node/32 and were denominated as cell-type-specific signature genes. Subsequently, the one-tailed Fisher's exact test was utilized to assert enrichment of HSPC signature genes in the cluster marker gene list for each cluster[88], and a top-associated cell type was assigned to each cluster. Cell types of BM cells were assigned with the same strategies using the Human Cell Atlas as reference[89]. The cell-type annotations were refined with known marker genes.

**Differential abundance analysis.** Similar to the data analysis of CyTOF, differences in cell abundances between patients' baseline samples versus healthy donors, and between baseline samples versus samples after treatment were tested by the differential abundance testing with the miloR package (https://bioconductor.org/packages/release/bioc/html/miloR.html). Specifically, the Seurat objects were converted to SingleCellExperiment objects. The PCAs and UMAPs of Seurat objects were also assigned to the new objects. Each neighborhood was assigned a cell-type label based on the majority voting of cells belonging to that neighborhood. A neighborhood was labeled as "Mixed" if the most abundant label was present in <75% of cells within that neighborhood[90].

**Reconstruction of hematopoiesis trajectories using scRNA-seq data and dynamic gene expression.** We used Monocle 2, a widely used trajectory-detection algorithm for pseudotime ordering. Monocle 2 orders cells along pseudotime that recapitulate known marker trends in the differentiation. Tracking gene expression, pathway activity, or cell-type composition changes along pseudotime enables the determination of the differentiation change for each of the terminal fates.

**Projection of patients' cells to the map of normal hematopoiesis.** To characterize early hematopoiesis in SAA patients, individual cells were projected onto the map of normal hematopoietic differentiation based on cell-by-cell comparison of patterns of global gene expression and localization to the most similar healthy donor cells with the function of transfer_cell_labels() and fix_missing_cell_labels() functions in Monocle 2. This strategy was used for mapping patients' cells on UMAP plots and assigning pseudotime estimation for cells in individual patients.

**Comparison of lineage gene with area under the receiver operating characteristic curve (AUC) scores.** AUCell package in Bioconductor was used to calculate AUC scores of lineage-specific gene expression (HSCs, MEPs, GMPs, and LymPs) of single cells in individual patients, and average AUC scores of specific lineages of all cells were plotted.

**Differential expression of genes and generation of heatmaps.** Differentially expressed genes were defined with the FindMarkers function in Seurat, by comparing gene expression in one cell subset with expression in all others. Heatmaps and network visualization were generated with ggplot2 and heatmap2 in the R package.

**GSEA.** GSEA is based on fold-changes of all detected genes. To create gene sets for a genome with custom annotations, we associated our genes with known hallmark gene sets and manually created gene sets. Fgsea[91] was used for GSEA and to plot the running normalized enrichment scores along the ranked gene list.

**Definition of pathway activity scores.** We downloaded gene lists of HALLMARK_INFLAMMATORY_RESPONSE, HALLMARK_HALLMARK_TNFA_SIGNALING_VIA_NFKB, and HALLMARK_INTERFERON_GAMMA_RESPONSE from MSigDB of GSEA, and their activity scores (expression levels) in different samples or cell populations were calculated with the AddModuleScore() function built in Seurat (http://satijalab.org/seurat/).

The activity scores were normalized with healthy donors included in individual studies, and the double-sided t-test was used to assess the difference between samples or cell populations. We defined a cell activation score, a cell cytotoxicity score, and an exhaustion score based on a set of cell cytotoxicity and exhaustion-related genes identified. The cell activation score was calculated with expression of the gene set GO:0001775. The cytotoxicity score was calculated by the AddModuleScore() function with the expression of *GZMA, GZMH, GZMM, PRF1, and GNLY*. The exhaustion score was calculated by AddModuleScore() function with the expression of *CXCL13, HAVCR2, PDCD1, TIGIT, LAG3, CTLA4, LAYN, RBPJ, VCAM1, GZMB, TOX*, and *MYO7A*.

**Analysis of sources of transcriptomic variation.** Gene expression is impacted by a number of biological factors. Decomposing gene expression into the percentage attributable to multiple biological sources of variation helps to understand their contribution magnitude. The expression variance for each gene was partitioned into the variance attributable to each variable using a linear mixed model implemented in the variancePartition package in Bioconductor. In this study, categorical variables (disease status, treatment, and disease severity) were modeled as the contributors, and the results were visualized using the ggtern package, an extension of ggplot2.

**Ligand-receptor analysis.** CellPhoneDB (v2.0.0)[46] was used to identify the putative cell–cell interactions of SAA patients and healthy donors, and between different cell populations. The code of Ktplots (https://github.com/zktuong/ktplots) was modified to visualize ligand-receptor interacting strength across cell populations from CellPhoneDB results. We also used the R package NicheNet[92] to predict ligand-receptor interactions that might drive gene expression changes in our cell types of interest. We combined all HSPCs, monocytes, and neutrophils for this analysis. All default parameters were used with the exception of setting lower cutoff thresholds of 0.3 and 0.6 for "prepare_ligand_target_visualization".

**Reconstruction of T cell differentiation trajectories using scRNA-seq data and dynamic gene or gene programs expression.** In order to estimate T cell differentiation, trajectory inference was performed with the R package Slingshot. The analyses were performed for CD8+ and CD4+ T cells separately. For each analysis, the UMAP matrix was fed into Slingshot, and a naive T cell population was manually designated as the root of all inferred trajectories, considering naive T cells differentiate to other T cells. The pseudotime variable was inferred by fitting simultaneous principal curves for further analysis.

**Estimation of RNA velocity.** Spliced and unspliced transcripts were quantified by Velocyto function with sorted BAM files of cellranger. T cell clusters were evaluated by RNA velocity analysis using scVelo (V0.2.4), separately for healthy controls and patients. Velocity-derived counts were processed, filtered, and normalized before velocity estimation on the basis of top 2000 highly variable genes with at least 20 UMIs for both spliced and unspliced transcripts across all cells. The moments facilitated the RNA velocity estimation implemented in I function scv.tl.velocity with a mode set to "dynamical". The estimated velocities were used to construct a velocity graph representing the transition probabilities among cells by the function scv.tl.velocity_graph. Finally, the velocity graph was used to embed the RNA velocities into the UMAP by the function scv.pl.velocity_embedding_stream.

**Diversity index calculation.** In this study, several methods represent the number of clones (identical TCR/BCR chains) present (richness) and their relative frequency (evenness). The Shannon entropy weighs both of these aspects of diversity equally, and it is an intuitive measure whereby the maximum value is determined by the total size of the repertoire. Entropy values decrease with increasing inequality of frequency as a result of clonal expansion. The Shannon entropy in a population of N clones with nucleotide frequency $p_i$ is defined by Eq. (1):

$$H(P) = -\sum_{i=1}^{n} p_i \log_2 p_i \qquad (1)$$

The Gini coefficient is a number aimed at measuring the inequality in a distribution. It is most often used in economics to measure a country's wealth distribution and has been widely used in diversity assessment of TCRs/BCRs[93]. The Gini index and Shannon entropy for clonality and diversity analyses are calculated with the R package of tCR (https://imminfo.github.io/tcr/).

**Definition of T cell clonal expansion and clonal dynamics.** Following a widely used criteria and considering controversial definitions of "a clone" (a single cell versus a few cells), we defined T cells as "clonally expanded" when there were >2 T cells with an identical TCR, "not-expanded" when there were 1 or 2 T cells with an identical TCR, "highly clonally expanded" when there were ≥10 T cells with an identical TCR. We repeated some analyses based on different cut-offs of defining clonally expanded clones, to test the validity of our results and conclusion; consistent results were obtained and shown in Supplementary Fig. 14a–c. For clonal dynamic pattern definition, to ensure sufficient numbers of cells for differential gene expression analysis, we first filtered to retain putative T cell clones with at least 20 cells. Then we categorized T cell clonal size dynamics after treatment into three groups: increased (>2 fold, treatment-resistant), unchanged (0.5–2 fold, treatment-insensitive), and decreased (<0.5 fold, treatment-sensitive). For statistical determination, we also analyzed clonal size dynamics using edgeR[44,45], a package wide-used in gene differential expression: increased (*P* value < 0.05 and Log Fold change > 1), decreased (*P* value > 0.05 and Log Fold change < −1), unchanged (*P* value ≥ 0.05, or *P* value < 0.05 but −1 ≤ Log Fold change ≤ 1). These two methods showed largely consistent in defining clone size dynamics and subsequent transcriptomic analysis (Supplementary Fig. 14d, e)

**Identification of TCR motifs with shard antigen specificity using GLIPH2.** GLIPH2[94] was applied to T cells of SAA patients to identify clusters of TCRs that recognized the same epitope based on CDR3β amino acid sequence similarities, with default parameters. CDR3β amino acid sequences of top 1000 most abundant CDRs were used to identify significant motif lists and associated TCR convergence groups (Supplementary Data 3). WebLogo (https://weblogo.berkeley.edu) was used to generate the sequence logos of the motifs.

**Detection of chromosome abnormalities.** We used the infercnv package in Bioconductor to distinguish aneuploid and diploid cells, with similar results. Our in-house method used Chromosome Relative Expression to identify and visualize chromosomal gain and loss. The differentially expressed genes between aneuploid and diploid cells were identified with the findMarkers function in Seurat.

**GWAS data processing and analysis**
We downloaded GWAS summary statistics of AA from the GWAS Catalog at EBI, and used the qqman R package to make a Manhattan plot. Then we used a 10-kb window around the gene body to map SNPs to genes, and H-MAGMA (v1.07) to compute gene-level association *P* values and z-scores from the GWAS data. We selected top 100 genes based on MAGMA *P* values as genetic putative disease genes.

**Pathway enrichment analysis.** We applied the built-in functions of H-MAGMA, using the results from GWAS summary statistics as its input, to examine genome-wide enriched biological pathways for AA. We calculated competitive *P* values by examining results that the combined effect of genes within a pathway was significantly greater than the combined effect of all other genes, and 10,000 permutations were used to adjust competitive *P* values.

**Specific procedures for the analysis of GWAS data at gene and pathway level.** (1) We downloaded the GWAS statistics results for AA from GWAS Catalog (https://www.ebi.ac.uk/gwas/downloads/summary-statistics), and H-MAGMA and related sources files from GitHub (https://github.com/thewonlab/H-MAGMA); (2) we ran H-MAGMA with command: magma --bfile../../MAGMA/g1000_eur --pval use=hm_rsid, p_value ncol=N --gene-annot to test the joint association of all markers and local linkage disequilibrium to assign each gene a z-score; (3) we created a rank file with the H-MAGMA *z*-score of genes; and (4) we ran GSEA with the rank files for function analysis.

After getting the *z*-scores (as to putative disease) of all genes, we integrated with scRNA-seq data to identify disease-relevant cell types and to calculate disease scores of individual cells by gene expression, as described below.

**Integration of GWAS data and scRNA-seq data.** H-MAGMA was used to perform gene-based analysis of the GWAS data to integrate scRNA-seq data.

**Inference of relevant cell types with GWAS data.** To prioritize trait-relevant cell types by looking for enrichment of the GWAS signal within functional regions, we used RolyRoly[49], which was designed to calculate effects of SNPs near protein-coding genes on cell types contributing to complex traits. RolyPoly treats a variance of each gene as the linear combination of each cell type and estimates the related coefficients. Then variances of cell-type effects were estimated using block bootstrap, and used t-statistics to test significance. Utilizing GWAS summary statistics for all SNPs, Rolypoly performed joint analysis with gene expression of a variety of cell types, to define prioritized trait-relevant cell types.

On the single-cell level, we used scDRS (v1.0.1) to quantify the aggregate expression of putative disease genes derived from GWAS summary statistics using H-MAGMA (each putative disease gene is weighted by its GWAS H-MAGMA z-score and inversely weighted by its gene-specific technical noise level in the single-cell data) in each cell of scRNA-seq data to generate cell-specific raw disease scores[48]. We created 1000 matched control gene sets (matching the gene set size, mean expression, and expression variance of the putative disease genes) and calculated 1000 sets of cell-specific raw control scores. Then, we normalized the raw disease scores and raw control scores for each cell, producing normalized disease and normalized control scores. To compute the scores described above, we used the function scdrs compute-score (--n_ctrl = 1000, --cov-file = age, sex, the number of genes per cell, and disease severity). Cells with high scores were highlighted in UMAP in Fig. 6d.

**Quantification and statistical analysis**
No statistical method was used to predetermine a sample size. No data were excluded from the analyses. We did not use any study design that required randomization or blinding. Statistical analyses were performed as described in the figure legends. Pearson correlation was used to estimate correlations among immune cell subsets. Survival was measured using the Kaplan–Meier method. Statistical significance was determined by the Multivariate Cox regression, the Kruskal–Wallis test, the Wilcoxon test, the one-way ANOVA test, and the Limma moderated t-test with the Benjamini–Hochberg method for multiple comparisons correction. A ROC curve and AUC values were generated using Prism (v.9.5.1; GraphPad software).

**Reporting summary**
Further information on research design is available in the Nature Portfolio Reporting Summary linked to this article.

## Data availability
The raw and analyzed sequencing data in this study have been deposited in the NCBI's Gene Expression Omnibus (under series accession code GSE247531) and Sequence Read Archive (under accession code PRJNA1039299). Previously published data used for analysis in this study include: GSE101660 and GSE168859. All other data are available in the article and its Supplementary files or from the corresponding author upon request. Source data are provided with this paper.

## Code availability
Code supporting this study is available at a dedicated Github repository [https://github.com/shouguog/SAA] and Zenodo [https://zenodo.org/records/15170704]. Analysis and visualization of the scRNA-seq datasets in this study can be performed at the interactive website https://haoranli22.shinyapps.io/combined_shinyapp/.

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

## Acknowledgements

The authors thank Olga Rios (NHLBI/NIH), Katherine Roskom (NHLBI/NIH), and Tania Machado (NHLBI/NIH) for assistance in obtaining samples, and patients and healthy volunteers who donated BM. Sequencing and technical support were provided by the DNA Sequencing and Genomics Core of NHLBI. FACS sorting was provided by the NHLBI flow cytometry core. This research was supported by the Intramural Research Program of the National Heart, Lung, and Blood Institute. N.S., D.C., R.A.D.R., and S.K. are supported by the Cancer Research UK City of London Centre Award [CTRQQR-2021/100004] at King's College London.

## Author contributions

Z.W. designed and performed experiments, analyzed data, and wrote the manuscript. S.G. did bioinformatics analysis and wrote the manuscript. X.F. designed and performed experiments, analyzed data, and wrote the manuscript. H.L. did bioinformatics analysis and set up the interactive website. N.S and S.J. analyzed CyTOF data. D.C., R.A.D.R., Q.G., L.A., and D.Q.R. performed experiments. S.Ka. supervised data analysis and edited manuscript. E.M.G. and B.P. collected and analyzed clinical data. S.Ko., B.P,. and N.S.Y. conceived, designed, and supervised the experiments, analyzed results, and edited the manuscript.

## Competing interests

The authors declare no competing interests.
