## [Transparent Peer Review file · Nature Communications]

Human Autoimmunity at Single Cell Resolution in Aplastic Anemia Before and After Effective Immunotherapy

Corresponding Author: Dr Zhijie Wu

Version 0:

Reviewer comments:

Reviewer #1

(Remarks to the Author)

The manuscript by Wu et al. titled "Human Autoimmunity at Single Cell Resolution: Aplastic Anemia Before and After Effective Immunotherapy" investigates the pathophysiology of severe aplastic anemia (SAA) using advanced single-cell methodologies, including CyTOF, scRNA-seq, and single-cell TCR/BCR sequencing. The study focuses on the interactions between hematopoietic stem and progenitor cells (HSPCs) and effector immune cells, both before and after immunosuppressive therapy combined with eltrombopag. The authors present findings on T cell clonal dynamics, cell-cell interactions, and the genetic background contributing to the disease, both confirming already known aspects of the disease and providing novel insights into the cellular and molecular mechanisms underlying SAA and its response to treatment.

Strengths

1. The use of single-cell multi-omics provides a comprehensive view of the cellular interactions and clonal dynamics in SAA.
2. The study ties experimental findings to clinical outcomes, enhancing the understanding of how immunotherapy affects hematopoietic recovery in SAA patients.
3. The integration of GWAS data with single-cell RNA-seq findings provides novel insights into the genetic predisposition to SAA.
4. Well documented methods for all the different experiments/analyses.

Major Comments

- o The manuscript bases some of the findings on computational predictions and inferences e.g. Lines 284-303. Functional validation of findings, such as the role of specific T cell subsets or ligand-receptor interactions in disease progression and treatment response, would strengthen these suggestive conclusions.
- o The study cohort is relatively small and not fully representative of the diverse genetic and clinical backgrounds of SAA patients. This limitation should be explicitly discussed, and caution should be advised in case of generalizing the findings.
- o While the study presents extensive data on T cell clonal dynamics, the implications of these findings for long-term treatment outcomes are not fully explored. A deeper discussion on how these clonal dynamics might influence relapse rates and overall patient prognosis would be valuable.

Minor Comments

- o Certain obvious weaknesses, many of them addressed, e.g. in lines 462-474
- o The figures are informative but dense. Simplifying some of the figures or breaking them into multiple panels could enhance readability. Additionally, including more detailed legends would help readers understand better the data presented.
- o Maybe it would be useful to state (e.g. in the supplement) the number of single cells from each sample (before/after filtering).

Very Minor Comments

- 1) Lines 115-116: "With median follow-up time of 24 months, 5 responders had a relapse requiring re-initiation of IST and 4 had developed secondary myeloid malignancy (Fig 1b and Extended Data Fig. 1d)". In Extended Data Fig. 1d, only 4 patients are marked as relapsed, whereas in the text it is stated that 5 of them did.
- 2) Extended Data Fig. 8e: CD4+T cell gens -> CD4+T cell genes
- 3) Lines 74-75: "Immune AA has clinical similarity immune-mediated hematologic diseases" -> Immune AA has clinical similarity to immune-mediated hematologic diseases

- 4) Fig.1b: Modality legend: Modalit -> Modality, what is the point of the grey CyTOF in the legend?
- 5) Fig.1c: Expression legend a bit out of place?
- 6) Line 149 & Extended Data Fig. 8e: "confirmed with well-established marker genes", which?
- 7) Line 270-271: showing more similarity

Reviewer #3

(Remarks to the Author)

This study by Wu et al. presents a detailed exploration of the immune and hematologic dynamics of severe aplastic anemia (SAA), utilizing advanced single-cell sequencing and multi-omics methodologies, including scRNAseq, CyTOF, and TCR immunosequencing. The study involves a cohort of 20 patients treated with ATG, CSA, and Eltrombopag, with a total of 93 samples analyzed. The key findings include:

- Cytotoxic CD8+ T cells are the primary effectors in the immune-mediated destruction of hematopoietic stem cells in SAA, exhibiting clonal expansion. Notably, these T cell clones are unique to individual patients and not shared across cases.
- There is incomplete recovery of early-stage hematopoietic stem cells, while later-stage progenitor cells show more robust recovery, suggesting that hematopoiesis is restored at the progenitor level rather than at the true stem cell level.
- New T cell clones may emerge during or after therapy, and their presence is associated with less robust recovery.
- The IFNG/IFNGR signaling pathway is highlighted as a significant axis of interaction, which diminishes after successful immunosuppressive therapy.

Major Comments:

• While the integrated exploration of SAA using multi-omics platforms undoubtedly provides insights into the disease's biology, it is important to acknowledge that few (if any) of these findings represent novel discoveries in the SAA field. The involvement of CD8+ T cells in the pathophysiology of aplastic anemia is well-documented, including in historical and recent references by the same authors (e.g., 10.1016/S0140-6736(04)16724-X, 10.1182/blood-2002-01-0236, 10.3324/haematol.2017.176701). The persistence of hyperexpanded T cell clones has already been shown as a marker of limited response to immunosuppressive therapy (IST). Similarly, the signatures of hematopoietic stem cells (HSCs) and progenitor cells have been extensively explored at the single-cell level (e.g., 10.1182/blood.2020008966, 10.3389/fonc.2023.1075408, 10.3389/fimmu.2023.1274116, 10.1038/s41375-022-01723-w). The role of IFN- γ signaling is also well-established as a primary biological pathway in aplastic anemia, supported by decades of in vitro and in vivo studies (e.g., 10.1182/blood-2002-01-0035, 3918301, 8639773). The TCR dynamics have also been previously tracked in SAA patients (e.g., 10.1182/blood.2021012900, 10.1038/bcj.2011.6, 10216097, 10.1182/blood.v99.10.3668, 10.1016/S0140-6736(04)16724-X). Therefore, while this study provides a valuable in-depth, multimodal analysis of longitudinal samples from a homogeneous patient cohort treated with current standards of care, the novelty of the findings is somewhat limited. This should be reflected in the discussion, emphasizing the overlap with existing literature and the limitations regarding the novelty of the study's conclusions.

- The results section would benefit from restructuring. For example, lines 123-135 seem more suited for a general discussion on the application of single-cell methodologies to SAA samples, rather than as part of the results. Additionally, the paragraph on lines 156-160 does not present results but instead describes the capabilities of scRNAseq and CyTOF in providing high-resolution data. This content should be moved to a methods or discussion section. Please revise accordingly.
- The TCR analysis could be improved by incorporating quantitative diversity metrics, such as the Simpson index, inverse Simpson index, or Shannon index, with comparisons pre- and post-treatment, and against healthy controls. The Gini index is not the most appropriate measure of diversity for this context. Additionally, the specificity analysis (which would also be of interest for the BCR repertoire) could track common clusters across SAA patients. Are these clusters more related to pathogen responses? When performing this analysis, please refer to existing literature for validation (e.g., 34805223, 34748628, 38277625).
- A deeper exploration of the BCR repertoire would provide novel insights. Specifically, is there a tendency for these B cells to recognize autoantigens or display specific SAA-related signatures?
- Broadly speaking, is there any possibility of cross-reactivity between specific TCR/BCR specificities and both pathogens and self-antigens, given the available data?
- The paragraph on malignant clonal evolution is intriguing but appears underexplored. A differential expression analysis across all patients with MDS-defining chromosomal abnormalities—not just UPN10—would be highly valuable. If UPN10 is the only case, that should be clarified. When discussing "genes involved in immune response pathways" (line 311), please specify which pathways are implicated and in which direction (e.g., immune pressure or immune suppression).
- Finally, there is no mention of HLA genes in the current manuscript. Given the prevalence of HLA-lacking clones in SAA, this study presents a unique opportunity to track the dynamics of immune escape mechanisms at the single-cell level, correlating findings pre- and post-treatment, or even pre-malignant and malignant clones.

Reviewer #4

(Remarks to the Author)

The manuscript "Human Autoimmunity at Single Cell Resolution: Aplastic Anemia Before and After Effective Immunotherapy" by Wu and colleagues from Dr. Neal Young's NHLBI group is a tour de force that employs powerful single-cell methodologies to profile the immune and hematopoietic compartments of AA patients. Building upon their pivotal phase 2 prospective study of ATG, CSA, and eltrombopag combination therapy, the authors analyzed 93 samples from 20 AA

patients using cutting-edge techniques including flow cytometry, CyTOF, single-cell RNASeq, and TCR/BCR sequencing analysis. These are large datasets and powerful techniques, and the authors present their extensive analysis artfully, followed by a thoughtful discussion. The main conclusions reported by the study are: 1) Patients' bone marrow mononuclear cells (BMMNCs) differ from controls, with an overrepresentation of lymphoid cells and decreased myeloid cells. 2) There is a loss of early hematopoietic stem and progenitor cell (HSPC) populations in patients before treatment, with recovery after treatment. 3) Lack of full hematopoietic recovery at 6 months post-treatment. Pseudotime analysis indicates very few early HSCs progressing to trilineage differentiation. 4) Frequencies of effector cells were higher in patients at baseline and lower after treatment. 5) In contrast to previous studies, Tregs were similar in AA patients compared to controls. 6) CD8+ effector T cells were clonally expanded with largely private TCR usage. 7) The authors interrogated scRNASeq data for putative ligand/receptor interactions. 8) The authors genotyped single cell transcriptomes for mutations and performed differential gene expression analysis on cells with identified mutations versus cells where no mutations were identified. 9) Gene set enrichment analysis on genes near SNPs previously associated with AA revealed associations with immune response pathways, specifically IFN- γ .

In summary, this is an impressive analysis of a large number of patients with a rare blood disease, aplastic anemia, collected prospectively, using the latest single-cell technologies. The results are framed by a beautiful and thoughtful discussion from the top leaders of the aplastic anemia field. However, most findings are largely confirmatory of already established concepts, albeit now beautifully visualized with cutting-edge single-cell methodologies and thoughtfully discussed. Several novel findings based on cutting-edge single-cell analysis are mentioned, but these are exploratory and lack experimental confirmation, making them speculative. The T cell clonotype analysis was done on very few cells, and thus conclusions drawn from this analysis are not robust.

Specific Comments:

1. Throughout the manuscript, all graphs showing parallel data in figure panels should be graphed on the same x- and y-axis scale. Further, the results discussing the findings throughout the manuscript should carefully state where the findings were statistically significant, versus when only qualitative trends are described. For example, after reviewing the scales of each graph in Figure 3F, CD8 effector cells appear to be significantly reduced compared to CD8 naïve cells in patients. Or, the data described for Fig 4D appear to be not statistically significant: "IFN- γ signaling in T cells was uniformly decreased post-treatment in all three groups; a more dramatic decrease in cytotoxicity and increase in cell apoptosis was observed in unchanged and decreased clones compared to increased clones."
2. The authors define clonal expansion as having two cells with identical TCR. This is not an established measure of clonal expansion and likely reflects a very small number of total clonotypes analyzed. If I am interpreting the supplemental data correctly, it seems that only 200 clonotypes were analyzed per patient, which would be less than 0.1% of the total T cell repertoire in each patient. This would also suggest that what the authors describe as "new clones" is likely related to the sampling error/variation. How many total T cell clonotypes were identified per patient versus per control, and what algorithm was used to downsample the reads to 200 clones? Read depth and TCRA and TCRB, and BCR sequencing data for each patient and control should be provided in the supplement. It should be stated whether CD4+ and CD8+ cells were sorted before sequencing, or if bulk sequencing of all T cells was performed. With so few healthy donors analyzed, analysis of shared clonotypes and shared GLIPH2 convergence groups is incomplete. The authors should compare their analysis to the published TCR sequences from a larger number of healthy donors.
3. It is not clear from the description which TCR sequences were analyzed by GLIPH2. The authors state that "CDR3 β amino acid sequences of the top 1,000 most abundant CDRs were used to identify significant motif lists and associated TCR convergence groups". Were these from AA patients or also from healthy controls. The sequences used for GLIPH should be listed in the supplement. The methods section states that GLIPH2 was done for VEXAS patients instead of AA patients. What were the HLA types of these patients, and were convergent TCR clusters predicted to be restricted to particular HLA alleles?
4. Lines 253-255: "T cell clonal size dynamics after treatment were arbitrarily categorized into 3 groups: increased (>2-fold, treatment-resistant), unchanged (0.5-2-fold, treatment-insensitive), and decreased (<0.5-fold, treatment-sensitive). There were 65 increased, 73 decreased, and 39 unchanged clones in total. Among 16 patients who were evaluable at 6 months, 9 of them had clone size increased and 7 decreased (Fig. 4e-f, Extended Data Fig. 15). Clone size change was inversely correlated with diversity of TCR usage (using Gini index) and with robustness of blood count recovery (Extended Data Fig. 16a-c)." These small numbers of total clones analyzed (n=177 total among all patients) is very small. What do the authors refer to as clone size in this section? Are these medians for all clones? The variation in sampling and clone sizes shown in Extended Data 15" appears stochastic and similar for all three clone patterns. No statistics are shown.
5. Line 249: "Specific antigens inducing an aberrant immune response in SAA could not be identified due to existing limited virus databases." should be rephrased, as it may imply that AA has a viral pathogenesis, which has not yet been demonstrated.
6. "TCR sequences of preexisting clones and novel clones overlapped, showing more similarity compared to controls. These results are concordant with clonal TCR usage in SAA patients, suggesting that preexisting and novel clones may target shared (unknown) antigens. Samples with higher TCR similarity in existing clones showed higher similarities in novel clones and also between novel and pre-existing clones." Could the authors please rephrase the statement for clarity. What is the metric used to determine if clones are overlapping, if these are presumably different as these are called novel? There are no data currently presented in the manuscript to support the statement "that preexisting and novel clones may target shared (unknown) antigens."
7. The discussion of cell-cell interactions based on ligand/receptor expression patterns should be carefully clarified throughout the text to make it clear that these statements refer to inferences made solely from gene expression data. For example, "These aberrant cellular interactions reduced after treatment," should be clarified to state that expression of specific receptors or ligands was reduced, potentially implying that there is a reduced interaction. Line 301: "Novel T cell clones

exhibited similar ligand-receptor interactions with HSPC as observed with preexisting clones, suggesting functional similarity and similar antigen recognition (Fig. 5f).” Another way to interpret the data is that the authors are observing ligands/receptors expression patterns that are expected for particular cell types (i.e. T-cells) and do not imply pathology. Fig 5g plots presumed interactions that range from 30-40 ligand/receptor pairs—but whether these numerical differences correspond to meaningful differences and how these relate to disease pathogenesis is unclear.

8. Fig 5E: There are 15 rows of mutations genotyped from transcriptomes but only 11 gene names, with mutation data not corresponding to specific gene names. The legend color is not consistent with the graphic. It is unclear why “silent” mutations are included in the graph and whether these were also analyzed as a part of “mutant” cells. Specific mutations that were identified by DNA NGS, and their corresponding identification in transcriptome data, along with the corresponding VAFs in various cell subsets on transcriptome data should be listed in a table in the supplement. Allele dropout is a significant limitation of this analysis, and the authors should add an explicit discussion about the extent of allele dropout and types of mutations amenable to this analysis, and what is the expected frequency of allele drop out cells being mis-called as presumed wild type. Were gene expression changes in this manuscript section analyzed by cell type, or in total?

9. Lines 44-46: “An inherited genetic background contributes to susceptibility to disease, likely by dysregulated immune activation in a cell-type-specific manner and by enhanced crosstalk between target and effector cells.” Because the authors have not genotyped patients for germline differences and have not directly shown that differences in inherited genetic background contributes to disease susceptibility, this statement is more suited for discussion, rather than a key point. Similarly, “Second, when we linked our RNA sequencing results to existing GWAS datasets, we discovered multiple immune-related genes potentially suggesting a germline background to an acquired immune disease. These novel data, uncontrived in silico, implicate genetic backgrounds as contributors to disease phenotype: cell-type-specific expression of S1PR5+ CD8+ T cells to dysregulated immune activation, and IRF5/NFKB1+ myeloid cells to exaggerated inflammation when interacting with lymphocytes.” This statement is not supported by data because no germline differences between patients or patients and controls were identified or analyzed, and no experimental confirmation that these specific genes (S1PR5, etc) are involved in pathogenesis has been performed.

10. One of the surprising findings discrepant with prior studies was that the Tregs were higher in AA patients compared to controls. Fig 1C and 1i should show Tregs along with other cells. Were these observations made with CyTOF data concordant with T reg numbers detected by standard flow cytometry? One possibility for the discrepancy can be due to having a large number of various different Treg clusters in CyTOF compared to the regular flow data. Notably, the authors' CyTOF antibody panel includes three different FOXP3 antibodies and two different CD25 antibodies. Why were there multiple Treg marker antibodies, while all other cell surface markers were analyzed using a single antibody? Were all patients and controls analyzed with the same type and number of antibodies, and were these analyzed with all redundant antibodies at the same time? Could differences in the number or type of antibodies used explain the authors' discrepant findings of increased Tregs?

Minor comments:

11. Line 118: The authors should clarify that they analyzed 93 samples from 20 patients and specify the number of samples analyzed from the three controls.

12. The hematopoietic recovery score should be validated in the whole cohort, not only the 20 patients.

13. Lines 181-184: “Differentiation potentials of HSC to MEP, GMP, and LymP trajectories in pre-treatment, post-treatment and healthy samples were comparable, appropriately governed by key transcriptional factors (Fig. 2g-h), supporting the inference that destruction of target cells occurs after lineage specification and differentiation rather than biased cell fate determination.” Could the authors please perform statistical analysis to support this comparison and rephrase the second part of the statement (“destruction of target cells occurs after lineage specification and differentiation rather than biased cell fate determination”) to clarify the conclusion.

14. Line 726: The authors included a question asking if they performed an FDR correction for this analysis in the methods. Please clarify if multiple comparison correction was performed.

15. Fig 1f legend color does not match the mutated dots.

16. Fig 1B is comprehensive but very complicated. The age color gradient from 10 to 80 is difficult to interpret; it would be easier to indicate the actual age or bin by decile with different colors. All samples were analyzed using both CyTOF and scRNASeq—so that column does not add value. Response at 3 months uses the wrong color (what does orange represent in this figure?).

Version 1:

Reviewer comments:

Reviewer #1

(Remarks to the Author)

The authors have responded to my queries adequately. I have no further comments.

Reviewer #3

(Remarks to the Author)

The authors have responded to all the comments raised. I have no further queries.

Reviewer #4

(Remarks to the Author)

The authors addressed some of the critiques, but concerns regarding rigor of methodology, validity of data interpretation and data availability, and conclusions still remain.

1. T cell analysis:

a. The use of 2 cells with the same TCR to define clonal expansion is an arbitrary, convenience-based metric that was likely selected by the authors due to the limited number of total clones that were available for analysis. 2 cells with the same TCR is not an established metric to define clonal expansion. While in their rebuttal letter, the authors cite various references that they say support this approach, none of the provided references actually use this definition of clonal expansion. In fact, one of the authors' cited studies specifically states that they used a threshold of >2 cells with the same TCR as a filter for sequencing artifacts! "In order to guard against errors produced by FACS sorting or sequencing misreads, we restricted our analysis to clones with at least two cells sharing the same TCR α and β chain CDR3 nucleotide sequences."

b. Another one of the authors' cited articles reinforces another one of this reviewer's concerns, stating: "A lack of clonal overlap among three genetically identical organisms speaks to the enormous diversity of the T cell repertoire, which has been estimated to range from 10^{15} to 10^{20} possible TCRs (Davis and Bjorkman, 1988; Lieber, 1991)." This underscores the point that analyzing a few hundred TCRs within a vast repertoire leads to concerns for sampling variability.

c. The authors' categorization of T cell clonal size dynamics after treatment into three groups—Increased (>2-fold, treatment-resistant); Unchanged (0.5–2-fold, treatment-insensitive); Decreased (<0.5-fold, treatment-sensitive)—is arbitrary. Statistical analysis should be performed to determine which of these pre- to post- treatment differences are statistically robust based on the number of T cell clones and total TCRs sampled pre- and post-treatment—to make sure that these numerical differences are not mere artifacts of limited sampling (e.g. 0 vs 1, 1 vs 3 reads with the same TCR may not statistically meaningful differences). Similarly, statistical analysis should be done to determine whether appearance of "new clones" is a clinically significant change and not simply caused limited sampling.

d. The authors should provide the sequence data for their identified T cell clones (the DNA and protein-level CDR3 sequences) in the supplement, and/or deposit in a public database for review and data sharing. Currently, the authors provided only the number of TCR clones analyzed per patient in Extended Data 4, but do not provide the TCR NGS data.

2. Concerns regarding the methodology in the section "malignant clonal evolution at single-cell resolution":

a. Authors state that they included mutation data in Extended Figure Table 3, but they only list gene names from bulk sequencing at diagnosis, without specific mutation information. Actual mutation data (Gene name, gene isoform, c., p. nomenclature and VAF) for bulk or single cell sequencing should be provided. Some of the mutations in the table are annotated as "germline", while another mutation is crossed out without indication as to why. The mutation data (gene names) in Extended Figure Table 3 do not agree with what is shown as mutations detected from scRNASeq in Figures 6d and 6e.

b. The presence of "silent" and "synonymous" mutations in Figures 6d and 6e is unclear, raising concerns that many of the depicted data points are artifacts.

c. Furthermore, based on what is shown in Figure 6 F/G, many of the listed mutations may be germline and not reflect "malignant clonal evolution". In 6F/G caption, the mutations are listed in the format U2AF1 ("T>C, alt/ref: 33/33") which should instead be listed with the proper nomenclature (p. c.). The mutations appear to be at heterozygous frequency and some are even listed as germline. The rationale for tracking "U2AF1 (germline) mutations (T>G) across baseline, 3-month, and 6-month samples" is unclear, as these are germline and non-pathogenic. Further, "CSF3R mutations (T>C, alt/ref: 33/33) in the baseline CD34+ cell sample and the same mutation (alt/ref: 55/62) in the CD34+ cell sample at 6 months; clone size was stable despite relapse and subsequent MDS-EB2 evolution."—again, the heterozygous VAF and its stability suggests the variant is a germline polymorphism rather than an acquired somatic mutation.

d. Figure 6e still contains an inconsistency, with the mutation frequency bar graph on the left side appears to be showing data for a subset of 11 mutations, while the labels and the rest of the figure show 15 mutations. The figure legend does not match the figure colors (Figures 6D and 6E), specifically gray is not explained.

e. Given all these issues, the validity of this dataset is questionable. One solution can be to remove this section on transcriptome genotyping altogether and instead focus on the more robust portions of the already extensive manuscript.

3. GWAS and transcriptome analysis:

a. It is still not completely clear how the authors are integrating the third party GWAS data with their transcriptome dataset. Figure 7C appears to show a Hallmark interferon gamma response GSEA, which is not derived from dataset uncovered by GWAS. Can the authors show a GSEA for the gene list containing their top 100 genes identified from GWAS? Is there evidence for upregulation of genes in that geneset? Figure 7d is difficult to interpret—it appears to show a small number (27-30) large dots overlaying the UMAPs corresponding to the disease scores. Lines 335-336 state that there are differences between controls and SAA patients, but no statistical analyses is provided to support that statement.

b. If inherited contribution of specific SNPs associated with AA is suspected ("An inherited genetic background appears to contribute to susceptibility to disease"), targeted genotyping of a small number of candidate germline loci for the 20 patients analyzed by transcriptomes could strengthen the conclusion of there being an inherited genetic background. Genotyping for these markers could be done quickly and inexpensively using such methods as Sanger sequencing.

4. Supplemental data should include information on detection and gating strategy for T regulatory cells using CyTOF panel. If the authors cannot support CyTOF data on Tregs with independently acquired standard multicolor flow cytometry of the same sample (as opposed to manually regating CyTOF data in flojo)—concerns that detection methods likely impacted T reg quantification leading to unexpectedly high Treg % in patients remain. Thus, the possibility of a technical issue leading to erroneously high Treg frequency should be more clearly stated when discussing T reg frequency in results.

Minor issues:

1. While the authors list the corrections to Figure 1B in their response to review, the corrections were not done, and the legend is still not matching the colors for Response and Relapse.

Version 2:

Reviewer comments:

Reviewer #4

(Remarks to the Author)

The authors have responded to all comments adequately.

Summary:

In this response letter, we have addressed all of the reviewers' comments in point-by-point fashion. We have strengthened the paper by adding new data, complementary approaches, and new analyses, leading to a significant revision. We thank the reviewers for their comments since our manuscript has improved throughout this process. As an overview, we incorporated data from additional patients and datasets, and refined analyses to directly address comments by reviewers. We believe all of their comments have been addressed with edits to the paper and summarize these changes here:

First, the reviewers asked several questions about TCR from different aspects. **1)** To expand the cohort for TCR similarity analysis, we added the previous bulk ImmunoSeq data from 12 patients and 9 healthy controls (GSE101660) to the current scTCR-seq data (20 patients pre- and post-treatment and 4 healthy controls), and provided a **new** Extended Data Table 3 to show TCR sequences for GLIPH2 analysis. **2)** We provided HLA typing data of patients in a **new** Extended Data Table 1. Patients' characteristics. **3)** We included scTCR-seq data from 6 healthy donors (published data GSE168859) for clonality and diversity scores calculation, in a **new** Extended Data (Fig. 14a). To infer potential antigens and relevant clonotypes in these patients, we added **new** analysis of TCR/BCR sequences in patients and healthy donors with IEDB database, TCRmatch and BCRmatch, and summarized results in **new** Extended Data Figs. 16 and 17. **5)** We added detailed TCR/BCR sequencing metrics, and columns of total TCR clonotypes, definition and number of T cell clones, definition of clone size, and total and top1 clone size in revised Extended Data Table 4, in response to Reviewer #4. We also followed the suggestions to look for autoantigens based on TCR/BCR specificity in response to Reviewer #3 and provided rationale for the current limitations.

Second, for suggestions of functional validation of computational analysis of cell-cell interaction, we clarified rationales and limitations of using computational analysis for cell-cell interactions based on ligand-receptor gene expression in revised Discussion. We have included analysis of downstream genes affected by ligand-receptor pairs (using NicheNet) in Extended Data Fig. 21.

Please refer to our discussion of functional validation of genomic data in our *Blood* review [1]. In brief: 1) confirmation of historical results validates single cell methods; 2) functional assays are intrinsically reductionist, almost always artificially constrained, and inadequate to the complexity of holistic single cell data; 3) there is no established statistic for "confirmation;" choice of assays, interactions, and cells is likely biased, nor is there a threshold for confirmation (50%, 80%, 10% correlation?)—indeed, we hold that decoration of genomics papers with the occasional functional correlate is likely to be misleading in the context of the vast dataset from which such testing is selected.

Requests for "functional validation" are often reflexive, self-contradictory, and reflective of a prejudice towards genomic data. *Nature Communications* itself is appropriately suspicious of immunologic results, as conventional assays are so frequently not reproducible. Although both are based on DNA sequence, GWAS is much different from single cell analyses, especially when the latter are combined with orthogonal methods as in the current work. We would also note that scRNA-seq in both basic laboratory work and disease studies is reproducible from lab to lab, as exemplified in hematology by the new picture of hematopoiesis presented compared to the conventional hierarchical models founded on colony culture and flow cytometry and consistency of inferences regarding immune aberrancies in marrow failure reports. Please also see more extensive comments below.

Third, we updated cell numbers (before and after filtering) in Extended Data Table 4, and revised figures and tables accordingly.

Fourth, for **Treg** questions from Reviewer #4, we clarified antibodies, gating strategies, and biological rationale for the current data, and included in the response letter and Extended Data Fig. 5f.

Fifth, we added relapse and clonal evolution information of each patient in Extended Data Table 1. We used blood counts data (before and after treatment) of entire cohort (137 patients) for validation of blood count scores, in a **new** Extended Data Fig. 12.

REVIEWER COMMENTS

Reviewer #1 (Remarks to the Author):

The manuscript by Wu et al. titled “Human Autoimmunity at Single Cell Resolution: Aplastic Anemia Before and After Effective Immunotherapy” investigates the pathophysiology of severe aplastic anemia (SAA) using advanced single-cell methodologies, including CyTOF, scRNA-seq, and single-cell TCR/BCR sequencing. The study focuses on the interactions between hematopoietic stem and progenitor cells (HSPCs) and effector immune cells, both before and after immunosuppressive therapy combined with eltrombopag. The authors present findings on T cell clonal dynamics, cell-cell interactions, and the genetic background contributing to the disease, both confirming already known aspects of the disease and providing novel insights into the cellular and molecular mechanisms underlying SAA and its response to treatment.

Strengths

1. The use of single-cell multi-omics provides a comprehensive view of the cellular interactions and clonal dynamics in SAA.
2. The study ties experimental findings to clinical outcomes, enhancing the understanding of how immunotherapy affects hematopoietic recovery in SAA patients.
3. The integration of GWAS data with single-cell RNA-seq findings provides novel insights into the genetic predisposition to SAA.
4. Well documented methods for all the different experiments/analyses.

Response: We thank the reviewer for these positive responses to our work!

Major Comments

o The manuscript bases some of the findings on computational predictions and inferences e.g. Lines 284-303. Functional validation of findings, such as the role of specific T cell subsets or ligand-receptor interactions in disease progression and treatment response, would strengthen these suggestive conclusions.

Response: We appreciate and understand the limitation of the current methods of “computational imputing cell-cell interactions”.

First, Inference of cell-cell communication from scRNA-seq data is a powerful technique to uncover intercellular communication pathways, and has been widely used in developmental biology, infectious, autoimmune, cancer and various other diseases and biological studies [2-12]. The cell-cell interaction analysis relies on the interactions of molecules cross cells, ligand in one cell interacting with receptor in another cell, for instance. However, traditional “orthogonal assays” of the underlying protein-protein interactions include yeast two-hybrid screening, co-immunoprecipitation, proximity labelling proteomics, fluorescence resonance energy transfer imaging and X-ray crystallography [13,14] have limited scope, and are particularly affected by individual heterogeneity and in vitro culture conditioning when applied in patients’ samples.

Secondly, while most algorithms of imputing cell-cell interactions are based on evaluating coordinated gene expression level of ligands and receptors, they also rely on assembly of protein-protein interaction databases (from proteomics studies and wet lab experiments), based on literature curated lists of interacting proteins from well-established databases (including The Human Protein Atlas [15], STRING [16], BioGRID [17], PICKE [18], APID [19], IntAct [20], et al), facilitating biological interpretation of results. Additionally, many algorithms (including CellPhone DB used in the current study) consider multi-subunit structure of ligands and receptors to represent heteromeric complexes accurately [13,14]. So far the datasets are still far from complete and accurate.

Additional efforts have incorporated information beyond ligand-receptor pairs to reveal other aspects of cell-cell communications, like downstream and targeted genes (ie. NicheNet). We now included analysis using NicheNet to study downstream genes affected by ligand-receptor interactions in CD8+ T cells and CD34+ HSPCs (**Extended Data Fig. 21 and legend**). Those downstream genes are involved in HSPC differentiation, cell cycling and apoptosis (ie. *OFS*, *MYC*, *MKP1*, *RANBP1*, *BAX*, *BIRC5*, *HMGA1*, *CDK6*), immune response (*JUNB*, *NFKBIA*, *PSMB9*, *ZFP36*, *S100A10*, *IRF8*, *IL1B*, *FKBP1A*, *VDAC1*), and differentiation (*S100A9* and *TNFSF13B*). NicheNet provided a link that ligand-to-receptor and receptor-to-target genes for intercellular communications.

Our ligand-receptor analysis first confirmed experimental observations of CD8+ T cells interacting with HSPCs via IFN γ signaling decades ago [21-23], and additionally provide a more comprehensive landscape of cell-cell interactions via other ligand-receptors, and between other cell types in SAA. We now have discussed rationales and limitations in revised Discussion.

o The study cohort is relatively small and not fully representative of the diverse genetic and clinical backgrounds of SAA patients. This limitation should be explicitly discussed, and caution should be advised in case of generalizing the findings.

Response: We acknowledge this limitation in the revised Discussion. Severe aplastic anemia is a rare disease and the 20 patients included in the current explorative laboratory cohort for single cell multi-omics study are typical of immune AA, sharing similar characteristics as the entire cohort of 137 patients treated with hATG, cyclosporine, and eltrombopag (summarized in Extended Data Fig. 1 and a new Extended Data Table 1). Note also the very high cost of sequencing.

o While the study presents extensive data on T cell clonal dynamics, the implications of these findings for long-term treatment outcomes are not fully explored. A deeper discussion on how these clonal dynamics might influence relapse rates and overall patient prognosis would be valuable.

Response: Due to the prospective design, the primary end point for the current study is 6 months after IST, and samples were thus collected at baseline and the 6 month time point, and analyses and associations were made accordingly.

We added analysis of T cell clonal dynamics and relapse during follow-up and event-free survive, there were no correlation, partly due to low number of cases (5 responders relapsed during follow-up; relative risk of 0.333 (0.05563-1.579), P value 0.3007).

Fig. Event-free survival curve and relapse-free survival curve of SAA patients with T cell clone size increased or decreased after treatment.

We observed association of a smaller T cell clone size at baseline, appearance of novel clones and a less optimal response to treatment, and novel clones share similar antigen-specificity and ligand-receptor interactions with HSPCs. These results indicate that monitoring T cell clonal dynamics could guide intensification strategies for patients showing inadequate recovery, and cyclosporine tapering during follow-up. Also, more intensive immune regimens are inferior to conventional therapy in treatment-naïve cases, yet they salvage patients who have failed initial therapy and lead to sustained hematologic recovery. These issues are addressed in the revised Discussion.

Minor Comments

o Certain obvious weaknesses, many of them addressed, e.g. in lines 462-474

Response: We have revised limitation part in Discussion according to the reviewer's comments.

o The figures are informative but dense. Simplifying some of the figures or breaking them into multiple panels could enhance readability. Additionally, including more detailed legends would help readers understand better the data presented.

Response: We have revised figures and figure legends according to the reviewer's comments, to include more information to help readers better understand.

o Maybe it would be useful to state (e.g. in the supplement) the number of single cells from each sample (before/after filtering).

Response: We have added a new column in Extended Data Table 4. Sequencing metrics summary include cell number before and after filtering.

Very Minor Comments

1) Lines 115-116: “With median follow-up time of 24 months, 5 responders had a relapse requiring re-initiation of IST and 4 had developed secondary myeloid malignancy (Fig 1b and Extended Data Fig. 1d)”. In Extended Data Fig. 1d, only 4 patients are marked as relapsed, whereas in the text it is stated that 5 of them did.

Response: There is one more patient (UPN2) relapsed on week 128 after IST. We have corrected this typo in Extended Data Fig. 1d.

2) Extended Data Fig. 8e: CD4+T cell gens -> CD4+T cell genes

Response: We have corrected this typo in the new Extended Data Fig. 9e.

3) Lines 74-75: “Immune AA has clinical similarity immune-mediated hematologic diseases” -> Immune AA has clinical similarity to immune-mediated hematologic diseases

Response: We corrected this typo in Introduction.

4) Fig.1b: Modality legend: Modalit -> Modality, what is the point of the grey CyTOF in the legend?

Response: For the majority of patients, both baseline and follow-up time points, scRNA-seq was done on BMMNCs and enriched CD34+ HSPCs, and CyTOF was done on BMMNCs. For 2 patients, only CyTOF was done on BMMNCs at baseline, without scRNA-seq, due to low cell numbers. Modality for different samples at different time points cannot be shown in the figure, so we now have removed gray CyTOF legend, and corrected typo in the modality legend.

5) Fig.1c: Expression legend a bit out of place?

Response: We revised Fig. 1c (now moved to Extended Fig. 2a) accordingly.

6) Line 149 & Extended Data Fig. 8e: “confirmed with well-established marker genes”, which?

Response: Expression of well-established marker genes for lineages were plotted on UMAP. We have revised this sentence for clarity.

7) Line 270-271: showing more similarity

Response: We have revised this sentence for clarity.

Reviewer #3 (Remarks to the Author):

This study by Wu et al. presents a detailed exploration of the immune and hematologic dynamics of severe aplastic anemia (SAA), utilizing advanced single-cell sequencing and multi-omics methodologies, including scRNAseq, CyTOF, and TCR immunosequencing. The study involves a cohort of 20 patients treated with ATG, CSA, and Eltrombopag, with a total of 93 samples analyzed. The key findings include:

- Cytotoxic CD8+ T cells are the primary effectors in the immune-mediated destruction of hematopoietic stem cells in SAA, exhibiting clonal expansion. Notably, these T cell clones are unique to individual patients and not shared across cases.
- There is incomplete recovery of early-stage hematopoietic stem cells, while later-stage progenitor cells show more robust recovery, suggesting that hematopoiesis is restored at the progenitor level rather than at the true stem cell level.
- New T cell clones may emerge during or after therapy, and their presence is associated with less robust recovery.
- The IFNG/IFNGR signaling pathway is highlighted as a significant axis of interaction, which diminishes after successful immunosuppressive therapy.

Major Comments:

- While the integrated exploration of SAA using multi-omics platforms undoubtedly provides insights into the disease's biology, it is important to acknowledge that few (if any) of these findings represent novel discoveries in the SAA field. The involvement of CD8+ T cells in the pathophysiology of aplastic anemia is well-documented, including in historical and recent references by the same authors (e.g., 10.1016/S0140-6736(04)16724-X, 10.1182/blood-2002-01-0236, 10.3324/haematol.2017.176701). The persistence of hyperexpanded T cell clones has already been shown as a marker of limited response to immunosuppressive therapy (IST). Similarly, the signatures of hematopoietic stem cells (HSCs) and progenitor cells have been extensively explored at the single-cell level (e.g., 10.1182/blood.2020008966, 10.3389/fonc.2023.1075408, 10.3389/fimmu.2023.1274116, 10.1038/s41375-022-01723-w). The role of IFN- γ signaling is also well-established as a primary biological pathway in aplastic anemia, supported by decades of in vitro and in vivo studies (e.g., 10.1182/blood-2002-01-0035, 3918301, 8639773). The TCR dynamics have also been previously tracked in SAA patients (e.g., 10.1182/blood.2021012900, 10.1038/bcj.2011.6, 10216097, 10.1182/blood.v99.10.3668, 10.1016/S0140-6736(04)16724-X). Therefore, while this study provides a valuable in-depth, multimodal analysis of longitudinal samples from a homogeneous patient cohort treated with current standards of care, the novelty of the findings is somewhat limited. This should be reflected in the discussion, emphasizing the overlap with existing literature and the limitations regarding the novelty of the study's conclusions.

Response: We thank the reviewer for acknowledging our work, and have Discussed validating and novel results in revised Discussion, based on suggested references.

- The results section would benefit from restructuring. For example, lines 123-135 seem more suited for a general discussion on the application of single-cell methodologies to SAA samples, rather than as part

of the results. Additionally, the paragraph on lines 156-160 does not present results but instead describes the capabilities of scRNAseq and CyTOF in providing high-resolution data. This content should be moved to a methods or discussion section. Please revise accordingly.

Response: We have restructured the Results as requested .

- The TCR analysis could be improved by incorporating quantitative diversity metrics, such as the Simpson index, inverse Simpson index, or Shannon index, with comparisons pre- and post-treatment, and against healthy controls. The Gini index is not the most appropriate measure of diversity for this context. Additionally, the specificity analysis (which would also be of interest for the BCR repertoire) could track common clusters across SAA patients. Are these clusters more related to pathogen responses? When performing this analysis, please refer to existing literature for validation (e.g., 34805223, 34748628, 38277625).

Response:

TCR diversity metrics. We calculated other scores representing clonality or diversity for TCR clone size in the current 20 SAA patients and four healthy donors, and combined with five more healthy controls in our previous scTCR-seq study [24]. Consistent with results shown in Fig. 4a, TCR clone size in SAA patients exhibited higher clonality (higher Gini index) and lower diversity (lower Inverse Simpson index, lower diversity index) than in healthy controls, and now results were included in a new Extended Data Fig. 14a.

TCR specificity and potential pathogen response. We now included potential pathogens imputed from TCR sequences and diseases (according to IEDB database [25], the largest repository of immune epitopes, <https://www.iedb.org/>) in SAA patients in a new Extended Data Fig. 16, and discussed limitations in revised Discussion. First, in disease perspective, compared to those in healthy donors, TCR sequences from SAA patients are more related with infection, autoimmune disease and neoplasm. Second, antigens imputed from TCR sequences in SAA patients and healthy donors shared similarity in frequencies. Majority of antigens overlapped and frequencies of antigens >0.5% were almost the same in patients and healthy donors (a new Extended Data Fig. 16b). Potential antigens were dominated by Covid virus, CMV, EBV and influenza virus, with Covid virus-related TCR sequences >60%. Apparently, interpreting potential antigens of TCR sequences is limited by the current available database of source virus of antigens, and much biased by prevalence of particular organism and how much researches focused on antigen sequences of particular organism (ie. most TCR sequences are related with Covid in both patients and healthy donors, and Covid-related sequences constitute >60% of all TCR sequences, potentially due to pandemic in the past a few years and intensive investigation of TCR/BCR sequences in Covid using advanced (single-cell) sequencing technology).

- A deeper exploration of the BCR repertoire would provide novel insights. Specifically, is there a tendency for these B cells to recognize autoantigens or display specific SAA-related signatures?

Response: With current data, we did not observe clonal expansion of B cells in SAA patients (Extended Data Fig. 14), and potential pathogens recognized by BCR repertoire in SAA patients are now summarized in a new Extended Data Fig. 17. Based on IEDB database, most BCR sequences were not

predicted to be related with any diseases. Antigens imputed from BCR sequences in SAA patients and healthy donors shared similarity in frequencies. Majority of antigens overlapped and frequencies of antigens >1% were almost the same in patients and healthy donors. Limitations for interpreting potential (auto)antigens and diseases for BCRs are same as in TCRs as discussed in response to the next comment.

- Broadly speaking, is there any possibility of cross-reactivity between specific TCR/BCR specificities and both pathogens and self-antigens, given the available data?

Response: According to the reviewer's suggestion, we have incorporated into our analysis the referenced public available data to examine TCR/BCR for pathogens and self-antigens.

First, we used TCR/BCR match to identify epitopes in IEDB [25]. We analyzed TCR and BCR sequences with IEDB database for association with allergy, autoimmune, infection, neoplasm and other diseases, and TCRs in SAA patients tend to be related more with infection, autoimmune diseases and neoplasm. While most BCR sequences were categorized as "no disease" according to IEDB (a new Extended Data Fig. 17a).

Second, we downloaded the human resource from IEDB (resource of experimentally-derived epitope information for human) with the filtering method in the website, and defined these epitopes as "self-antigens".

Last, we downloaded proteomic data from cell type-specific proteome analyses of human bone marrow cells [26], same dataset used in the study mentioned by reviewer [27], and were defined as "AA-associated autoantigens". We used these epitopes to track related proteins, and incorporated protein expression and binding specificity to our data. However, none of the self-antigens were found in TCR sequences in our current cohort, potentially due to heterogenous HLA background and a small cohort size (there are ~300 patients and ~3000 healthy controls in that study [27]).

In summary, interpretation of pathogens and self-antigens was limited by cohort size in the current study, and by current available databases. This limitation was included in revised Discussion.

- The paragraph on malignant clonal evolution is intriguing but appears underexplored. A differential expression analysis across all patients with MDS-defining chromosomal abnormalities—not just UPN10—would be highly valuable. If UPN10 is the only case, that should be clarified. When discussing "genes involved in immune response pathways" (line 311), please specify which pathways are implicated and in which direction (e.g., immune pressure or immune suppression).

Response: All 20 patients had normal karyotype at baseline, and 6 developed cytogenetic abnormalities during follow-up. We now have added a new Extended Data Table 1 to include clinical characteristics of 20 patients, including karyotypes (baseline and at the time of clonal evolution). Karyotype can be inferred using gene expression data, better for gain or loss of entire or large fraction of chromosome; current scRNA-seq data and analytical algorithms are not sensitive to detect small clones or small fraction of chromosomal gain/loss/translocation. We also have clarified in Results accordingly that "other patients were not included in the analysis due to timing of sampling, clone size, or small fraction

of chromosome change". We revised Results to explicitly describe particular pathways, also included in Fig. 6.

- Finally, there is no mention of HLA genes in the current manuscript. Given the prevalence of HLA-lacking clones in SAA, this study presents a unique opportunity to track the dynamics of immune escape mechanisms at the single-cell level, correlating findings pre- and post-treatment, or even pre-malignant and malignant clones.

Response: HLA typing results are now listed in a new Extended Data Table 1. Due to limitation of the platform and current technique (3' or 5' biased capturing of RNA, not full length, and lack of targeted amplification of targeted fracture), HLA typing information is not available on single cell level.

Reviewer #4 (Remarks to the Author):

The manuscript "Human Autoimmunity at Single Cell Resolution: Aplastic Anemia Before and After Effective Immunotherapy" by Wu and colleagues from Dr. Neal Young's NHLBI group is a tour de force that employs powerful single-cell methodologies to profile the immune and hematopoietic compartments of AA patients. Building upon their pivotal phase 2 prospective study of ATG, CSA, and eltrombopag combination therapy, the authors analyzed 93 samples from 20 AA patients using cutting-edge techniques including flow cytometry, CyTOF, single-cell RNASeq, and TCR/BCR sequencing analysis. These are large datasets and powerful techniques, and the authors present their extensive analysis artfully, followed by a thoughtful discussion. The main conclusions reported by the study are: 1) Patients' bone marrow mononuclear cells (BMMNCs) differ from controls, with an overrepresentation of lymphoid cells and decreased myeloid cells. 2) There is a loss of early hematopoietic stem and progenitor cell (HSPC) populations in patients before treatment, with recovery after treatment. 3) Lack of full hematopoietic recovery at 6 months post-treatment. Pseudotime analysis indicates very few early HSCs progressing to trilineage differentiation. 4) Frequencies of effector cells were higher in patients at baseline and lower after treatment. 5) In contrast to previous studies, Tregs were similar in AA patients compared to controls. 6) CD8+ effector T cells were clonally expanded with largely private TCR usage. 7) The authors interrogated scRNASeq data for putative ligand/receptor interactions. 8) The authors genotyped single cell transcriptomes for mutations and performed differential gene expression analysis on cells with identified mutations versus cells where no mutations were identified. 9) Gene set enrichment analysis on genes near SNPs previously associated with AA revealed associations with immune response pathways, specifically IFN- γ .

In summary, this is an impressive analysis of a large number of patients with a rare blood disease, aplastic anemia, collected prospectively, using the latest single-cell technologies. The results are framed by a beautiful and thoughtful discussion from the top leaders of the aplastic anemia field. However, most findings are largely confirmatory of already established concepts, albeit now beautifully visualized with cutting-edge single-cell methodologies and thoughtfully discussed. Several novel findings based on cutting-edge single-cell analysis are mentioned, but these are exploratory and lack experimental confirmation, making them speculative. The T cell clonotype analysis was done on very few cells, and thus conclusions drawn from this analysis are not robust.

Response: We thank the reviewer for positive comments and helpful criticism. We have clarified the number of clones and clone size definition (meaning number of cells bearing the same TCR sequence, and % in total T cell repertoire) in the responses 2 and 4 below.

Specific Comments:

1. Throughout the manuscript, all graphs showing parallel data in figure panels should be graphed on the same x- and y- axis scale. Further, the results discussing the findings throughout the manuscript should carefully state where the findings were statistically significant, versus when only qualitative trends are described. For example, after reviewing the scales of each graph in Figure 3F, CD8 effector cells appear to be significantly reduced compared to CD8 naïve cells in patients. Or, the data described for Fig 4D appear to be not statistically significant: “IFN- γ signaling in T cells was uniformly decreased post-treatment in all three groups; a more dramatic decrease in cytotoxicity and increase in cell apoptosis was observed in unchanged and decreased clones compared to increased clones.”

Response: We revised figures so that the same y-axis scale is shown for a particular comparison, as there are large variations of frequency among different cell subsets. We added *P* values in Fig. 4d, and revised legends and Results accordingly to clarify if statistically difference or there is a trend.

2. The authors define clonal expansion as having two cells with identical TCR. This is not an established measure of clonal expansion and likely reflects a very small number of total clonotypes analyzed. If I am interpreting the supplemental data correctly, it seems that only 200 clonotypes were analyzed per patient, which would be less than 0.1% of the total T cell repertoire in each patient. This would also suggest that what the authors describe as “new clones” is likely related to the sampling error/variation. How many total T cell clonotypes were identified per patient versus per control, and what algorithm was used to downsample the reads to 200 clones? Read depth and TCRA and TCRB, and BCR sequencing data for each patient and control should be provided in the supplement. It should be stated whether CD4+ and CD8+ cells were sorted before sequencing, or if bulk sequencing of all T cells was performed. With so few healthy donors analyzed, analysis of shared clonotypes and shared GLIPH2 convergence groups is incomplete. The authors should compare their analysis to the published TCR sequences from a larger number of healthy donors.

1. Clone size definition and number of clonotypes. We now have clarified these definitions and numbers for the reviewer and readers to understand.

The current single cell TCR-sequencing technique significantly improved resolution of defining T cell clonal expansion, in capturing small clones, compared to conventional flow cytometry and bulk ImmunoSeq. In single cell work, it has been standard that clonally expansion is defined as at least two cells with identical TCR (consistent with our biological knowledge), and hyperexpanded clonotypes are defined as at least 10 cells with identical TCR, in our and other’s previous publications [24, 28-31]

By this definition, as shown in Extended Data Fig. 14b, on average, ~30% of the total T cell repertoire were clonally expanded in patients, and ~20% of T cells were clonally expanded in healthy donors. Due to difference of T cell numbers, we showed % of (number of T cells with identical TCR sequences) in total T cell repertoire to represent T cell clone size, an approach used by many other single cell studies [32,33]. Top 10 T cell clones constitute ~10-20% in total T cell repertoire in patients, and ~5% in healthy

donors. For most novel clones after treatment, single clone size ranged 0.5– 15% of entire T cell repertoire.

2. Number of total T cell clonotypes for each sample is now included in TCR expression metrics sheet in Extended Data Table 4. Sequencing metrics summary. Read depth and TCRA and TCRB, and BCR sequencing data for each patient and control are also included in Extended Data Table 4. Sequencing metrics summary.

On average, there are 6770 T cell clonotypes in individual SAA patients, and 4192 in healthy donors, and total clone size 30.2% in individual SAA patients (range 10-70.3%) and 18% in healthy donors. For the sharing analysis, we arbitrarily analyzed top 200, top 500, and top 1000 clones (data not shown), on either level, there were not much sharing among individual patients and with healthy donors, indicating TCR usage is individual-specific in disease (and probably more broadly, in general population). We revised the sentence to be “There was very little sharing of top TCR clones among patients” to be more general.

3. In the current study, total BMMNCs were used for scRNA-seq and aliquoted cDNA was used for scTCR-seq. So scTCR-seq only captured T cells (expressing TCRs) in total BMMNCs. Then CD4+T and CD8+T cells were annotated based on expression of marker genes (CD4 and CD8). No sorting or enrichment of CD4+T and CD8+T was performed.

Line 242. To expand the cohort, we merged TCR-seq data from our previously published AA cohort (12 patients and 9 healthy donors, GSE101660), to examine for disease-specific TCR sequences. We are aware that a study focusing on looking for disease-specific TCR usage requires a large cohort of patients and controls; but it is a small part, not the scope of the current study. Also due to limited and biased resources to interpret potential antigens of TCR sequences as discussed above, our current work focused more on TCR clone size dynamics after treatment, not TCR sequences. We now have included this limitation in the revised Discussion.

3. It is not clear from the description which TCR sequences were analyzed by GLIPH2. The authors state that “CDR3 β amino acid sequences of the top 1,000 most abundant CDRs were used to identify significant motif lists and associated TCR convergence groups”. Were these from AA patients or also from healthy controls. The sequences used for GLIPH should be listed in the supplement. The methods section states that GLIPH2 was done for VEXAS patients instead of AA patients. What were the HLA types of these patients, and were convergent TCR clusters predicted to be restricted to particular HLA alleles?

Response: We have corrected this typo and revised method section for clarification. We now have include HLA typing of patients in a new Extended Data Table 1. Patients’ characteristics, and all TCR sequences for GLIPH2 analysis in a new Extended Data Table 3. Due to a small patient number and heterogeneous HLA background of the current cohort, we were not able to examine the convergent TCR clusters restricted to particular HLA alleles.

4. Lines 253-255: “T cell clonal size dynamics after treatment were arbitrarily categorized into 3 groups: increased (>2-fold, treatment-resistant), unchanged (0.5-2-fold, treatment-insensitive), and decreased

(<0.5-fold, treatment-sensitive). There were 65 increased, 73 decreased, and 39 unchanged clones in total. Among 16 patients who were evaluable at 6 months, 9 of them had clone size increased and 7 decreased (Fig. 4e-f, Extended Data Fig. 15). Clone size change was inversely correlated with diversity of TCR usage (using Gini index) and with robustness of blood count recovery (Extended Data Fig.16a-c).” These small numbers of total clones analyzed (n=177 total among all patients) is very small. What do the authors refer to as clone size in this section? Are these medians for all clones? The variation in sampling and clone sizes shown in Extended Data 15” appears stochastic and similar for all three clone patterns. No statistics are shown.

Response: Clone size was defined in % (T cell number of particular clone/total T cell repertoire of that individual). For the clone size dynamics analysis, dynamics were defined based on clone size change (increased (> 2 fold, treatment-resistant), unchanged (0.5-2 fold, treatment-insensitive) and decreased (< 0.5 fold, treatment-sensitive) and clone size (to keep relatively big clones and to make sure enough cells for gene expression comparison): for increased clones, pretreatment clone size should be >0.5%, and for decreased clones, posttreatment clone size should be >0.5). With this criteria, there were 65 increased, 73 decreased, and 39 unchanged clones remained for dynamics and gene expression analysis.

Clonal dynamics were analyzed in several ways. First, at the clone level, an individual clone can be categorized as increased, unchanged, or decreased based on changes of its clone size. Second, on individual patient level, each patient has various clones of increasing, decreasing or unchanged, the dominant clone dynamics pattern was classified arbitrarily (Extended Data Fig. 19) and turned out to be heterogenous but not correlated with response to treatment. We have clarified in Results, and further analysis were based on clone level, not individual patient level.

5. Line 249: “Specific antigens inducing an aberrant immune response in SAA could not be identified due to existing limited virus databases.” should be rephrased, as it may imply that AA has a viral pathogenesis, which has not yet been demonstrated.

Response: We revised this sentence for clarification.

6. “TCR sequences of preexisting clones and novel clones overlapped, showing more similarity compared to controls. These results are concordant with clonal TCR usage in SAA patients, suggesting that preexisting and novel clones may target shared (unknown) antigens. Samples with higher TCR similarity in existing clones showed higher similarities in novel clones and also between novel and pre-existing clones.” Could the authors please rephrase the statement for clarity. What is the metric used to determine if clones are overlapping, if these are presumably different as these are called novel? There are no data currently presented in the manuscript to support the statement “that preexisting and novel clones may target shared (unknown) antigens.”

Response: The TCR similarity was calculated with pairwiseAlignment function in Biostrings library using substitutionMatrix=BLOSUM100 [34]. Clones was defined as overlapped when amino acid sequences of two or more TCR sequences were the same.

Novel clones were defined in particular individual levels: for each patient, clones that were not present at baseline but present after treatment were defined as novel clones (for this individual). Clone

overlapping were analyzed across samples (among patients) to examine if there are shared TCR usages (indicating potential common antigens). This analysis was done among three groups: among preexisting clones in baseline samples (those present at baseline among patients), among novel clones only in posttreatment samples (those only present after treatment among patients), and between preexisting and novel clones only in post treatment samples.

TCR sequences of preexisting clones (in baseline samples) and novel clones (in post treatment samples) had high similarity, indicating preexisting and novel clones may target shared antigens. We now revised the sentence for clarity.

7. The discussion of cell-cell interactions based on ligand/receptor expression patterns should be carefully clarified throughout the text to make it clear that these statements refer to inferences made solely from gene expression data. For example, “These aberrant cellular interactions reduced after treatment,” should be clarified to state that expression of specific receptors or ligands was reduced, potentially implying that there is a reduced interaction. Line 301: “Novel T cell clones exhibited similar ligand-receptor interactions with HSPC as observed with preexisting clones, suggesting functional similarity and similar antigen recognition (Fig. 5f).” Another way to interpret the data is that the authors are observing ligands/receptors expression patterns that are expected for particular cell types (i.e. T-cells) and do not imply pathology. Fig 5g plots presumed interactions that range from 30-40 ligand/receptor pairs—but whether these numerical differences correspond to meaningful differences and how these relate to disease pathogenesis is unclear.

Response: We have revised several statements for clarity. As the reviewer pointed out, many ligand/receptor expression/interaction patterns are cell-type specific, and changes in disease are able to imply pathophysiology. When we compared the strength of interactions in patients before- and after-treatment, and healthy donors, and they were in general highest in patients at baseline (Fig. 5d, 5e). We acknowledge the lack of statistical comparison of ligand-receptor pairs in Fig. 5g, as there is currently no algorithm to quantitatively compare ligand-receptor usage in two different groups. We have included these limitations in revised Discussion.

8. Fig 5E: There are 15 rows of mutations genotyped from transcriptomes but only 11 gene names, with mutation data not corresponding to specific gene names. The legend color is not consistent with the graphic. It is unclear why “silent” mutations are included in the graph and whether these were also analyzed as a part of “mutant” cells. Specific mutations that were identified by DNA NGS, and their corresponding identification in transcriptome data, along with the corresponding VAFs in various cell subsets on transcriptome data should be listed in a table in the supplement. Allele dropout is a significant limitation of this analysis, and the authors should add an explicit discussion about the extent of allele dropout and types of mutations amenable to this analysis, and what is the expected frequency of allele drop out cells being mis-called as presumed wild type. Were gene expression changes in this manuscript section analyzed by cell type, or in total?

Response: We correct the error and revised the legend color in Fig. 6e accordingly. We have added Extended Figure Table 3. Patients’ characteristics to include their genomics data.

The gene expression changes described in the manuscript are in total BMMNCs or total CD34+HSPCs, and we have included limitations of genotyping using 10x Genomics scRNA-seq data in the revised Discussion.

9. Lines 44-46: “An inherited genetic background contributes to susceptibility to disease, likely by dysregulated immune activation in a cell-type-specific manner and by enhanced crosstalk between target and effector cells.” Because the authors have not genotyped patients for germline differences and have not directly shown that differences in inherited genetic background contributes to disease susceptibility, this statement is more suited for discussion, rather than a key point. Similarly, “Second, when we linked our RNA sequencing results to existing GWAS datasets, we discovered multiple immune-related genes potentially suggesting a germline background to an acquired immune disease. These novel data, uncontrived in silico, implicate genetic backgrounds as contributors to disease phenotype: cell-type-specific expression of S1PR5+ CD8+ T cells to dysregulated immune activation, and IRF5/NFKB1+ myeloid cells to exaggerated inflammation when interacting with lymphocytes.” This statement is not supported by data because no germline differences between patients or patients and controls were identified or analyzed, and no experimental confirmation that these specific genes (S1PR5, etc) are involved in pathogenesis has been performed.

Response: We thank the reviewer for pointing out this limitation. In the current study, we linked publicly available GWAS data with our scRNA-seq data, identified and linked germline variants (in a large population) to our scRNA-seq dataset to examine for cell-type specificity. This approach, although inferior to direct examination of the same cohort/individuals, has been widely used in high quality publications (MAGAM, published in 2015 in PLOS Computational Biology, has been cited >3000 times in less than ten years [35]). Many high-quality studies used MAGMA on THIRD PARTY GWAS, and most of studies [36-38] used the summary statistics of GWAS from two largest collections: <https://www.ebi.ac.uk/gwas/downloads/summary-statistics> and <https://pheweb.org/MGI/>. To increase the power, some summary statistics results are from the meta-analysis of large cohorts of different groups and are used for linking genetics data from these databases with in-house transcriptomic data. A study of simultaneous GWAS and scRNA-seq in a large cohort is very unlikely, mainly limited by the cost. We revised key point statements to be conservative, and now have clarified these limitations in Discussion.

10. One of the surprising findings discrepant with prior studies was that the Tregs were higher in AA patients compared to controls. Fig 1C and 1i should show Tregs along with other cells. Were these observations made with CyTOF data concordant with T reg numbers detected by standard flow cytometry? One possibility for the discrepancy can be due to having a large number of various different Treg clusters in CyTOF compared to the regular flow data. Notably, the authors' CyTOF antibody panel includes three different FOXP3 antibodies and two different CD25 antibodies. Why were there multiple Treg marker antibodies, while all other cell surface markers were analyzed using a single antibody? Were all patients and controls analyzed with the same type and number of antibodies, and were these analyzed with all redundant antibodies at the same time? Could differences in the number or type of antibodies used explain the authors' discrepant findings of increased Tregs?

Response: We thank the reviewer for insightful questions and agree that the lack of difference in Treg numbers between AA patients and healthy donors is indeed surprising. We have carefully examined this from both technical and scientific perspectives, as outlined below.

1. Antibodies. The current CyTOF antibody panel for Tregs was used in our previous publication [39] and thoroughly optimized since then. During optimization, we observed that using multiple clones of both CD25 and FOXP3 antibodies significantly improved signal detection and the identification of Treg populations. This is likely due to structural alterations in antibodies introduced during the metal-tagging process.

2. Gating strategy. To investigate whether the results were influenced by manual gating versus automated clustering, we reanalyzed all samples with conventional manual gating, approach used by conventional flow cytometry, by an independent expert. The comparison of Treg percentages in AA patients and healthy donors still showed no significant difference.

Fig. Comparison of Treg percentages in patients and healthy donors, defined by an automated process using BinaryClust and by manual gating using Flowjo.

We also assessed the correlation between Treg percentages obtained via manual gating and automated clustering, and a significant correlation was observed between the two methods, indicating consistent results using different gating or cell clustering strategies. Figure below shows the scatterplots distribution of Treg percentages for all samples (patients before and after treatment, and healthy controls), obtained through an automated process using BinaryClust (x-axis) or via manual gating using FlowJo (y-axis). Spearman and Pearson correlations were calculated between the percentages predicted by the two approaches, with the corresponding p-values reported. A linear regression line, along with the 95% confidence interval of the regression estimate, is plotted below.

Fig. a. The scatterplots distribution of Treg percentages for all patients and conditions (before treatment, after treatment, and healthy donors), obtained either through an automated process using BinaryClust (x-axis) or via

manual gating using FlowJo (y-axis). b. The scatterplots distribution of Treg percentages when the outlier UPN4 was excluded.

When considering all patients, the Spearman correlation is 0.4769 ($p=0.0019$) and the Pearson correlation is 0.3947 ($p = 0.0117$). However, we observed that a single sample from the "before" condition of the "UPN4" patient acted as a strong outlier, with BinaryClust yielding the lowest Treg percentage among all patients and FlowJo yielding the highest. After removing this outlier, the Spearman correlation increased to 0.5857 ($p < 0.0001$), and the Pearson correlation increased to 0.5965 ($p < 0.0001$).

These results suggest that the Treg percentages predicted by the automated BinaryClust method closely align with those obtained through expert manual gating. Based on these findings, we conclude that the absence of a difference in Treg numbers between AA patients and healthy donors is not attributable to technical issues.

2. Treg frequencies. Treg frequencies are not reduced in the current cohort of patients compared to healthy donors. This finding is surprising, as studies of our and others have reported reductions in the number and composition of Tregs in AA [39,40]. This discrepancy is likely attributable to multiple factors. Firstly, our previously published data, along with many other studies, were based on peripheral blood samples, whereas the samples in this study are from bone marrow. The use of bone marrow samples may reflect the relocation of Tregs to the site of inflammation.

More important, our cohort primarily consists of patients with very severe AA, where Tregs are exposed to higher levels of inflammation, for shorter periods (compared to non-severe AA with a chronic disease course). We have previously shown that the most functional and proliferative Tregs express high levels of CD95 (Fas) and are sensitive to FasL-induced apoptosis. We hypothesize that these Tregs (Treg-B) adopt a 'protective' phenotype by downregulating FAS expression to resist the highly inflammatory environment of the bone marrow. Indeed, our data indicate that these Tregs exhibit lower expression of the Fas-FasL pathway compared to Tregs in healthy donors (HD Treg-B)(Extended Data Fig. 5f). While these Tregs are more resistant to FASL-mediated apoptosis [41], they are less effective at suppressing pro-inflammatory cytokines such as IFN- γ and TNF- α [39].

Extended Data Fig. 5f. Comparison of Fas/FasL pathway scores of Type B Tregs in patients before-, post-treatment, and in healthy donors.

In our current cohort, following treatment, FAS expression returns to normal, and the ratio of Treg-B to CD8 T cells increases (manuscript Fig.3h). This mechanism could explain why Tregs in AA are not inherently

defective but instead switch phenotypes to protect themselves from inflammation-induced apoptosis. Given previous findings showing that Treg-B, rather than Treg-A, is the more functional and proliferative Treg subpopulation [39], we observed that the ratio of Treg-B to CD8 effector cells decreased in SAA and increased after treatment. This suggests the recovery of the immunosuppressive function of Tregs post-therapy.

As a subpopulation of CD4+ T cells, Treg abundance were shown in Fig. 3g-h, and supplemental Fig. 4. We now have included technical and biological backgrounds in Methods and Discussion.

Minor comments:

11. Line 118: The authors should clarify that they analyzed 93 samples from 20 patients and specify the number of samples analyzed from the three controls.

Response: We now specify the number of samples analyzed for the four healthy controls (Line 119).

12. The hematopoietic recovery score should be validated in the whole cohort, not only the 20 patients.

Response: In addition to the current Extended Data Fig. 11 (hematopoietic recovery score validation in 20 patients), we now have added a **new Extended Data Fig. 12** to show good correlation of hematopoietic score and recovery score with blood counts and clinical response in entire cohort of 137 patients.

13. Lines 181-184: "Differentiation potentials of HSC to MEP, GMP, and LymP trajectories in pre-treatment, post-treatment and healthy samples were comparable, appropriately governed by key transcriptional factors (Fig. 2g-h), supporting the inference that destruction of target cells occurs after lineage specification and differentiation rather than biased cell fate determination." Could the authors please perform statistical analysis to support this comparison and rephrase the second part of the statement ("destruction of target cells occurs after lineage specification and differentiation rather than biased cell fate determination") to clarify the conclusion.

Response: We have added a statistic (*P* values) to Fig. 2g-h and rephrased the statement in Lines 181-184.

14. Line 726: The authors included a question asking if they performed an FDR correction for this analysis in the methods. Please clarify if multiple comparison correction was performed.

Response: Milo uses an adaptation of the Spatial FDR correction introduced by cydar, which accounts for the overlap between neighborhoods. We now clarified and removed the question in text.

15. Fig 1f legend color does not match the mutated dots.

Response: The color of new Fig. 1g is now swapped for consistency with Fig. 1f, and we fixed the same issue for Extended Data Fig. 3a.

16. Fig 1B is comprehensive but very complicated. The age color gradient from 10 to 80 is difficult to interpret; it would be easier to indicate the actual age or bin by decile with different colors. All samples were analyzed using both CyTOF and scRNASeq—so that column does not add value. Response at 3 months uses the wrong color (what does orange represent in this figure?).

Response: We have revised Fig 1b accordingly: 1) changed age color legend, 2) changed modality legend, 3) revised colors for response and relapse.

References

1. Wu Z, Young NS. Single-cell genomics in acquired bone marrow failure syndromes. *Blood*. 2023 Oct 5;142(14):1193-1207.
2. Chen, S., Peng, Y., Zhang, X., Jiang, T., Fang, B., Zhang, P., Li, Y., Ren, Y., & Sun, Y. (2024). Ligand-Receptor Interactions for Cell-Cell Communication Analysis in Rat, Chicken, Pig, and Monkey Single-Cell and Spatial Transcriptomics. *bioRxiv*.
3. Zhao, W., Johnston, K., Ren, H., Xu, X., & Nie, Q. (2023). Inferring neuron-neuron communications from single-cell transcriptomics through NeuronChat. *Nature Communications*, 14.
4. Wahiduzzaman, M., Liu, Y., Huang, T., Wei, W., & Li, Y. (2022). Cell-cell communication analysis for single-cell RNA sequencing and its applications in carcinogenesis and COVID-19. *Biosafety and Health*.
5. Jin, J., Yu, S., Lu, P., & Cao, P. (2023). Deciphering plant cell–cell communications using single-cell omics data. *Computational and Structural Biotechnology Journal*, 21, 3690 - 3695.
6. AlMusawi, S., Ahmed, M., & Nateri, A.S. (2021). Understanding cell-cell communication and signaling in the colorectal cancer microenvironment. *Clinical and Translational Medicine*, 11.
7. Puram, S. V. et al. Single-cell transcriptomic analysis of primary and metastatic tumor ecosystems in head and neck cancer. *Cell* 171, 1611–1624 (2017).
8. Camp, J. G. et al. Multilineage communication regulates human liver bud development from pluripotency. *Nature* 546, 533–538 (2017).
9. Pavlièev, M. et al. Single-cell transcriptomics of the human placenta: inferring the cell communication network of the maternal–fetal interface. *Genome Res.* 27, 349–361 (2017).
10. Zepp, J. A. et al. Distinct mesenchymal lineages and niches promote epithelial self-renewal and myofibrogenesis in the lung. *Cell* 170, 1134–1148 (2017).
11. Cohen, M. et al. Lung single-cell signaling interaction map reveals basophil role in macrophage imprinting. *Cell* 175, 1031–1044 (2018).
12. Vento-Tormo, R. et al. Single-cell reconstruction of the early maternal–fetal interface in humans. *Nature* 563, 347–353 (2018).
13. Armingol, E., Officer, A., Harismendy, O. et al. Deciphering cell–cell interactions and communication from gene expression. *Nat Rev Genet* 22, 71–88 (2021).
14. Almet AA, Cang Z, Jin S, Nie Q. The landscape of cell-cell communication through single-cell transcriptomics. *Curr Opin Syst Biol*. 2021 Jun;26:12-23.
15. Thul PJ, Lindskog C. The human protein atlas: A spatial map of the human proteome. *Protein Sci*. 2018 Jan;27(1):233-244.
16. Damian Szklarczyk, Rebecca Kirsch, Mikaela Koutrouli, Katerina Nastou, Farrokh Mehryary, Radja Hachilif, Annika L Gable, Tao Fang, Nadezhda T Doncheva, Sampo Pyysalo, Peer Bork, Lars J Jensen, Christian von Mering, The STRING database in 2023: protein–protein association networks and functional enrichment analyses for any sequenced genome of interest, *Nucleic Acids Research*, Volume 51, Issue D1, 6 January 2023, Pages D638–D646
17. Oughtred R, Rust J, Chang C, Breitkreutz BJ, Stark C, Willems A, Boucher L, Leung G, Kolas N, Zhang F, Dolma S, Coulombe-Huntington J, Chatr-Aryamontri A, Dolinski K, Tyers M. The BioGRID database: A comprehensive biomedical resource of curated protein, genetic, and chemical interactions. *Protein Sci*. 2021 Jan;30(1):187-200.
18. Georgios N Dimitrakopoulos, Maria I Klapa, Nicholas K Moschonas, PICKLE 3.0: enriching the human meta-database with the mouse protein interactome extended via mouse–human orthology, *Bioinformatics*, Volume 37, Issue 1, January 2021, Pages 145–146.

19. Diego Alonso-López, Francisco J Campos-Laborie, Miguel A Gutiérrez, Luke Lambourne, Michael A Calderwood, Marc Vidal, Javier De Las Rivas, APID database: redefining protein–protein interaction experimental evidences and binary interactomes, Database, Volume 2019, 2019, baz005.
20. Noemi del Toro, Anjali Shrivastava, Eliot Ragueneau, Birgit Meldal, Colin Combe, Elisabet Barrera, Livia Perfetto, Karyn How, Prashansa Ratan, Gautam Shirodkar, Odilia Lu, Bálint Mészáros, Xavier Watkins, Sangya Pundir, Luana Licata, Marta Iannuccelli, Matteo Pellegrini, Maria Jesus Martin, Simona Panni, Margaret Duesbury, Sylvain D Vallet, Juri Rappsilber, Sylvie Ricard-Blum, Gianni Cesareni, Lukasz Salwinski, Sandra Orchard, Pablo Porras, Kalpana Panneerselvam, Henning Hermjakob, The IntAct database: efficient access to fine-grained molecular interaction data, Nucleic Acids Research, Volume 50, Issue D1, 7 January 2022, Pages D648–D653.
21. Hoffman R, Zanjani ED, Lutton JD, Zalusky R, Wasserman LR. Suppression of erythroid-colony formation by lymphocytes from patients with aplastic anemia. N Engl J Med. 1977 Jan 6;296(1):10-3.
22. Bacigalupo A, Podestà M, Mingari MC, Moretta L, Van Lint MT, Marmont A. Immune suppression of hematopoiesis in aplastic anemia: activity of T-gamma lymphocytes. J Immunol. 1980 Oct;125(4):1449-53.
23. Gravano DM, Al-Kuhlani M, Davini D, Sanders PD, Manilay JO, Hoyer KK. CD8+ T cells drive autoimmune hematopoietic stem cell dysfunction and bone marrow failure. J Autoimmun. 2016 Dec;75:58-67.
24. Gao, S., Wu, Z., Arnold, B. et al. Single-cell RNA sequencing coupled to TCR profiling of large granular lymphocyte leukemia T cells. Nat Commun 13, 1982 (2022).
25. Randi Vita, Nina Blazeska, Daniel Marrama, IEDB Curation Team Members , Sebastian Duesing, Jason Bennett, Jason Greenbaum, Marcus De Almeida Mendes, Jarjapu Mahita, Daniel K Wheeler, Jason R Cantrell, James A Overton, Darren A Natale, Alessandro Sette, Bjoern Peters, The Immune Epitope Database (IEDB): 2024 update, Nucleic Acids Research, 2024;, gkae1092.
26. Henrich, M.L., Romanov, N., Horn, P. et al. Cell-specific proteome analyses of human bone marrow reveal molecular features of age-dependent functional decline. Nat Commun 9, 4004 (2018).
27. Pagliuca S, Gurnari C, Awada H, Kishtagari A, Kongkiatkamon S, Terkawi L, Zawit M, Guan Y, LaFramboise T, Jha BK, Patel BJ, Hamilton BK, Majhail NS, Lundgren S, Mustjoki S, Sauntharajah Y, Visconte V, Chan TA, Yang CY, Lenz TL, Maciejewski JP. The similarity of class II HLA genotypes defines patterns of autoreactivity in idiopathic bone marrow failure disorders. Blood. 2021 Dec 30;138(26):2781-2798.
28. Huuhtanen, J., Bhattacharya, D., Lönnberg, T. et al. Single-cell characterization of leukemic and non-leukemic immune repertoires in CD8+ T-cell large granular lymphocytic leukemia. Nat Commun 13, 1981 (2022).
29. Kasmani MY, Zander R, Chung HK, Chen Y, Khatun A, Damo M, Topchyan P, Johnson KE, Levashova D, Burns R, Lorenz UM, Tarakanova VL, Joshi NS, Kaech SM, Cui W. Clonal lineage tracing reveals mechanisms skewing CD8+ T cell fate decisions in chronic infection. J Exp Med. 2023 Jan 2;220(1):e20220679.
30. Kim HJ, Ban JJ, Kang J, Im HR, Ko SH, Sung JJ, Park SH, Park JE, Choi SJ. Single-cell analysis reveals expanded CD8+ GZMKhigh T cells in CSF and shared peripheral clones in sporadic amyotrophic lateral sclerosis. Brain Commun. 2024 Nov 27;6(6):fcae428.
31. Khatun A, Kasmani MY, Zander R, Schauder DM, Snook JP, Shen J, Wu X, Burns R, Chen YG, Lin CW, Williams MA, Cui W. Single-cell lineage mapping of a diverse virus-specific naive CD4 T cell repertoire. J Exp Med. 2021 Mar 1;218(3):e20200650.

32. Thapa DR, Tonikian R, Sun C, Liu M, Dearth A, Petri M, Pepin F, Emerson RO, Ranger A. Longitudinal analysis of peripheral blood T cell receptor diversity in patients with systemic lupus erythematosus by next-generation sequencing. *Arthritis Res Ther*. 2015 May 23;17(1):132.
33. Iarenbeek PL, de Hair MJ, Doorenspleet ME, van Schaik BD, Esveltdt RE, van de Sande MG, Cantaert T, Gerlag DM, Baeten D, van Kampen AH, Baas F, Tak PP, de Vries N. Inflamed target tissue provides a specific niche for highly expanded T-cell clones in early human autoimmune disease. *Ann Rheum Dis*. 2012 Jun;71(6):1088-93.
34. Malde K. The effect of sequence quality on sequence alignment. *Bioinformatics*. 2008 Apr 1;24(7):897-900. doi: 10.1093/bioinformatics/btn052. Epub 2008 Feb 23.
35. de Leeuw CA, Mooij JM, Heskens T, Posthuma D. MAGMA: generalized gene-set analysis of GWAS data. *PLoS Comput Biol*. 2015 Apr 17;11(4):e1004219.
36. Jagadeesh KA, Dey KK, Montoro DT, Mohan R, Gazal S, Engreitz JM, Xavier RJ, Price AL, Regev A. Identifying disease-critical cell types and cellular processes by integrating single-cell RNA-sequencing and human genetics. *Nat Genet*. 2022 Oct;54(10):1479-1492.
37. Kar SP, Quiros PM, Gu M, Jiang T, Mitchell J, Langdon R, Iyer V, Barcena C, Vijayabaskar MS, Fabre MA, Carter P, Petrovski S, Burgess S, Vassiliou GS. Genome-wide analyses of 200,453 individuals yield new insights into the causes and consequences of clonal hematopoiesis. *Nat Genet*. 2022 Aug;54(8):1155-1166.
38. Edahiro R, Shirai Y, Takeshima Y, Sakakibara S, Yamaguchi Y, Murakami T, Morita T, Kato Y, Liu YC, Motooka D, Naito Y, Takuwa A, Sugihara F, Tanaka K, Wing JB, Sonehara K, Tomofuji Y; Japan COVID-19 Task Force; Namkoong H, Tanaka H, Lee H, Fukunaga K, Hirata H, Takeda Y, Okuzaki D, Kumanogoh A, Okada Y. Single-cell analyses and host genetics highlight the role of innate immune cells in COVID-19 severity. *Nat Genet*. 2023 May;55(5):753-767.
39. Kordasti S, Costantini B, Seidl T, Perez Abellan P, Martinez Llordella M, McLornan D, Diggins KE, Kulasekararaj A, Benfatto C, Feng X, Smith A, Mian SA, Melchiotti R, de Rinaldis E, Ellis R, Petrov N, Povoleri GA, Chung SS, Thomas NS, Farzaneh F, Irish JM, Heck S, Young NS, Marsh JC, Mufti GJ. Deep phenotyping of Tregs identifies an immune signature for idiopathic aplastic anemia and predicts response to treatment. *Blood*. 2016 Sep 1;128(9):1193-205. doi: 10.1182/blood-2016-03-703702. Epub 2016 Jun 8.
40. Solomou EE, Rezvani K, Mielke S, Malide D, Keyvanfar K, Visconte V, Kajigaya S, Barrett AJ, Young NS. Deficient CD4+ CD25+ FOXP3+ T regulatory cells in acquired aplastic anemia. *Blood*. 2007 Sep 1;110(5):1603-6.
41. Lim SP, Costantini B, Mian SA, Perez Abellan P, Gandhi S, Martinez Llordella M, Lozano JJ, Antunes Dos Reis R, Povoleri GAM, Mourikis TP, Abarrategi A, Ariza-McNaughton L, Heck S, Irish JM, Lombardi G, Marsh JCW, Bonnet D, Kordasti S, Mufti GJ. Treg sensitivity to FasL and relative IL-2 deprivation drive idiopathic aplastic anemia immune dysfunction. *Blood*. 2020 Aug 13;136(7):885-897.

REVIEWER COMMENTS

Reviewer #4 (Remarks to the Author):

The authors addressed some of the critiques, but concerns regarding rigor of methodology, validity of data interpretation and data availability, and conclusions still remain.

1. T cell analysis:

a. The use of 2 cells with the same TCR to define clonal expansion is an arbitrary, convenience-based metric that was likely selected by the authors due to the limited number of total clones that were available for analysis. 2 cells with the same TCR is not an established metric to define clonal expansion. While in their rebuttal letter, the authors cite various references that they say support this approach, none of the provided references actually use this definition of clonal expansion. In fact, one of the authors' cited studies specifically states that they used a threshold of >2 cells with the same TCR as a filter for sequencing artifacts! "In order to guard against errors produced by FACS sorting or sequencing misreads, we restricted our analysis to clones with at least two cells sharing the same TCR α and β chain CDR3 nucleotide sequences."

Response:

Definition of clones.

1) Of course the reviewer is correct that the criterion of using 2 cells to define clone is "arbitrary," but a more accurate term is "by convention," among many other terms in the laboratory, clinic, and widespread practice: definition of differentially expressed genes using a fold change of 2; 20% of blasts in bone marrow for differential diagnosis of AML; differential of 200 cells for hematopoietic diseases diagnosis; 20 cells for cytogenetics analysis, definition of acute or chronic GVHD after transplantation using a 3 month time point, et al. All are commonly employed, without statistical bases.

As the reviewer pointed out, there is no established or uniform metric to define clonal expansion. We carefully reviewed dozens of scTCR-seq and Immuno-seq research papers with similar approach, and found definition of a clone remains controversial, especially when considering limitations of techniques in detecting "every cell". In fact, definition of a clone in scTCR-seq or bulk TCR-seq studies remain arbitrary: >2 cells (PMCID: PMC8776501, Cell. 2022; Han J, et al. Nat Cancer. 2021), at least 2 cells (Huuhtanen J, et al. Nat Commun. 2022; Gao S, et al. Nat Commun. 2022), 0.5% or 1% (Faurfax BP, et al. Nat

Med. 2020; Thapa DR, et al. Arthritis Res Ther. 2015; Klarenbeek PL, et al. Ann Rheum Dis. 2012).

2) “Clone” as a term is widely understood but resists categorical definition—see Mustjoki and Young (Mustjoki S, Young NS. N Engl J Med. 2021) and supporting literature, for the inherent genetic mosaicism of normal tissues and the difficulty of defining even malignant clones. For T cells, we chose ≥ 2 cells with identical TCR sequences following other groups (Huuhtanen J, et al. Nat Commun. 2022; Gao S, et al. Nat Commun. 2022), and for consistency with our many other publications (Wu Z, et al. Cell Rep Med. 2023; Rejeski K, et al. Blood. 2022; Wu Z, et al. J Leukoc Biol. 2022). **This methodology has never been questioned by any of the many reviewers of our work!**

3) Clonal expansion of T cells is normal. We applied the same criteria to both patients and healthy controls, and observed a higher degree of clonal expansion in patients. It should be obvious that we are comparing degrees of clonal expansion, *not* denying clonality in healthy controls. Indeed, we used different approaches to quantify clonality, including but not limited to number of expanded clones (based on the ≥ 2 cells definition): clone size with various cut-offs (2, 3-9, ≥ 10 in Fig. 4a right panel), Gini index of clone size (Figure 4b), % of top clones (Extended Data Fig. 14b,c). Results of these analyses are consistent and independent of clone definition thresholds.

4) For clarification, we have revised the submitted version to explicitly define: clonally expanded (>2 cells), highly clonally expanded (≥ 10 cells), not expanded (1 or 2 cells), and revised description of results, methods, and figures accordingly as below.

Results. Line 220, “Abundance of clonally expanded T cells (> 2 cells with identical TCR)”.

Methods. We now added a new paragraph of “definition of T cell clonal expansion and clonal dynamics” in Lines 954-968 as below.

Definition of T cell clonal expansion and clonal dynamics. Following a widely-used criterion and considering varying definitions of “clone” (a single cell versus a few cells), we defined T cells as “clonally expanded” when there were > 2 T cells with identical TCR, “not-expanded” when there were 1 or 2 T cells with identical TCR, “highly clonally expanded” when there were ≥ 10 T cells with identical TCR. We repeated some analysis based on different cut-offs of defining clonally expanded clones, to test validity of our results and conclusion; consistent results were obtained and shown in Extended Data Fig. 14a-c. For clonal dynamic pattern definition, to ensure sufficient numbers of cells for differential gene expression analysis, we first filtered to retain putative T cell clone of at least 20 cells. Then we categorized T cell clonal size dynamics after treatment into 3 groups: increased (> 2 fold, treatment-resistant), unchanged (0.5-2 fold, treatment-insensitive) and decreased ($<$

0.5 fold, treatment-sensitive). For statistical determination, we also analyzed clonal size dynamics using edgeR, a package wide-used in gene differential expression: increased (P value < 0.05 and Log Fold change > 1), decreased (P value > 0.05 and Log Fold change < -1), unchanged (P value ≥ 0.05 , or P value < 0.05 but $-1 \leq \text{Log Fold change} \leq 1$). These two methods showed largely consistency in defining clone size dynamics and subsequent transcriptomic analysis (Extended Data Fig. 14d,e).

Figures. We now have revised Fig. 4a (left panel, total clone%), Fig. 4c (right panel, cytotoxicity and TNF scores and cell subtypes), and Extended Data Table 4 using the updated definition (> 2 T cells with identical TCR). To be noted, although not as striking as in leukemic diseases (such as T-cell acute lymphoblastic leukemia, T-cell lymphoma, and T-cell large granular lymphocytic leukemia), we and others showed there were T cell clonal expansion in aplastic anemia (Giudice V, et al. Haematologica. 2018; Risitano AM, et al. Lancet. 2004; Schuster FR, et al. Blood Cancer. 2011), as compared to healthy donors; in the current study, the same criteria of clone definition was applied to SAA patients and healthy donors for comparison.

Additionally, we repeated analyses with different cut-offs for clonal expansion (≥ 5 cells, ≥ 10 cells, and ≥ 20 cells), and obtained largely consistent results, shown in a new Extended Data Fig. 15a-c. These results suggested the validity of our data analysis, with an inevitable arbitrary but rigorous methodology.

a With cut-off ≥ 5 cells

b With cut-off ≥ 10 cells

c With cut-off ≥ 20 cells

b. Another one of the authors' cited articles reinforces another one of this reviewer's concerns, stating: "A lack of clonal overlap among three genetically identical organisms speaks to the enormous diversity of the T cell repertoire, which has been estimated to range from 10^{15} to 10^{20} possible TCRs (Davis and Bjorkman, 1988; Lieber, 1991)." This underscores the point that analyzing a few hundred TCRs within a vast repertoire leads to concerns for sampling variability.

Response: There appears to some confusion between theoretical possibilities and empiric data: mathematically generated probabilities are not equivalent to experimental data, from within a population or individual. To our knowledge, the experiment to demonstrate sequences available in either has not been performed as impractical. In the current study, we captured an average 6518 clonotypes (median 6127, and 25-75 percentile range: 3184-7868) per sample, a number variable among samples due to technical limitations, amount of cells available for assays, disease severity and treatment, individual heterogeneity, etc. To be noted, our patients had severe hypoplasia of the bone marrow before treatment, with T cell dominance, and elimination of T cells after treatment. Therefore, profiling of complete T cell clonotype in the current study (and many other studies) was limited by patient cohort (a few or millions of patients), availability of samples (1ml bone marrow or

100ml bone marrow aspirate), disease and pathological/biological conditions (T cell infiltration or T cell elimination), cost (sample number and cell number), and sequencing techniques. We have included in the Discussion these and other limitations (Lines 483-486).

c. The authors' categorization of T cell clonal size dynamics after treatment into three groups—Increased (>2-fold, treatment-resistant); Unchanged (0.5–2-fold, treatment-insensitive); Decreased (<0.5-fold, treatment-sensitive)

—is arbitrary. Statistical analysis should be performed to determine which of these pre-to post- treatment differences are statistically robust based on the number of T cell clones and total TCRs sampled pre- and post-treatment—to make sure that these numerical differences are not mere artifacts of limited sampling (e.g. 0 vs 1, 1 vs 3 reads with the same TCR may not statistically meaningful differences). Similarly, statistical analysis should be done to determine whether appearance of “new clones” is a clinically significant change and not simply caused limited sampling.

Response:

1) Definition of clone size dynamics.

As one of our purposes was to characterize clonal dynamics and corresponding transcriptomic changes, we first filtered to retain clones with ≥ 20 cells for further analysis (for feasibility of comparison of gene expression), and a cut-off of 0.5 and 2 was used for consistency with the many, many publications addressing differentially expressed genes. The biological basis for these choices of increased/decreased genes has been understood and accepted since the introduction of microarrays decades ago (DeRisi J, et al. Nat Genet. 1996; ter Linde, et al. J Bacteriol. 1999; Wellmann A, et al. Blood. 2000; Tarca AL, et al. Am J Obstet Gynecol. 2006; Dalman MR, et al. BMC Bioinformatics. 2012). In microarray data analysis, low expressed genes are usually filtered out due to high risk to be falsely identified as differentially expressed; analogue to the problem raised by the reviewer of our fold change approach (e.g. 0 vs 1, 1 vs 3 reads with the same TCR may not statistically meaningful differences). The current fold-change method has a lower chance of false positives when restricted to larger clones (≥ 20 cells in our study), as illustrated in widely-used MA plot for RNA-seq or microarray analysis (Robinson MD, et al. Bioinformatics. 2010). The consistency of using the current fold-change method and edgeR in defining clone size dynamics is shown below.

edgeR is a package widely used for differential gene expression in RNA-seq and Chip-seq data analysis (Robinson MD, et al. Bioinformatics. 2010; Chen Y, et al. bioRxiv. 2014), and a few studies used it to identified change of clones with TCR-seq data (Goncharov MM, et al.

Elife. 2022; Pollastro S. et al. Front Immunol. 2021; Pétremand R, et al. Nat Biotechnol. 2024). Based on the reviewer's comments, we now analyzed clonal size dynamics using edgeR: increased (P value < 0.05 and Log Fold change > 1), decreased (P value > 0.05 and Log Fold change < -1), unchanged (P value ≥ 0.05 , or P value < 0.05 but $-1 \leq \text{Log Fold change} \leq 1$). There was largely consistency in defining clone size dynamics using edgeR and our fold-change-only method, and subsequent transcriptomic analysis also showed consistent results (Extended Data Fig. 14d,e). This alternative approach is now explicit in Methods, Lines 954-968, "Definition of T cell clonal expansion and clonal dynamics".

edgeR and DESeq are two widely-used packages (cited 39,702 times and 77,412 times so far, respectively) for differential analysis in the form of read counts for genes or genomic features, such as RNA-seq or ChIP-seq (Robinson MD, et al. Bioinformatics. 2010; Chen Y, et al. bioRxiv. 2014; Love MI, et al. Genome Biol. 2014). edgeR is not a perfect algorithm when lacking replicates, as for one-vs-one, as in the present study. Due to high cost of RNA-seq 10 years ago, DESeq has been widely employed for one to one data, but this function was removed

DESeq2(<https://bioconductor.org/packages/release/bioc/vignettes/DESeq2/inst/doc/DESeq2.htm>). New algorithms need to be developed for one vs one data in future.

2) Appearance of "new clones".

Clones with significant changes and large number of clone size are better recognized by edgeR, and smaller clones require larger fold change to be identified as difference, just as shown in famous MA plot (Robinson MD, et al. Bioinformatics. 2010).

We followed an approach to define preexisting clones and new clones published on Science Immunology (on page 11 of 19, Oliveira G, et al. Sci Immunol.2023).

d. The authors should provide the sequence data for their identified T cell clones (the DNA and protein-level CDR3 sequences) in the supplement, and/or deposit in a public database

for review and data sharing. Currently, the authors provided only the number of TCR clones analyzed per patient in Extended Data 4, but do not provide the TCR NGS data.

Response: We have revised Extended Data Table 4 for the updated cut-off of >2 T cells with identical TCR for clone definition, including total TCR clonotypes, number of expanded clones, top1 clone size (%), clone size of all expanded clones (%), for individual sample.

Due to file size, original TCR and BCR sequencing data and filtered contig_annotations files (including single cell barcode, nucleotide sequences, amino acid sequences, cell counts, et al.) for each sample were uploaded to GSE247531 as shown below in snapshot (<https://www.ncbi.nlm.nih.gov/geo/query/acc.cgi?acc=GSE247531>, now available to reviewers with token and will be available to public after publication).

Relations

BioProject PRJNA1039299

Supplementary file	Size	Download	File type/resource
GSE247531_BCR_filtered_contig_annotations.csv.gz	20.6 Mb	(http)	CSV
GSE247531_BMcounts.Rdata.gz	1.7 Gb	(http)	RDATA
GSE247531_CD34counts.Rdata.gz	1.5 Gb	(http)	RDATA
GSE247531_TCR_filtered_contig_annotations.csv.gz	93.4 Mb	(http)	CSV

SRA Run Selector 
Raw data are available in SRA

2. Concerns regarding the methodology in the section “malignant clonal evolution at single-cell resolution”:

a. Authors state that they included mutation data in Extended Figure Table 3, but they only list gene names from bulk sequencing at diagnosis, without specific mutation information. Actual mutation data (Gene name, gene isoform, c., p. nomenclature and VAF) for bulk or single cell sequencing should be provided. Some of the mutations in the table are annotated as “germline”, while another mutation is crossed out without indication as to why. The mutation data (gene names) in Extended Figure Table 3 do not agree with what is shown as mutations detected from scRNASeq in Figures 6d and 6e.

b. The presence of “silent” and “synonymous” mutations in Figures 6d and 6e is unclear, raising concerns that many of the depicted data points are artifacts.

c. Furthermore, based on what is shown in Figure 6 F/G, many of the listed mutations may be germline and not reflect “malignant clonal evolution”. In 6F/G caption, the mutations are listed in the format U2AF1 (“T>C, alt/ref: 33/33”) which should instead be listed with the proper nomenclature (p. c.). The mutations appear to be at heterozygous frequency and some are even listed as germline. The rationale for tracking “U2AF1 (germline) mutations (T>G) across baseline, 3-month, and 6-month samples” is unclear, as these are germline

and non-pathogenic. Further, “CSF3R mutations (T>C, alt/ref: 33/33) in the baseline CD34+ cell sample and the same mutation (alt/ref: 55/62) in the CD34+ cell sample at 6 months; clone size was stable despite relapse and subsequent MDS-EB2 evolution.”—again, the heterozygous VAF and its stability suggests the variant is a germline polymorphism rather than an acquired somatic mutation.

d. Figure 6e still contains an inconsistency, with the mutation frequency bar graph on the left side appears to be showing data for a subset of 11 mutations, while the labels and the rest of the figure show 15 mutations. The figure legend does not match the figure colors (Figures 6D and 6E), specifically gray is not explained.

e. Given all these issues, the validity of this dataset is questionable. One solution can be to remove this section on transcriptome genotyping altogether and instead focus on the more robust portions of the already extensive manuscript.

Response: We have simply removed this (interesting) set of observations from the manuscript, figures, and methods, as not essential (although uncriticized by the other readers), to avoid further contention with the Reviewer and expedite acceptance of the work.

3. GWAS and transcriptome analysis:

a. It is still not completely clear how the authors are integrating the third party GWAS data with their transcriptome dataset. Figure 7C appears to show a Hallmark interferon gamma response GSEA, which is not derived from dataset uncovered by GWAS. Can the authors show a GSEA for the gene list containing their top 100 genes identified from GWAS? Is there evidence for upregulation of genes in that geneset? Figure 7d is difficult to interpret—it appears to show a small number (27-30) large dots overlaying the UMAPs corresponding to the disease scores. Lines 335-336 state that there are differences between controls and SAA patients, but no statistical analyses is provided to support that statement.

Response:

We have now added the detailed steps needed to generate Figure 7C-D (new Fig. 6c,d) and to clarify the procedures of integrating third party GWAS data with our scRNA-seq data, although this is an approach used in many other studies (Ma Y, et al. Genome Med. 2022; de Vries DH, et al. PLoS Pathog. 2020). The GSEA plot in Figure 7C (new Fig. 6c) is based on HMAGMA results of GWAS data, not scRNA-seq data.

Procedures of analysis of GWAS data at gene and pathway level are: 1) we downloaded the GWAS statistics results for AA, HMAGMA and related sources files; 2) we run HMAGMA with command: `magma --bfile ../MAGMA/g1000_eur --pval use=hm_rsid,p_value ncol=N --gene-annot` to test the joint association of all markers and local linkage disequilibrium to assign each gene a zscore; 3) we created rank file with HMAGMA zscore of genes. 4) we run

GSEA with the rank files (Figure 7C and now new Fig. 6c). 5) Then we used the top 200 genes of HMAGAMA, and used scDRS (single-cell disease-relevance score) to calculate the disease scores, and cells with high score were highlighted in UMAP in Fig. 7d (now new Fig. 6d). scDRS is a method for associating individual cells in single-cell RNA-seq data with disease GWASs (Zhang MJ, et al. Nat Genet. 2022). We now clarified in Methods “GWAS data processing and analysis” Lines 991-1000. Disease scores were similar across BMMNC cell types in controls, but highest in CD8+ T and NK cells rather than in myeloid cell types for SAA patients (Fig. 6d, fisher test, $P = 0.04$).

b. If inherited contribution of specific SNPs associated with AA is suspected (“An inherited genetic background appears to contribute to susceptibility to disease”), targeted genotyping of a small number of candidate germline loci for the 20 patients analyzed by transcriptomes could strengthen the conclusion of there being an inherited genetic background. Genotyping for these markers could be done quickly and inexpensively using such methods as Sanger sequencing.

Response: Sanger sequencing of SNPs is better utilized for direct validation of monogenic disease, and its use in a cohort 20 would raise further statistical concerns. For a complex disease in which each individual germline genes likely provide modestly to pathophysiology, results from a large cohort (>5000) are preferable.

4. Supplemental data should include information on detection and gating strategy for T regulatory cells using CyTOF panel. If the authors cannot support CyTOF data on Tregs with independently acquired standard multicolor flow cytometry of the same sample (as opposed to manually regating CyTOF data in flojo)--concerns that detection methods likely impacted T reg quantification leading to unexpectedly high Treg % in patients remain. Thus, the possibility of a technical issue leading to erroneously high Treg frequency should be more clearly stated when discussing T reg frequency in results.

Response: We added definition of Treg in legend of Extended Data Table 2. Surface markers for CyTOF that “Regulatory T cell (Treg), which is a subpopulation in CD4+ T cells, was defined as CD4+CD25+FOXP3+”.

Due to the explorative and prospective nature of the current study, and limited number of cells from patients, simultaneous explorative (CyTOF and scRNA-seq) and validation using a standard flow cytometry was impossible not. As for the unexpected Treg%, see Discussion lines 433-440 as below.

“While Treg numbers were not reduced compared to matched healthy donors and were not changed after therapy, frequency of Treg-B (specified by higher IL-2 receptor, FAS and CD45RO

expression) was higher in pre-treatment samples was surprising,²³ but most likely due to bone marrow rather than blood as the source of samples; regulatory T cells may relocate to the site of inflammation. Additionally, as our cohort included many patients with very severe AA, in which Tregs would be exposed to higher levels of inflammation, the most functional and proliferative Tregs expressing CD95 (FAS) would be sensitive to FASL-induced apoptosis and adopt a 'protective' phenotype by downregulating FAS and their functional suppressive capabilities.^{41,42}

Minor issues:

1. While the authors list the corrections to Figure 1B in their response to review, the corrections were not done, and the legend is still not matching the colors for Response and Relapse.

Response: We have now revised Fig. 1b as below.